# Cholesterol restriction primes antiviral innate immunity via SREBP1-driven noncanonical type I IFNs

Tasuku Nishimura [1,3], Takahisa Kouwaki [1,3✉], Ken Takashima [1,3], Akie Ochi[2], Yohana S Mtali [1] & Hiroyuki Oshiumi [1✉]

## Abstract

Cholesterol metabolism is associated with innate immune responses; however, the underlying mechanism remains unclear. Here, we perform chemical screening to isolate small molecules influencing RIG-I activity, a cytoplasmic viral RNA sensor. We find that statins, which inhibit cholesterol synthesis, dramatically enhance RIG-I-dependent antiviral responses in specific cell types. Since statins exhibit pleiotropic effects on type I interferon (IFN) responses, we further focus on their effects on RIG-I signaling. The restriction of cholesterol synthesis induces expression of noncanonical type I IFNs, such as IFN-ω, in an SREBP1 transcription factor-dependent manner. This pathway subsequently enhances RIG-I-mediated signaling following viral infection. Administration of statins augments RIG-I-dependent cytokine expression in the lungs of mice. Conversely, a mouse obesity model shows a diminished RIG-I response. Single-cell transcriptome analyses reveal a subset of alveolar macrophages that increase RIG-I expression in response to inhibited cholesterol synthesis in vivo. This study reveals SREBP1-mediated noncanonical type I IFN expression, linking cholesterol metabolism and RIG-I signaling.

Keywords RIG-I; Cholesterol; Type I Interferon; Virus; Innate Immunity
Subject Categories Immunology; Microbiology, Virology & Host Pathogen Interaction; Signal Transduction

## Introduction

The antiviral innate immune response is the first line of defense against viral infections. Recent studies have uncovered the integration of metabolism with immune responses, termed immunometabolism (Wang et al, 2019), and a growing body of evidence has demonstrated that cholesterol metabolism is linked to innate immune responses (Castrillo et al, 2003; Shin and Chung, 2023; York et al, 2015). Following the discovery of immunometabolism, recent cohort studies have revealed that metabolic syndromes increase the risk of severity and death in viral infectious diseases (Stefan et al, 2021). However, the mechanisms underlying this have not been fully elucidated.

Innate immune responses, including the expression of type I interferon (IFN), are critical for early-phase protection against viral infection (Bastard et al, 2020; Carvalho et al, 2021; Park and Iwasaki, 2020; Zhang et al, 2020). Viral nucleic acids are recognized by pattern recognition receptors (PRRs), such as RIG-I-like receptors (RLRs), Toll-like receptors (TLRs), and cGAS (Hur, 2019; Jenson and Chen, 2024; Miyake et al, 2018). TLRs, including TLR3, 7, and 8, play crucial roles in detecting viral RNAs in endosomes and lysosomes (Miyake et al, 2018). Conversely, cytoplasmic viral DNAs are targeted by cGAS, which utilizes the STING adaptor molecule to induce type I IFN expression (Jenson and Chen, 2024), while cytoplasmic viral double-stranded RNAs (dsRNAs) are specifically recognized by RLRs, including RIG-I and MDA5 (Hur, 2019). RIG-I activation is regulated by several ubiquitin ligases, such as Riplet and TRIM25 (Kato et al, 2021; Rehwinkel and Gack, 2020). This activation triggers the expression of type I IFNs through the MAVS adaptor molecule, which in turn induces the expression of interferon stimulatory genes (ISGs), including RIG-I itself (Yoneyama et al, 2024). RIG-I senses relatively shorter dsRNAs in the cytoplasm, whereas MDA5 preferentially recognizes longer dsRNAs (Kato et al, 2008). Consequently, RIG-I and MDA5 effectively recognize different types of viruses (Kato et al, 2006). For example, the influenza A virus, Sendai virus (SeV), and several other negative-sense single-stranded RNA viruses are detected by RIG-I. In contrast, picornaviruses (e.g., encephalomyocarditis virus (EMCV) and poliovirus) and coronaviruses (e.g., SARS-CoV and SARS-Cov-2) are primarily recognized by MDA5 (Liu et al, 2021; Thorne et al, 2021; Yoo et al, 2014).

Secreted type I IFNs bind to the type I IFN receptors on the cell surface, composed of IFNAR1 and IFNAR2 subunits, which trigger the expression of ISGs and induce antiviral activities (Gibbert et al, 2013). Several type I IFN genes in the human genome are clustered on chromosome 9, whereas in mice, this cluster is located on chromosome 4. Among these, IFN-α and -β are well-characterized type I IFNs, with more than 10 IFN-α genes encoded in the human genome (Pestka et al, 2004; Wittling et al, 2020). Specifically, IFN-α1 and -α2 expression are highly induced in human peripheral

[1]Department of Immunology, Graduate School of Medical Sciences, Faculty of Life Sciences, Kumamoto University, 1-1-1 Honjo, Kumamoto 860-8556, Japan. [2]School of Medicine, Kumamoto University, 1-1-1 Honjo, Kumamoto 860-8556, Japan. [3]These authors contributed equally: Tasuku Nishimura, Takahisa Kouwaki, Ken Takashima.
✉E-mail: kouwaki@kumamoto-u.ac.jp; oshiumi@kumamoto-u.ac.jp

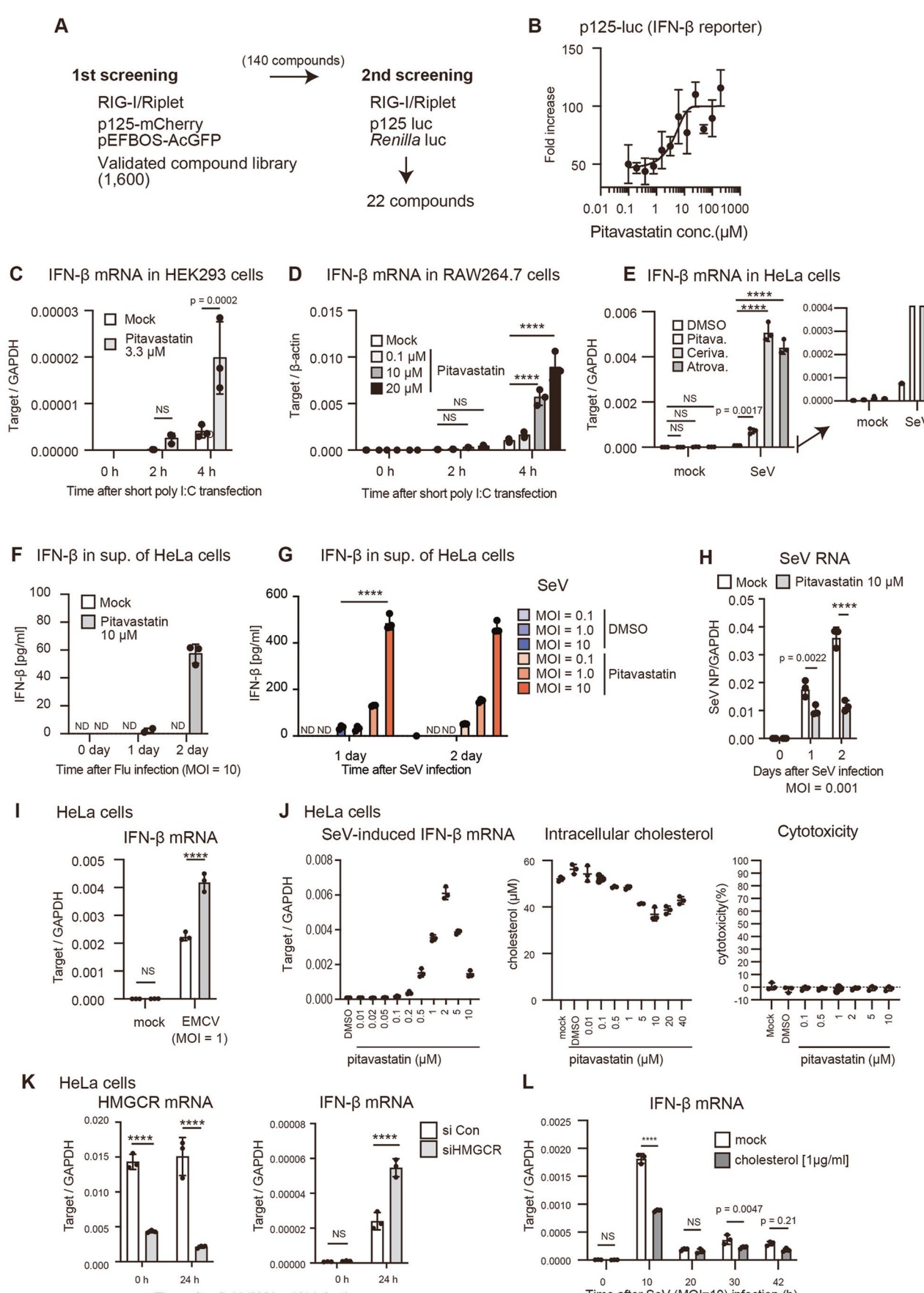

**Figure 1.  Inverse correlation between cellular cholesterol levels and RIG-I-dependent type I IFN expression levels.**

(A) A schematic representation of the chemical screening. (B) HEK293 cells were transfected with the p125-luc reporter (IFN-β promoter) and *Renilla* luciferase (internal control) together with RIG-I- and Riplet-expressing vectors. Pitavastatin was added to the cell-culture medium at the indicated concentrations (final concentration) and incubated for 24 h, and the luciferase activity was evaluated. The EC50 was determined to be 3.8 μM using nonlinear regression fitting of the dose-response curve ($n = 3$, biological replicates). (C, D) HEK293 and RAW264.7 cells were treated with pitavastatin at the indicated concentrations, and subsequently stimulated by short poly I:C (200 ng/mL) transfection. The expression of IFN-β mRNA was determined by RT-qPCR and normalized to that of GAPDH or β-actin as indicated ($n = 3$, technical replicates). (E) HeLa cells were treated with pitavastatin (2 μM), cerivastatin (5 μM), or atorvastatin (20 μM) and then infected with mock or Sendai virus (SeV) at MOI = 10 for 24 h. The expression of IFN-β mRNA was determined by RT-qPCR and normalized to that of GAPDH ($n = 3$, biological replicates). (F–I) HeLa cells were treated with pitavastatin or DMSO (mock) for 1 day and subsequently infected with the influenza A virus (Flu) (F), SeV (G, H), or EMCV (I) at indicated MOI for 1 or 2 days. IFN-β protein levels in cell culture sup. were determined by ELISA (F, G). Total cellular RNAs were collected, and type I IFN and viral RNA levels within cells were determined by RT-qPCR and normalized to cellular GAPDH levels (H, I) ($n = 3$, biological replicates, ND: not detected). (J) HeLa cells were treated with pitavastatin at the indicated concentrations for 24 h, and cellular cholesterol levels (center panel) and cytotoxicity (right panel) were determined. Pitavastatin-treated HeLa cells were infected with SeV at MOI = 10 for 24 h, and IFN-β mRNA levels were determined by RT-qPCR (left panel) ($n = 3$ biological replicates). (K) HeLa cells were transfected with siRNAs targeting HMGCR or control siRNA. After 2 days, the cells were infected with SeV at MOI = 10. The mRNA expression of HMGCR and IFN-β was determined by RT-qPCR and normalized to that of GAPDH ($n = 3$, biological replicates). (L) Cholesterol was added to the HeLa cell culture medium for 24 h, and the cells were infected with SeV. The expression of IFN-β mRNA was determined by RT-qPCR and normalized to that of β-actin ($n = 3$, technical replicates). Data Information: In (B–L), data are represented as mean ± standard deviation (SD). Statistical significance was determined by one-way ANOVA (J) and two-way ANOVA (C, D, E, G–I, K, L). NS: not significant, ****$p < 0.0001$. Source data are available online for this figure.

blood mononuclear cells (PBMCs) following infection with several viruses (Loseke et al, 2003), and various types of viruses induce IFN-β expression in macrophages as well as epithelial cells and fibroblasts (Oshiumi et al, 2010). These viral-induced type I IFNs would be called canonical type I IFNs, whereas there are several so-called noncanonical type I IFNs, including IFN-ε, -κ, and -ω, as well as minor IFN-α subsets (including IFN-α16) whose roles are less clear compared to the canonical type I IFNs because of their weak expression. However, previous studies have reported that patients with life-threatening COVID-19 possessed neutralizing autoantibodies against IFN-ω and IFN-α (Bastard et al, 2020), suggesting the crucial roles of the noncanonical type I IFN. Because noncanonical type I IFNs also stimulate the type I IFN receptor (Gibbert et al, 2013), their distinct roles remain unclear.

The correlation between metabolic diseases and immune responses to viral infections is noteworthy (Rohm et al, 2022), and recent studies have revealed that obesity increases the risk of COVID-19 severity and death (Simonnet et al, 2020; Williamson et al, 2020). Moreover, retrospective cohort studies have shown that the use of statins, which inhibit cholesterol biosynthesis, reduces the risk of severity in patients with COVID-19 (Gupta et al, 2021). Increasing evidence has begun to clarify the molecular mechanisms that link cholesterol metabolism to antiviral innate immunity. Type I IFN signaling itself regulates cholesterol biosynthesis, and disturbances in cholesterol biosynthesis are associated with an enhanced type I IFN response in a STING and SREBP2-dependent manner (Shin and Chung, 2023; York et al, 2015). The addition of a 20-carbon geranylgeranyl lipid to MAVS by geranylgeranyl transferase type I significantly attenuates type I IFN expression following viral infection, and this geranylgeranylation is associated with cholesterol metabolism (Akula et al, 2019; Yang et al, 2019). In contrast, previous studies have reported that statins reduce surface type I IFN receptors, thereby inhibiting I IFNs (Tenesaca et al, 2021; Vasquez et al, 2016). In addition, scavenger receptor class B type I links cholesterol metabolism and type I IFN response (Tenesaca et al, 2021; Vasquez et al, 2016). These studies suggest the intimate association between cholesterol metabolism and type I IFN responses. However, the mechanism through which cholesterol restriction boosts the antiviral response has not been fully elucidated, and the physiological significance of these mechanisms in vivo remains elusive.

In this study, we performed a chemical screening to isolate small molecules that enhance RIG-I-dependent type I IFN expression and identified statins. Since statins dramatically enhanced RIG-I-dependent innate immune responses in specific cell types, we aimed to uncover mechanisms linking cholesterol metabolism with the RIG-I-dependent antiviral responses and assess their physiological significance in vivo using mouse models.

## Results

### Statins dramatically enhance RIG-I-dependent antiviral responses

To identify small molecules that modulate RIG-I-mediated type I IFN expression signaling, we developed a chemical screening assay system in which the activation of RIG-I was induced by ectopic expression of the Riplet protein, a positive regulator for RIG-I (Cadena et al, 2019). RIG-I/Riplet signaling led to enhanced mCherry reporter protein expression, driven by IFN-β promoter, with AcGFP, driven by EF1α promoter, serving as an internal control. Approximately 1600 approved compounds were screened, and those exhibiting more than a 1.5-fold difference in the mCherry/AcGFP intensity ratio compared to the control (DMSO) were chosen for the second screening, totaling 140 compounds (Fig. 1A). The second screening was conducted using the plasmid-carrying luciferase gene driven by the IFN-β promoter (p125-luc) to eliminate artificial fluorescence. Twenty-two compounds exhibited significant differences in the luciferase activity compared to the control and were selected as candidates (Fig. 1A). Among these, pitavastatin, atorvastatin, and cerivastatin were included. To confirm the effect of pitavastatin, we investigated the dose dependency and found that pitavastatin treatment significantly increased the activation of the IFN-β promoter by RIG-I in a dose-dependent manner in HEK293 cells (Fig. 1B).

To further assess the effects of statins on RIG-I signaling, we measured the mRNA expression and protein production of cytokines after RIG-I activation in statin-treated cells. Short poly I:C is a ligand of RIG-I, and treatment of cells with pitavastatin or cerivastatin markedly enhanced the expression of IFN-β, IL-6, and

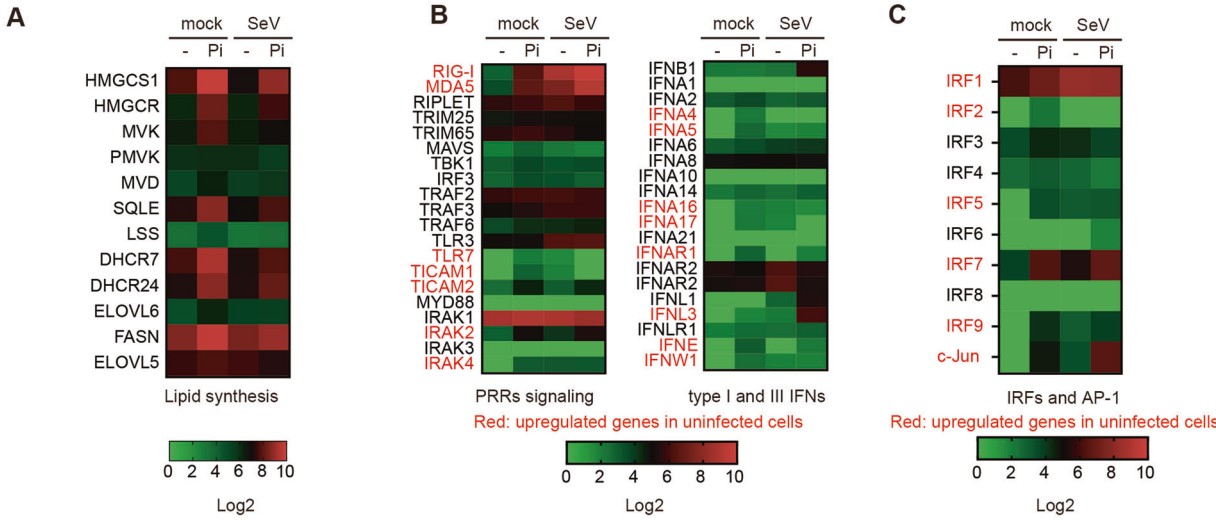

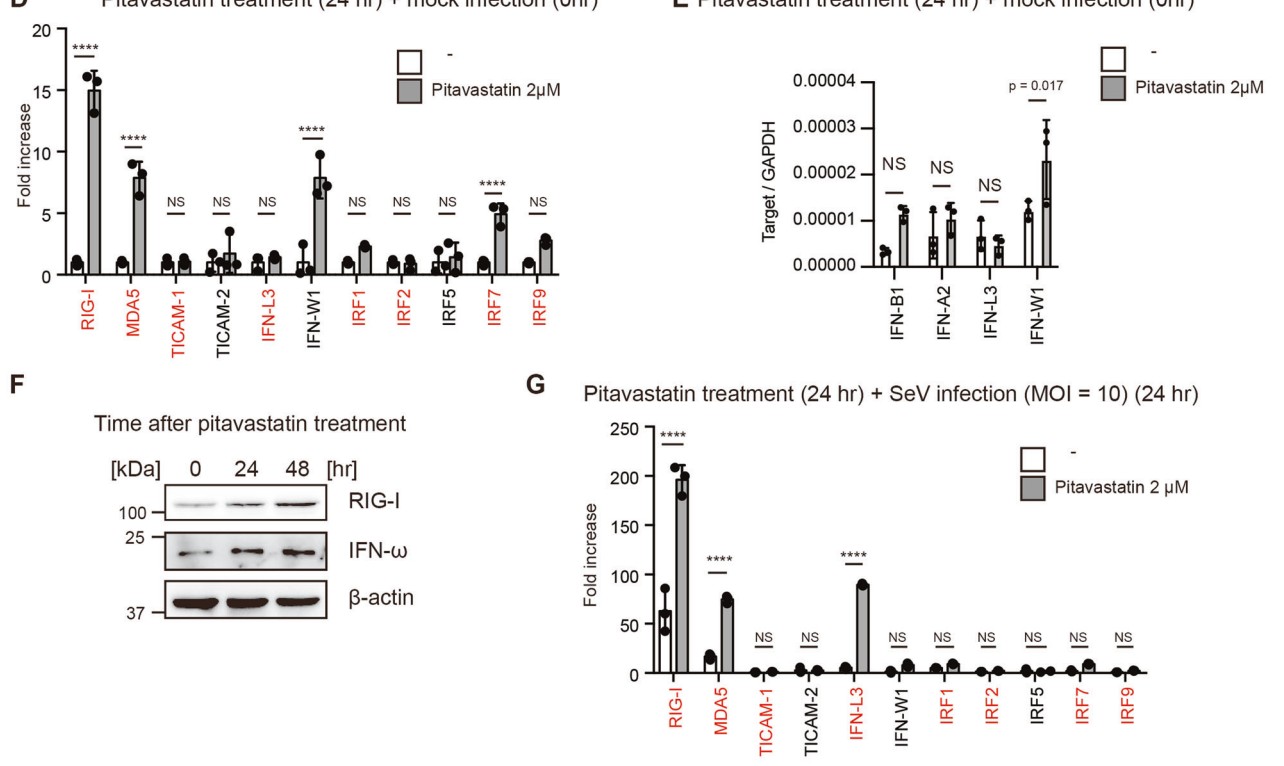

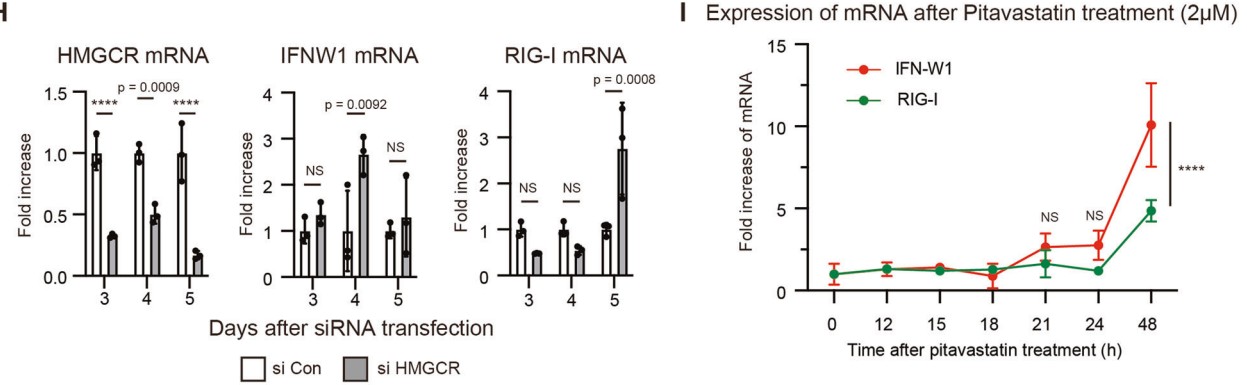

**Figure 2. Pitavastatin-induced expression of RIG-I and non-canonical type I IFNs in the absence of cell stimulation.**

(A–C) Heat maps of gene expression in pitavastatin-treated HeLa cells with or without SeV infection. HeLa cells were treated with pitavastatin (Pi) for 24 h, followed by infection with SeV (MOI = 10) or a mock treatment. Gene expression profiles of each sample were determined by a microarray analysis. (D, E) HeLa cells were treated with pitavastatin for 24 h. The expression of each gene was determined by RT-qPCR and normalized to that of β-actin or GAPDH. The fold increase was calculated by dividing the normalized expression level of a pitavastatin-treated sample by the corresponding value of a non-treated sample. ISGs were shown in red (n = 3, biological replicates). (F) HepG2 cells were treated with 2 µM of pitavastatin. Whole-cell lysates were prepared at indicated time points and subjected to SDS-PAGE. The proteins were detected by western blotting with anti-RIG-I, IFN-ω, and β-actin. (G) HeLa cells were treated with pitavastatin for 24 h, followed by infection with SeV (MOI = 10) for 24 h. The expression of each gene was determined by RT-qPCR and normalized to that of β-actin. The fold increase was calculated by dividing the normalized expression level of a pitavastatin-treated sample by the corresponding value of a non-treated sample. ISGs were shown in red (n = 3, biological replicates). (H) HeLa cells were transfected with siRNAs for negative control (si Con) or HMGCR (si HMGCR), and then the expression of genes was measured by RT-qPCR and normalized to GAPDH. Fold increase was calculated by dividing normalized mRNA levels at each time point by those at day 3 of si Con transfected sample (n = 3, biological replicates). (I) HeLa cells were treated with pitavastatin or a solvent (DMSO), and the expression of RIG-I and IFN-W1 at the indicated time points was determined by RT-qPCR and normalized to that of β-actin. The fold increase was calculated by dividing the mRNA levels of pitavastatin-treated cells by those of DMSO-treated cells. *P*-vales were calculated between IFN-W1 and RIG-I at each time point (n = 3, technical replicates). Data Information: In (D–I), data are represented as mean ± SD (NS: not significant, ****p < 0.0001, two-way ANOVA). Source data are available online for this figure.

IP10 mRNA in HEK293, RAW264.7, and HeLa cells following short poly I:C transfection or infection with Sendai virus (SeV), which is recognized by RIG-I (Kato et al, 2006; Lopez et al, 2004) (Figs. 1C–E and EV1A–C). The influenza A virus (Flu) is also mainly recognized by RIG-I (Kato et al, 2006; Lopez et al, 2004), and pitavastatin treatment increased the levels of IFN-β proteins produced by HeLa cells in response to infection with Flu or SeV (Figs. 1F,G and EV1D), although Flu infection alone hardly produced the IFN-β protein in HeLa cells under our experimental condition. Corresponding with the increased IFN-β production, pitavastatin treatment significantly reduced viral RNA levels within infected cells (Fig. 1H). We confirmed that knockdown of MAVS reduced the IFN-β mRNA expression (Appendix Fig. S1A,B). These data suggested that pitavastatin treatment markedly enhanced RIG-I-mediated cytokine expression.

EMCV is primarily recognized by MDA5. Pitavastatin treatment also enhanced EMCV-induced type I IFN expression (Fig. 1I). Next, we assessed the effects on the other PRRs. HeLa cells express TLR3 (Oshiumi et al, 2003). The addition of poly I:C to the cell culture medium leads to TLR3 activation (Matsumoto and Seya, 2008), and the mRNA expression of IFN-β and IFN-L3 was enhanced by pitavastatin treatment (Fig. EV1E,F). We confirmed that siRNA for TLR3 attenuated type I IFN expression induced by adding poly I:C to the cell culture medium (Fig. EV1G). These data suggest that the effect of pitavastatin is not specific to RIG-I signaling. In contrast, cGAMP, a ligand of STING, failed to increase the expression of IFN-β mRNA in HeLa cells, irrespective of the presence or absence of pitavastatin (Fig. EV1H), implying the cGAS/STING-independent mechanism. We confirmed that cGAMP could induce type I IFN expression in RAW264.7 cells under our experimental condition (Appendix Fig. S1C). In the case of RAW264.7 cells, pitavastatin treatment enhanced IFN-β mRNA expression induced by adding poly I:C to the cell culture medium (Appendix Fig. S1D). In addition, we found that IFN-β mRNA expression induced by LPS stimulation was also enhanced by treatment with a higher dose of pitavastatin (Appendix Fig. S1E). These data suggest that pitavastatin treatment enhances the RIG-I-dependent pathway, with additional effects on MDA5-, TLR3-, and TLR4-mediated signaling.

Considering that pitavastatin inhibits cholesterol synthesis, we hypothesized that cellular cholesterol levels would influence RIG-I-mediated innate immune responses. Firstly, we compared cellular cholesterol levels with the expression levels of type I IFN at various pitavastatin concentrations to evaluate this hypothesis. The results showed that ~1–5 µM of pitavastatin was sufficient to reduce cellular cholesterol levels in HeLa cells, and this same pitavastatin concentration could enhance type I IFN expression following SeV infection (Fig. 1J). Furthermore, we confirmed that these concentrations did not induce cytotoxicity (Fig. 1J, right panel). This correlation was also observed in HepG2 and A549 cells (Fig. EV1I,J), although the effective concentrations differed between these cell lines. A similar correlation was observed when cerivastatin was used instead of pitavastatin (Fig. EV1K). To further explore the correlation between cellular cholesterol levels and type I IFN expression, we performed knockdown experiments using siRNA targeting HMGCR, a target of statins. The siRNA for HMGCR markedly reduced HMGCR mRNA and enhanced RIG-I-dependent IFN-β mRNA expression in HeLa and HepG2 cells (Figs. 1K and EV1L,M). Conversely, the addition of cholesterol to the cell culture medium reduced SeV-induced IFN-β expression (Fig. 1L). These data indicate that cellular cholesterol levels are inversely correlated with type I IFN expression levels following RIG-I stimulation.

## Treatment with statins induces noncanonical type I IFN expression

To investigate the mechanism underlying these above findings, we performed a microarray analysis of HeLa cells before and after SeV infection with or without pitavastatin treatment. We confirmed that pitavastatin treatment enhanced the expression of genes essential for lipid synthesis, including cholesterol synthesis (Fig. 2A). Following this, we focused on the expression of genes involved in the innate immune response. Interestingly, several noncanonical type I IFN genes, including those for IFN-W1 (IFN-ω), IFN-A5, IFN-A16, IFN-A17, IFN-L3, and IFN-E, as well as IRF transcription factors and RLRs, were moderately upregulated by pitavastatin pretreatment in the absence of SeV infection (Fig. 2B,C). In contrast, IFN-B1, a canonical type I IFN, was markedly upregulated only after SeV infection (Fig. 2B). These data suggest that pitavastatin treatment leads to the expression of noncanonical type I IFNs and RLRs before viral infection. Along with the expression of noncanonical type I IFNs, subsets of ISGs were upregulated by pitavastatin treatment even in the absence of SeV infection (Fig. EV2A).

RT-qPCR was conducted to confirm the results of the microarray analysis. The increases in the expression of RIG-I,

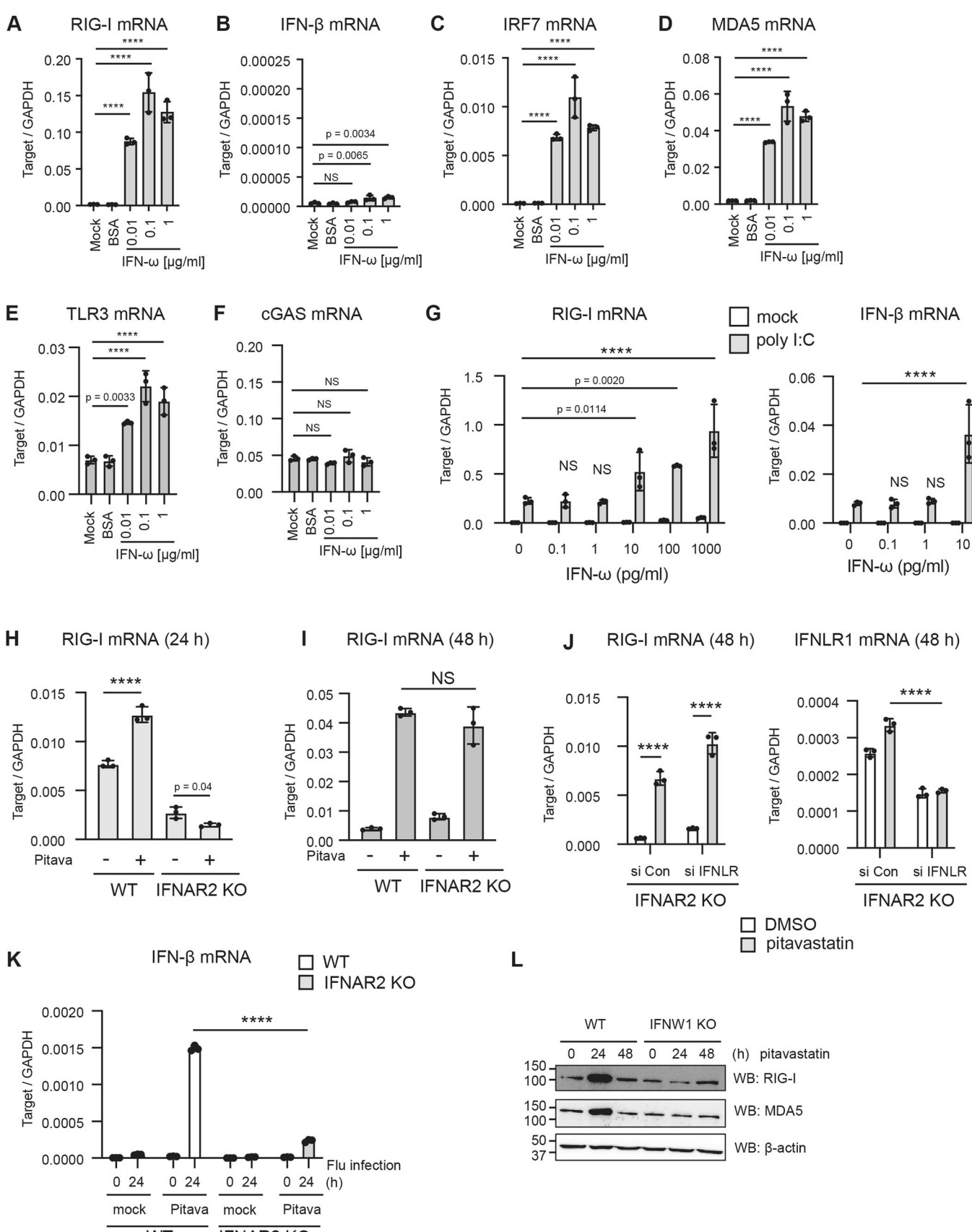

**Figure 3.  The type I IFN receptor is involved in statin-mediated enhancement of RIG-I signaling.**

(A–F) HeLa cells were treated with recombinant IFN-ω protein at the indicated concentration for 24 h, and expression of each gene was determined by RT-qPCR and normalized to that of GAPDH (n = 3, technical replicates). (G) HeLa cells were treated with recombinant IFN-ω protein at the indicated concentration for 24 h, and subsequently stimulated via short poly I:C transfection. Four hours after transfection, IFN-β mRNA expression was determined by RT-qPCR and normalized to that of GAPDH (n = 3, biological replicates). (H, I) Wild type (WT) and IFNAR2 knockout (KO) HeLa cells were treated with or without pitavastatin for 24 (H) or 48 h (I), and the expression of RIG-I was determined by RT-qPCR and normalized to that of GAPDH (n = 3, biological replicates). (J) siRNA for negative control (si Con) and IFNLR1 (si IFNLR) were transfected into IFNAR2 KO cells for 2 days, and then the cells were treated with pitavastatin for 48 h. The expression of RIG-I and IFNL1 was determined by RT-qPCR and normalized to that of GAPDH (n = 3, biological replicates). (K) WT and IFNAR2 KO HeLa cells were treated with pitavastatin (Pitava) or DMSO (mock) for 24 h, and then infected with Flu at MOI = 10. The expression of IFN-β mRNA at indicated time points was determined by RT-qPCR and normalized to that of GAPDH (n = 3, biological replicates). (L) WT and IFN-W1 KO HeLa cells were treated with pitavastatin at indicated h. Cell lysates were prepared and were then subjected to SDS-PAGE. The proteins were detected by western blotting with indicated antibodies. Data Information: In (A–K), data are represented as mean ± SD. Statistical significance was determined using one-way ANOVA (A-F, H, I) and two-way ANOVA (G, J, K). ****p < 0.0001. NS: not significant. Source data are available online for this figure.

MDA5, and IFN-W1 mRNA expression following pitavastatin treatment in the absence of SeV infection were statistically significant (Fig. 2D). Furthermore, pitavastatin treatment also significantly increased IFN-A16 mRNA expression without SeV infection (Fig. EV2B). In contrast, canonical type I IFNs, such as IFN-β and α2, were not induced by pitavastatin treatment in uninfected cells (Fig. 2E). We confirmed that the protein levels of IFN-ω and RIG-I were increased by pitavastatin treatment even in the absence of viral infection (Fig. 2F). Next, we measured the gene expression in virus-infected cells. The increase in IFN-W1 expression following SeV infection was moderate, whereas RIG-I and MDA5 expression were markedly increased following SeV infection (Fig. 2G). Based on these observations, we further investigated whether HMGCR knockdown could induce the expression of those genes even in the absence of viral infection as pitavastatin did. We found that siRNA for HMGCR significantly induced the mRNA expression of IFN-W1 and RIG-I (Fig. 2H). In addition, MAVS knockdown failed to attenuate pitavastatin-induced RIG-I expression in the absence of viral infection (Fig. EV2C), implying that geranylgeranylation of MAVS is not involved in statin-induced RIG-I expression. Collectively, these data indicate that pitavastatin treatment induces the expression of RIG-I and noncanonical type I IFNs prior to viral infection.

Pitavastatin treatment increased IFN-W1 mRNA levels in HeLa and HepG2 cells but not in A549 and THP-1 cells under the experimental condition (Fig. EV2D), suggesting the cell-type specific effects of statins by unidentified mechanisms (see Discussion). Thus, we selected specific cell types that were responsive to statin treatment for the following studies to better understand the mechanism of statin-induced effects.

Because RIG-I is an interferon-inducible gene, we compared the kinetics of IFN-W1 and RIG-I gene expression after pitavastatin treatment and found that RIG-I expression was delayed compared with IFN-W1 expression (Fig. 2I), suggesting that RIG-I expression occurs following IFN-W1 expression. Based on the above data, we anticipated that the initial expression of noncanonical type I IFNs would induce subsequent RIG-I expression. To evaluate this hypothesis, cells were treated with the IFN-ω recombinant proteins, and RIG-I expression was measured. As expected, stimulation with the IFN-ω protein markedly induced the RIG-I expression (Fig. 3A). However, at IFN-β mRNA expression was hardly induced by the IFN-ω protein at 0.01 μg/ml, while higher concentrations showed slight induction (Fig. 3B). Recombinant IFN-ω protein also increased the expression of IRF7, MDA5, and TLR3, but not cGAS mRNAs (Fig. 3C–F). Adding 10 pg/mL of recombinant IFN-ω

protein was sufficient to enhance the expression of IFN-β and RIG-I mRNA following short polyI:C stimulation, similar to the effect of pitavastatin (Figs. 3G and EV3). If pitavastatin treatment increases RIG-I expression via noncanonical type I IFNs, it is expected that pitavastatin-induced RIG-I expression will be impaired by a defect in the type I IFN receptor. IFNAR2 is an essential component of the type I IFN receptor, and we generated IFNAR2 knockout (KO) cells using the CRISPR-Cas9 system (Appendix Fig. S2). When cells were treated with pitavastatin for 24 h, IFNAR2 KO abolished pitavastatin-induced RIG-I expression (Fig. 3H), suggesting that the type I IFN receptor was involved in pitavastatin-induced RIG-I mRNA expression. However, 48 h treatment with pitavastatin increased RIG-I expression even in IFNAR2 KO cells (Fig. 3I), implying the existence of both type I IFN receptor-dependent and -independent pathways. Knockdown of type III IFN receptor did not reduce the RIG-I expression in IFNAR2 KO cells at the later time point (Fig. 3J). Notably, pitavastatin-mediated enhancement of IFN-β mRNA expression following Flu infection was markedly reduced by IFNAR2 KO (Fig. 3K). Interestingly, IFN-W1 KO impaired pitavastatin-induced expression of the RIG-I and MDA5 proteins (Fig. 3L). These data are consistent with our hypothesis that initially produced noncanonical type I IFNs, including IFN-ω, induce subsequent RIG-I expression via the type I IFN receptor.

## The noncanonical type I IFN expression pathway is dependent on the SREBP1 transcription factor

A decrease in cellular cholesterol levels is known to induce the activation of the transcription factors, SREBP1 and SREBP2 (Shimano, 2001). Therefore, we investigated the roles of SREBP1 and SREBP2 in this process. First, we measured promoter activity using reporter assay with constitutively active forms of the SREBPs (SREBP1a-N, 1c-N, and 2-N) and RIG-I (RIG-I CARDs). Ectopic expression of RIG-I CARDs markedly activated the IFN-β promoter but not the IFN-ω promoter (Fig. 4A). In contrast, ectopic expression of SREBP1a-N, 1c-N, and 2-N markedly activated the IFN-ω promoter (Fig. 4A). The IRF7 promoters were also activated by SREBP1a-N and 2-N, but not by RIG-I CARDs (Fig. 4A). The RIG-I promoter itself was not activated by the ectopic expression of SREBP1a-N, 1c-N, and 2-N, although the promoter was effectively activated by treatment with IFN-α (Fig. 4A). The ectopic expression of MAVS, MyD88, and STING activated the IFN-β promoter but hardly activated the IFN-ω promoter, while SREBP1a-N induced the activation of the IFN-ω

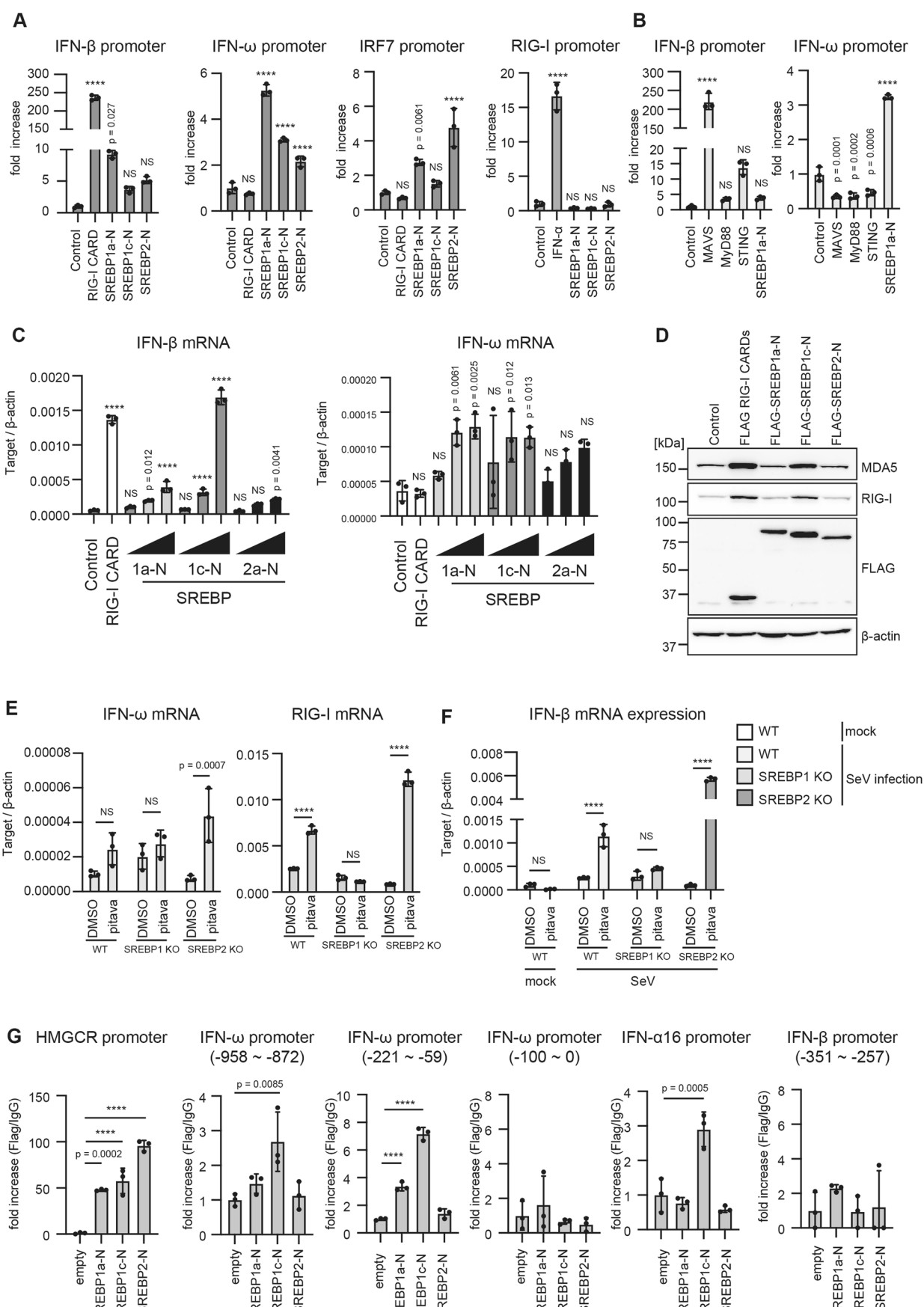

Figure 4. SREBP1 is essential for stain-mediated enhancement of RIG-I signaling.

(A) HEK293 cells were transfected with either the reporter plasmid for each gene promoter along with the expression vector expressing the indicated gene and an empty vector (control) or stimulated with 100 U of IFN-α. Twenty-four hours after transfection or stimulation, cells were lysed, and the activity of each promoter was determined by measuring luciferase activities. The fold increase was calculated by dividing the luciferase activity of each sample by the average luciferase activity of the control samples. Each p-value represents the comparison with the control (n = 3, biological replicates). (B) HEK293 cells were transfected with a p125-luc (IFN-β promoter) or IFN-ω promoter reporter plasmid along with a *Renilla* luciferase-expressing vector (internal control) and MAVS, MyD88, STING, or SREBP1a-N expressing vector. Twenty-four hours after transfection luciferase activities were measured and normalized to *Renilla* luciferase activity. The statistical analyses compared the expression of genes induced by the proteins with that of control. Each p-value represents the comparison with the control (n = 3, biological replicates). (C, D) HEK293 cells were transfected with 200 ng/mL of RIG-I CARD expressing vectors along with 200, 1000, or 2000 ng/mL of SREBP1a-N, 1c-N, or 2a-N expressing vectors in a 24-well plate. The total amount of DNAs was normalized to the same amount by adding the empty vector. Twenty-four hours after transfection, the expression of each gene was determined by RT-qPCR and normalized to that of β-actin (C) (n = 3, technical replicates), or cell lysates were prepared and subjected to SDS-PAGE and stained by western blotting with the indicated antibodies (D). Each p-value represents the comparison with the control. (E, F) WT, SREBP1 KO, and SREBP2 KO HeLa cells were treated with pitavastatin (pitava) and solvent (DMSO) for 24 h. Following this, the expression of each gene was determined by RT-qPCR and normalized to that of β-actin (E), or treated cells were infected with SeV for 24 hr at MOI = 10, and the expression of IFN-β mRNA was determined by RT-qPCR and normalized to that of β-actin (F) (n = 3, biological replicates). (G) FLAG-tagged SREBP1a-N, SREBP-1c-N, and SREBP2-N were transfected into HEK293 cells. A chromatin immunoprecipitation (ChIP) assay was performed using anti-FLAG antibodies. The binding of the protein to the indicated DNA regions was detected by qPCR (n = 3, technical replicates). The data shown are from a representative experiment. Data Information: In (A–C, E–G), data are presented as mean ± SD. Statistical significance was determined using one-way ANOVA (A–C, G) and two-way ANOVA (E, F). ****p < 0.0001. NS: not significant. Source data are available online for this figure.

promoter (Fig. 4B). These data suggested that the regulation of IFN-ω expression was distinct from that of IFN-β expression.

Next, we measured the expression of endogenous IFN-β and IFN-ω genes. Ectopic expression of RIG-I CARDs induced the mRNA expression of IFN-β, along with that of IP-10, IRF7, and RIG-I but not the mRNA expression of IFN-ω (Figs. 4C and EV4A–D). In contrast, ectopic expression of SREBP1a-N and SREBP1c-N increased the mRNA expression of IFN-ω and those of RIG-I, HMGCR (Figs. 4C and EV4A–D), and RIG-I protein levels (Fig. 4D). Moreover, short poly I:C stimulation of HeLa cells failed to increase the mRNA expression of IFN-ω but did increase IFN-β mRNA expression (Fig. EV4E), although it could induce IFN-ω expression in THP-1 cells (Fig. EV4F). These data indicate that SREBP1 and SREBP2, but not RIG-I, have the potential to activate the promoter of the IFN-ω gene. SREBP1a-N could modestly induce IFN-β promoter activation (Fig. 4B), and SREBP1c-N also could increase the expression of IFN-β mRNA (Fig. 4C). Thus, we do not exclude the possibility that SREBP1 has the ability to induce IFN-β expression (see Discussion).

To further assess the role of SREBPs on IFN-ω expression, we generated SREBP1 and SREBP2 KO cells using the CRISPR-Cas9 system (Appendix Fig. S3). Pitavastatin-induced IFN-ω expression was abolished by SREBP1 KO but not by SREBP2 KO (Fig. 4E). Pitavastatin-treatment-induced expression of RIG-I mRNA was also abrogated by SREBP1 KO (Fig. 4E). Moreover, SREBP1 KO abolished the effect of pitavastatin treatment on RIG-I-dependent IFN-β mRNA expression following SeV infection (Fig. 4F). Furthermore, ectopically expressed SREBP1 could compensate for the defects (Fig. EV4G). These findings indicate that SREBP1 is responsible for the effect of pitavastatin on RIG-I signaling.

To confirm the involvement of SREBP1 in the expression of noncanonical type I IFNs, we performed a chromatin immunoprecipitation (ChIP) assay. The SRE and E-box motifs serve as the binding sites for SREBPs (Amemiya-Kudo et al, 2002) and are located in the promoter regions of IFN-ω and IFN-α16 (Fig. EV4H,I). The ChIP assay showed that SERBP1a-N and SREBP1c-N could bind to the IFN-ω promoter as well as the IFN-α16 promoter; however, SREBP2-N failed to bind to the IFN-ω and IFN-α16 promoters (Fig. 4G). We could not detect the binding of either SREBP1 or 2 to the E-box in the −351 to −257 region of the

IFN-β promoter (Fig. 4G; Appendix Fig. S4). Collectively, these data support the model that SREBP1 is involved in noncanonical type I IFN expression. There are potential SRE elements and E-boxes in the promoter region of type I IFNs as well as type III IFNs and IRF7 (Appendix Fig. S4), and thus the Fig. 4G data also implies that several E-box or SRE elements are not functional in this experimental condition. In addition, we confirmed that the IFN-ω promoter reporter was activated by pitavastatin treatment (Fig. EV4J).

Since noncanonical type I IFNs, but not type III IFNs, were important for the association between cholesterol metabolism and antiviral innate immune responses, we examined whether type II IFN signaling is also associated with cholesterol metabolism. IFN-γ is known to induce the expression of IP10, as shown in Appendix Fig. S5A. Interestingly, the ChIP-Atlas public database (Zou et al, 2024) suggested the presence of potential SREBP1, but not SREBP2, binding sites at their promoter regions. Therefore, we examined whether pitavastatin treatment affects IFN-γ-mediated IP10 expression. Pitavastatin treatment moderately enhanced IFN-γ-mediated IP10 expression (Appendix Fig. S5). These findings suggest that pitavastatin affects not only PRR-mediated innate immune response via noncanonical type I IFNs but also IFN-γ-mediated chemokine expression to some extent (see Discussion).

## Effect of cholesterol restriction on antiviral innate immunity in vivo

The mouse genome also encodes noncanonical type I IFN genes, although the IFN-ω gene is not conserved. In bone-marrow-derived macrophages of mice, pitavastatin treatment increased the expression of noncanonical type I IFNs, such as IFN-η, IFN-κ, and IFN-α5 (Fig. EV5A). This treatment enhanced the expression of IFN-β following poly I:C stimulation (Fig. EV5B). Next, we assessed whether the restriction of cholesterol synthesis augments type I IFN expression in vivo using mouse animal models. Pitavastatin was intraperitoneally administered to mice every three days, followed by the intraperitoneal injection of poly I:C. Since the IFN-W1 gene is not conserved in mice, we examined the expression of all the other type I IFNs, as well as RIG-I. Although we could not detect a significant increase in the expression of RIG-I and type I IFNs in

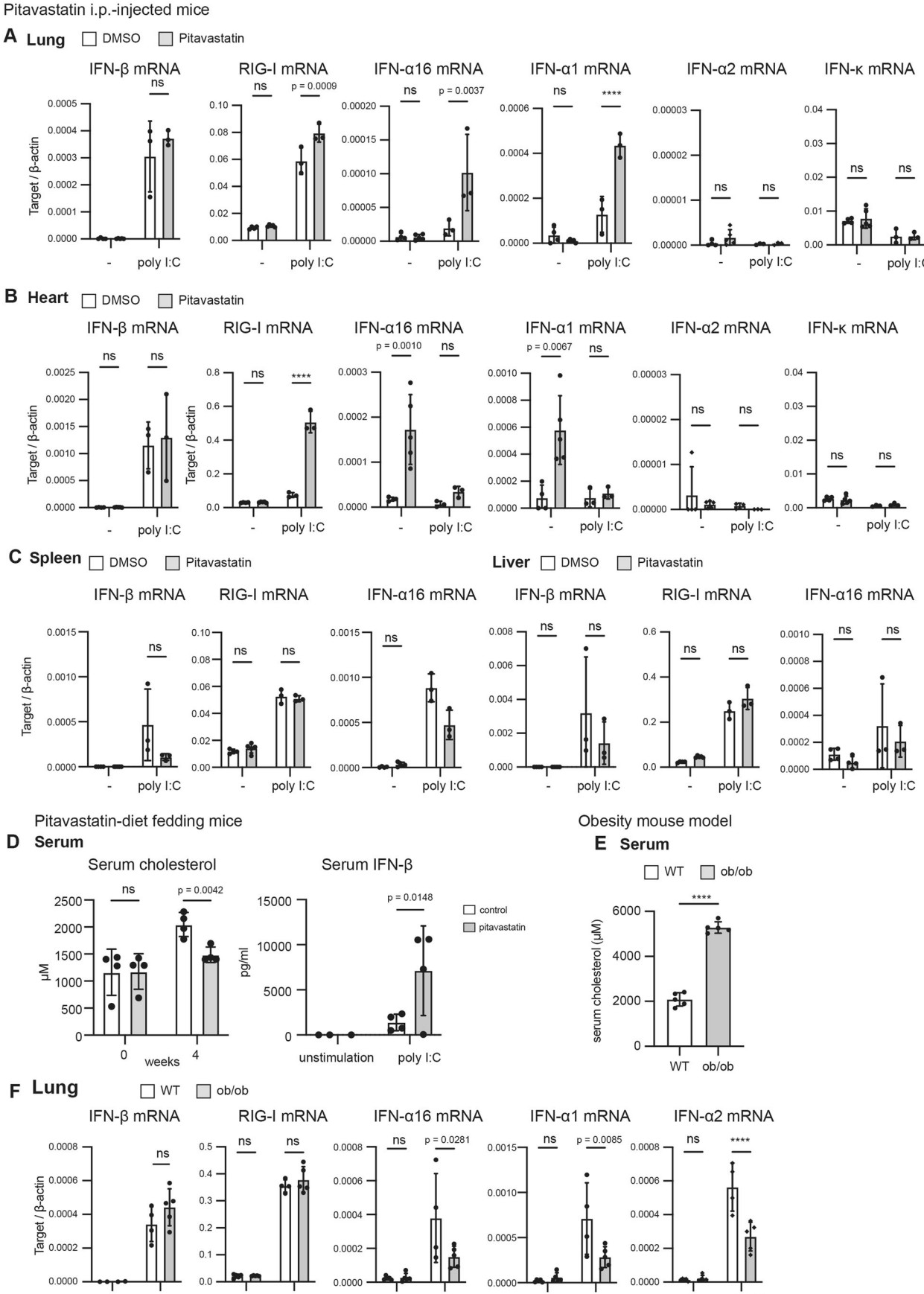

© The Author(s)

Figure 5. Tissue-specific effects of pitavastatin on RIG-I and type I IFN gene expression.

(A–C) Pitavastatin or solvent (DMSO) was intraperitoneally injected into mice every day for three days, followed by an intraperitoneal injection of poly I:C (200 µg/head) or solvent (DMSO). Twenty-four hours after poly I:C injection, the lungs (A), heart (B), and spleen (C) were isolated. The total RNA was extracted from each tissue sample. The expression of each gene was determined by RT-qPCR and normalized to that of β-actin ($n = 3$–5 mice for each group). (D) Mice were fed a normal or pitavastatin-containing diet for four weeks, following which they received intraperitoneal injections of poly I:C. Sera were collected at each time point, and serum cholesterol and IFN-β protein levels were determined ($n = 4$ mice for each group). (E, F) Sera from WT and ob/ob mice were collected, and serum cholesterol levels were determined (E). Poly I:C was intraperitoneally injected into WT and ob/ob mice. Six hours after injection, the mRNA expression of RIG-I and type I IFNs mRNA in the lung was determined by RT-qPCR and normalized to that of β-actin ($n = 5$ mice for each group). Data Information: In (A–F), data are represented as mean ± SD. Statistical significance was determined using two-way ANOVA (A–D, F) and t-test (E). ****$p < 0.0001$. ns: not significant. Source data are available online for this figure.

the lungs before poly I:C intraperitoneal injection, pitavastatin administration augmented the expression of RIG-I, IFN-α1, IFN-α16, and several IFN-α subtypes in the lung following poly I:C intraperitoneal injection (Figs. 5A and EV5C). The augmented expression of RIG-I and type I IFNs before and after poly I:C intraperitoneal injection was detected in the heart (Fig. 5B). However, this augmentation was not observed in the spleen, liver, muscle, or uterus (Figs. 5C and EV5C), suggesting tissue specificity.

Next, we used another mouse model, in which mice were fed a pitavastatin-containing diet for 4 weeks, after which they were injected with poly I:C. We confirmed that this treatment reduced blood cholesterol levels (Fig. 5D). Notably, the pitavastatin-containing diet markedly augmented serum IFN-β protein levels after poly I:C injection (Fig. 5D).

Following this, we used a mouse model of obesity (ob/ob mouse), and we confirmed that serum cholesterol levels were higher in the ob/ob mice, which spontaneously developed obesity (Zhang et al, 1994), than those in wild-type mice (Fig. 5E). Interestingly, IFN-α1, -α2, and -α16 expression in the lung after poly I:C injection was attenuated in ob/ob mice compared with WT mice (Fig. 5F). Collectively, these data suggest that cholesterol levels are inversely correlated with the expression of polyI:C-induced type I IFN and RIG-I in the lungs.

## Identification of CD140a⁺ alveolar macrophages priming the RIG-I pathway in response to cholesterol restriction

In the heart, we detected the increase in the expression of a noncanonical type I IFN, IFN-α16, by pitavastatin treatment without poly I:C stimulation. However, induced expression of noncanonical type I IFNs and RIG-I by pitavastatin treatment alone was not detectable in the lungs of the above animal models. In addition, we could not detect the increase in the expression of noncanonical type I IFNs in the lungs of pitavastatin-feeding mice, although serum IFN-β levels were enhanced by pitavastatin (Fig. 5D). Therefore, we anticipated that only a small subset of lung cells expressed RIG-I in response to pitavastatin administration before poly I:C injection. We expected that single-cell transcriptome analysis of the mouse lung would reveal the RIG-I expression in response to inhibited cholesterol synthesis in specific cell subtypes.

To investigate cell populations that express RIG-I in response to pitavastatin administration, we conducted a single-cell RNA sequence (scRNA-seq) analysis. Single-cell suspensions of whole lungs were prepared from mice injected with DMSO ($n = 3$) or pitavastatin ($n = 3$) (Appendix Fig. S6A). After quality control, a total of 41,259 cells (DMSO group: 18,415 cells and pitavastatin

group: 22,844 cells) were retained for downstream analysis (see the Methods section for details). UMAP analysis revealed a diverse array of lung cell populations, identifying numerous distinct cell types, such as alveolar epithelial cells, macrophages, T cells, and endothelial cells (Appendix Fig. S6B). Comparative UMAP plots showed the lung cell populations in DMSO- and pitavastatin-administered mice and suggested that the populations of each cell type were largely unaffected by pitavastatin treatment (Appendix Fig. S6C). We then focused on cells expressing genes involved in cholesterol metabolism (GO: 0008203) and type I IFN production (GO: 0032606). Accordingly, we found that the cluster identified as alveolar macrophages expressed genes associated with cholesterol metabolism and type I IFN production (Appendix Fig. S6D), prompting a more detailed investigation of the alveolar macrophage cluster.

Single-cell transcriptomics was employed to investigate the heterogeneity of alveolar macrophages within the lung tissue and to elucidate the response to pitavastatin treatment (Fig. 6A). Subclustering of alveolar macrophages revealed the presence of 11 distinct subpopulations, each characterized by a unique transcriptional profile (Fig. 6A–C). This stratification allowed for a detailed examination of gene expression variations within these populations. The violin plots in Fig. 6D provide a graphical representation of the expression levels of selected genes implicated in the innate immune response across the different alveolar macrophage subclusters. Notably, RIG-I (DDX58) expression was prominent in clusters 6, 8, and 10, while cGAS was predominantly expressed in cluster 5 (Fig. 6D). Interestingly, cluster 8 expressed CD11b (Itgam), and cluster 5, characterized by Mki67, indicating cell division, was predominantly associated with the G2/M phase of the cell cycle, as shown in the UMAP plots (Fig. 6D,E). This suggested that cGAS-expressing alveolar macrophages were proliferative.

Trajectory analysis suggested a transition from cluster 0 to either cluster 7 or cluster 5, with cluster 0 potentially representing a progenitor or earlier differentiation state (Fig. 6F). Furthermore, cluster 7 appeared to represent an intermediate state, subsequently resulting in further differentiation into clusters 1 and 2. Notably, comparative UMAP and violin plot analyses showed that pitavastatin treatment increased the expression of RIG-I (DDX58) in clusters 3, 6, 8, and 10 even in the absence of poly I:C injection (Fig. 6G,H). Moreover, MDA5 (IFIH1) expression was upregulated by pitavastatin treatment in clusters 8 and 10 (Fig. 6G,H), and cGAS expression was enhanced in cluster 5 (Fig. 6G,H). These single-cell analyses indicated that specific subsets of alveolar macrophages induced RIG-I expression in response to pitavastatin administration in vivo, and the cluster increasing the RIG-I expression was distinct from the cGAS expressing cluster.

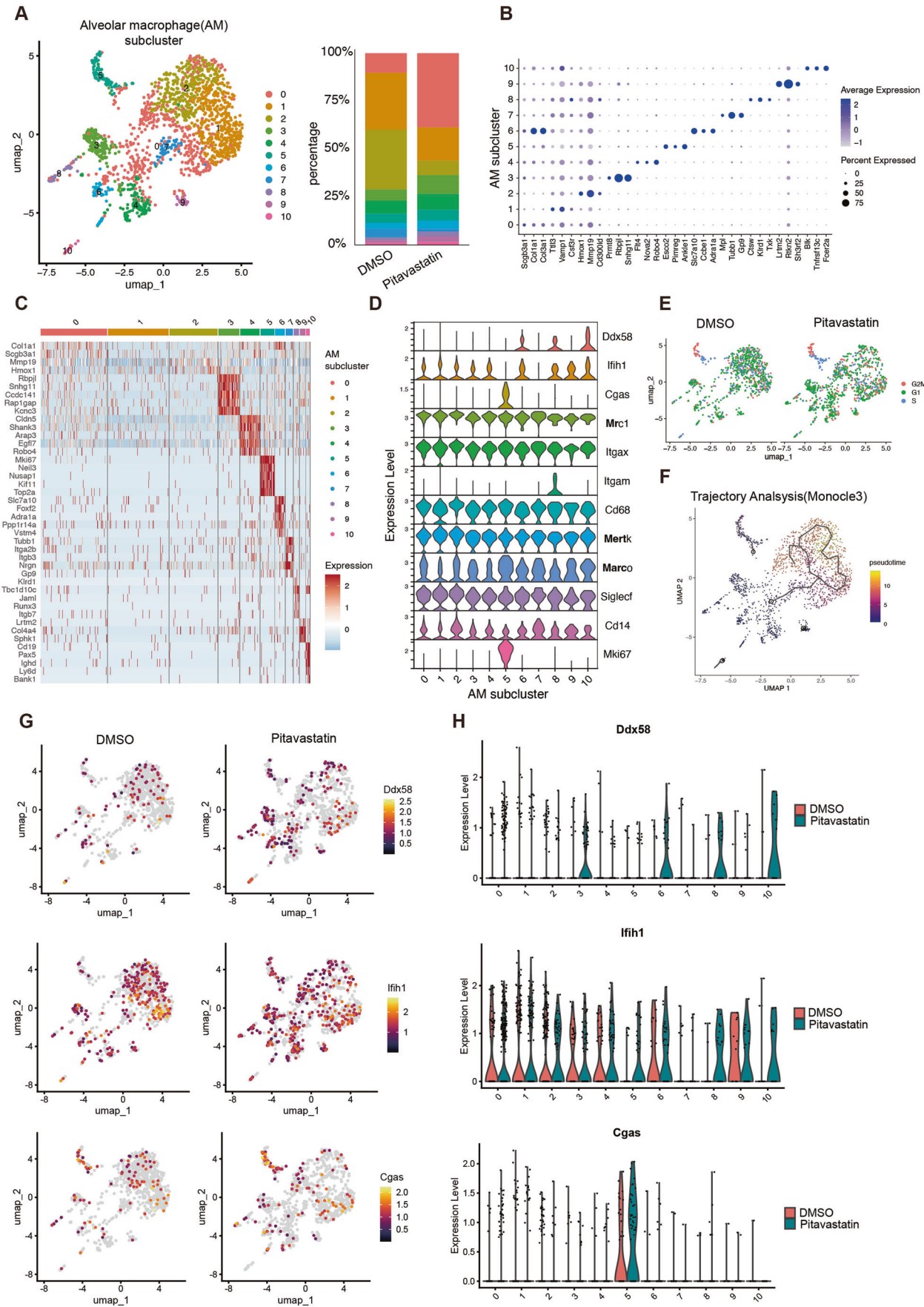

**Figure 6.  Detailed single-cell RNA Seq analysis of alveolar macrophage subpopulations in the lungs of pitavastatin-treated mice.**

(A) UMAP plots of alveolar macrophage subclusters identified within the lung cell populations. Each dot represents individual alveolar macrophages, with distinct colors indicating the different subclusters. The bar chart on the right shows the proportion of each subcluster in pitavastatin or DMSO-treated mice. (B) Dot blot illustrating the expression levels of selected genes in various alveolar macrophage subclusters. The size of each dot corresponds to the percentages of cells expressing the gene within each subcluster, while the color intensity indicates the average expression levels. (C) Heatmap of gene expression across alveolar macrophage subclusters, indicating the scaled expression level of a subset of genes. (D) Violin plots depicting the expression distribution of selected genes among alveolar macrophage subclusters including both DMSO and pitavastatin-treated groups. The expression level (log-normalized) is shown on the y-axis. The number of cells in each cluster is shown in Appendix Table S1. (E) Comparative UMAP plots showing cell cycle annotation of alveolar macrophages from DMSO- and pitavastatin-treated mice. (F) Trajectory analysis (Monocle 3) displaying the pseudo-time of alveolar macrophages, suggesting a developmental or activation trajectory. (G) UMAP plots showing the spatial distribution of gene expression on UMAP plots for Ddx58 (RIG-I), Ifih1 (MDA5), and Cgas (cGAS) in DMSO- and pitavastatin-treated mice. Color intensity indicates the expression levels. (H) Violin plots depicting the expression of selected genes across alveolar macrophage subclusters in DMSO (orange) and pitavastatin-treated (green) mice. The number of cells in each cluster is shown in Appendix Table S1.

To confirm the results obtained from the single-cell analyses, we first investigated whether the total fraction of alveolar macrophages express RIG-I and noncanonical type I IFNs in response to pitavastatin administration in the absence of poly I:C stimulation. Alveolar macrophages were collected from pitavastatin-injected mice, and the expression of RIG-I and noncanonical type I IFNs was measured (Fig. 7A). As expected, the mRNA expression levels of RIG-I and noncanonical type I IFNs, including IFN-α5, -ε, and -ζ, were higher in alveolar macrophages isolated from pitavastatin-administered mice than those from control mice, whereas IFN-λ and IFN-γ mRNA expression levels were not affected by the treatment (Fig. 7B). Subsequently, we sought to confirm the existence of subsets of alveolar macrophages corresponding to clusters 3, 6, 8, and 10 that can induce RIG-I expression in response to pitavastatin. Single-cell analyses determined the genes exclusively expressed in clusters 3, 6, 8, and 10 (Fig. 7C). We performed flow cytometry to detect the alveolar macrophages expressing the cluster-specific markers. Interestingly, approximately 10% of alveolar macrophages expressed CD140a encoded by Pdgfra, which was exclusively expressed in cluster 6 (Fig. 7D; Appendix Fig. S7A). Moreover, isolated CD140a⁺ alveolar macrophages markedly increased the expression of IFN-ζ and RIG-I in response to pitavastatin treatment in vitro (Fig. 7E,F; Appendix Fig. S7B). These data provide evidence for the existence of a subset of alveolar macrophages whose RIG-I expression is enhanced by cholesterol restriction.

## Discussion

The link between cholesterol metabolism and the risk of disease severity and death due to viral infections has gained increasing attention, with mounting evidence supporting this correlation (Almond et al, 2023; Gupta et al, 2021; Stefan et al, 2021). Previous studies have illustrated the activation of the cGAS-STING pathway by the defects in the mevalonate pathway regulated by SREBP2 as well as the enhancement of RIG-I signaling by cholesterol limitation via MAVS protein geranylgeranylation (Akula et al, 2019; Yang et al, 2019; York et al, 2015). Extending upon these findings, our study elucidated another mechanism by which cholesterol limitation triggers a cell-type-specific induction of noncanonical type I IFNs, such as IFN-ω and IFN-α16. This process increased RIG-I expression, which, in turn, amplified RIG-I-dependent type I IFN expression (Appendix Fig. S8). Our single-cell RNA transcriptome data corroborated that suppressing

cholesterol synthesis increased the expression of RIG-I in specific subsets of alveolar macrophages in vivo. These findings indicate that restricting cholesterol synthesis primes the RIG-I pathway via noncanonical type I IFNs to enhance antiviral response in specific cell types. Considering the distinct expression of cGAS and RIG-I in the alveolar macrophage clusters, we anticipate that both cGAS and RIG-I are involved in the immunometabolism related to cholesterol synthesis.

The protein geranylgeranylation ligand GGPP, a metabolite of the mevalonate pathway that is inhibited by statin treatment, is associated with innate immunity (Akula et al, 2019; Palsuledesai and Distefano, 2015). Since the geranylgeranylation of MAVS is inhibited by statins (Yang et al, 2019), this modification could be responsible for the statin-mediated enhancement of type I IFN expression. Indeed, a previous study reported that simvastatin enhanced IFN-β expression following Flu infection in macrophages (Yang et al, 2019). However, our study revealed that statin treatment can elevate the expression of noncanonical type I IFNs even in the absence of viral infection, a condition in which MAVS is not activated. Moreover, the expression of these noncanonical type I IFNs depended on the type I IFN receptor and SREBP1, and the absence of IFNAR2 or SREBP1 abolished the effect of statins in RIG-I-mediated IFN-β expression, as shown in Figs. 3K and 4E. These observations indicate that the geranylgeranylation of the MAVS protein and the noncanonical type I IFN pathway cooperatively enhance RIG-I-dependent antiviral response in response to the restriction of cholesterol.

Viral infection induces the expression of canonical type I IFNs, such as IFN-β, via IRF3 and IRF7 transcription factors. Over ten type I IFN genes are encoded in the human and mouse genomes and all type I IFNs activate the type I IFN receptor, comprising IFNAR1 and IFNAR2 (Platanias, 2005). However, the current study revealed a noncanonical type I IFN expression pathway that depends on the SREBP1 transcription factor. Both SREBP1 and SREBP2 bind to the SRE motif; however, there is a difference in their targeted genes due to the difference in their cofactors (Chandrasekaran and Weiskirchen, 2024; Shimano and Sato, 2017). SREBP2 preferentially activates cholesterol synthesis, including the mevalonate pathway, whereas SREBP1 favors the fatty acid biosynthetic pathway (Horton et al, 2002). Our data indicate that SREBP1, but not SREBP2, is responsible for the noncanonical type I IFN expression. This noncanonical type I IFN expression pathway does not require viral infection and can prime antiviral innate immune response, especially the RIG-I-dependent pathway, even in the absence of viral infection. In human cells, IFN-ω was regulated

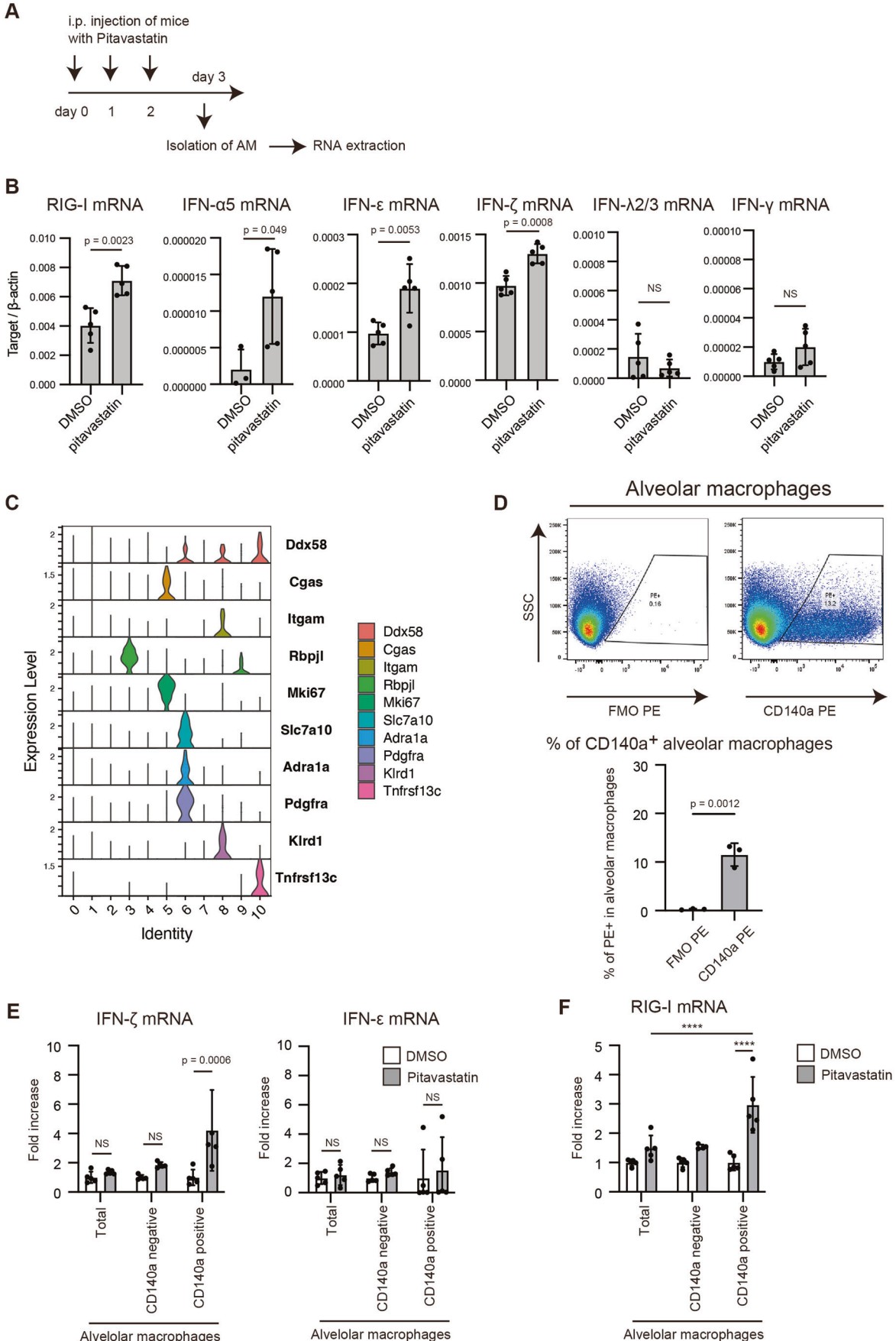

**Figure 7. Priming of RIG-I pathway in CD140a⁺ alveolar macrophages.**

(A, B) Mice were intraperitoneally injected with pitavastatin or solvent (DMSO) every three days, and alveolar macrophages were isolated (A). The expression of each gene in the isolated alveolar macrophages was determined by RT-qPCR and normalized to that of β-actin (B) ($n = 5$, biological replicates). (C) A violin plot depicting cluster-specific gene expression in alveolar macrophages. (D) Alveolar macrophages (CD45⁺, Siglect F⁺, and CD11c⁺) were isolated from the mouse bronchoalveolar lavage fluid (Appendix Fig. S6), and cells were analyzed by flow cytometry to observe cells expressing CD140a encoded by Pdgfra ($n = 3$, biological replicates). (E) CD140a-positive alveolar macrophages were isolated from the lung using a MACS column, and cells were treated with DMSO or pitavastatin (20 μM) for 24 h. The expression of noncanonical type I IFNs was compared among CD140a positive, CD140a negative (flow-through), and total (CD140a positive + CD140a negative) alveolar macrophage fractions ($n = 5$, biological replicates). (F) CD140a-positive alveolar macrophages isolated from the lung were treated with DMSO or pitavastatin (20 μM) for 30 h. RIG-I expression was measured by RT-qPCR and normalized to β-actin. Fold increase was calculated relative to untreated samples ($n = 4$, biological replicates). Data Information: In (B, D–F), data are represented as mean ± SD (****$p < 0.0001$, t-test (B, D), two-way ANOVA (E, F)). NS: not significant. Source data are available online for this figure.

by the SREBP1. Unlike humans, the mouse genome does not encode IFN-ω. However, several noncanonical type I IFNs, such as IFN-ε, IFN-ζ, and IFN-α5, were upregulated in mouse alveolar macrophages in response to pitavastatin administration without viral infection. It remains unclear whether the SREBP1 is responsible for the mouse noncanonical type I IFN expression pathway; thus, further studies are required to elucidate the noncanonical pathways in different species. It is also unclear whether canonical type I IFNs are induced in a specific cell type in vivo. This uncertainty arises because overexpression of SREBP1 could induce IFN-β mRNA levels, implying the potential role of SREBP1. Therefore, we do not exclude the possibility that pitavastatin treatment may induce not only noncanonical type I IFNs but also canonical type I IFNs via SREBP1 in specific cell types in vivo.

MDA5 is another member of the RLR family. Our in vitro and in vivo data demonstrate that MDA5 is also upregulated by statin treatment. Notably, type I IFN expression induced by EMCV, which is recognized by MDA5, was enhanced by statin treatment. This suggests that not only RIG-I but also MDA5 is regulated by cholesterol restriction. In addition, we observed that pitavastatin treatment enhanced IRF7 expression. Since IRF7 is a key transcription factor downstream of TLR7, 8, and 9, responsible for inducing type I IFNs in plasmacytoid dendritic cells, cholesterol restriction may also affect these TLR-dependent pathways. Our in vitro data also showed that TLR3 was involved in the pitavastatin-mediated enhancement of type I IFN expression. Therefore, we have to carefully consider the involvements of TLR3, MDA5, and other pathways in the type I IFN response in vivo. Further studies are needed to comprehensively understand the effects of cholesterol restriction on innate immune responses in vivo.

Our single-cell transcriptome analyses showed that only specific clusters of alveolar macrophages induce RIG-I expression in response to pitavastatin treatment. This observation also suggests the existence of unresponsive macrophages in vivo. Indeed, we could not observe the effect of statins in the spleen (Fig. 5C). Mouse alveolar macrophages can be identified using specific markers, such as CD11c, SiglecF, CD64, F4/80, and MerTK, and are conventionally classified into tissue-resident alveolar macrophages and monocyte-derived recruited alveolar macrophages (Aegerter et al, 2022). Recruited alveolar macrophages express relatively lower levels of Siglec F than tissue-resident alveolar macrophages (Aegerter et al, 2020; Aegerter et al, 2022). Although our single-cell transcriptome analysis indicated that cluster 9 expressed lower levels of Siglec-F than the other clusters, it is unclear whether

cluster 9 represented the recruited macrophages. Moreover, the single-cell transcriptome analyses revealed several clusters expressing RIG-I in response to pitavastatin. Flow cytometry revealed the presence of CD140a⁺ alveolar macrophages, and this macrophage fraction is expected to belong to cluster 6 and is distinct from cluster 5, which expresses cGAS and Mki67, which encodes the proliferative marker Ki67 (Bruno and Darzynkiewicz, 1992; Li et al, 2021). The differential expression of RIG-I and cGAS implies that alveolar macrophages are heterogeneous cell populations comprising several functionally distinct populations. Thus, we expect that both the cGAS and RIG-I pathways synergistically augment antiviral innate immune responses, reflecting cholesterol levels in vivo.

One of the limitations of this study is that it is still unclear why pitavastatin treatment upregulated many ISGs but some were not induced, as shown in Fig. EV2A. This might be due to the difference in the expression kinetics following stimulation of type I IFN receptor. Further studies are required to reveal the underlying mechanism. Another limitation is that the mechanism underlying cell-type specificity remains unclear. One possibility is that epigenetic regulation of noncanonical type I IFNs differs among cell types. Another possibility is that tolerance to statins varies between different cell types. It is also possible that immortalization of cells affects the tolerance to statins. Thus, cell-type specificity should be carefully understood when using immortalized cell lines. Although we utilized mouse primary cells in this study, human primary cells are not tested. These limitations should be addressed in future studies. In addition, the effects of statins on type II and type III IFNs are not fully elucidated. Since type II IFNs are mainly produced by NK and T cells, it is unlikely that type II IFNs played a major role under the experimental condition using only macrophages or epithelial cells. However, considering that pitavastatin treatment moderately enhanced type II IFN-induced IP10 mRNA expression (Appendix Fig. S5), it remains possible that type II IFN-mediated responses are affected by statins in the context of secondary or late-phase of viral infections, where T cells produce type II IFNs. Although our knockdown data showed that type III IFN receptor was dispensable for RIG-I expression following pitavastatin treatment, it is still possible that type III IFNs are involved in this phenomenon in specific cell types in vivo. Since the effect is not restricted to RIG-I, statins are expected to affect various innate immune responses. It is also possible that statins or a decrease in cellular cholesterol levels may affect membrane composition thereby affecting the infectivity and transfectability of cells. However, our data weaken this possibility. First, SeV RNA levels in infected cells were lower in statin-treated cells compared to

the control, suggesting that statins did not increase infectivity. Second, statin treatment did not affect the transfection efficiency. For instance, the knockdown efficiency of siRNA was comparable between statin-treated and untreated cells. Third, we found that the SREBP1 transcription factor binds to the promoter region of noncanonical type I IFNs, although there is a limitation in our ChIP assay, as it remains possible that other promoter regions are also targeted by the transcription factor. Careful consideration of the effect of statins on membrane composition is necessary to fully understand the mechanism underlying statin-mediated enhancement of innate immune responses.

In addition to the noncanonical type I IFN-dependent mechanism, other pathways might also link cholesterol metabolism to antiviral innate immune responses. For example, actin cytoskeleton disturbance caused by viral infection has been reported to trigger relocalization of the the PPP1R12C protein, thereby enhancing RLR response (Acharya et al, 2022). Considering that cellular cholesterol levels influence the actin cytoskeleton (Sun et al, 2007), it is possible that cholesterol metabolism affects the RLR response via cytoskeletal dynamics. Future investigations will be necessary to fully elucidate the connections between cholesterol metabolism and antiviral innate immunity.

Globally, we have repeatedly been confronted with pandemics caused by emerging viruses (Graham et al, 2013). Vaccines and medicines have been developed for various viral infectious diseases; however, these take time to develop to cope with new emerging viruses. Our findings indicate that cholesterol restriction or a limited amount of noncanonical type I IFNs primes RIG-I-dependent antiviral innate immune responses. Targeting this mechanism may be useful for establishing new treatment methods to cope with future viral pandemics.

# Methods

### Reagents and tools table

| Reagent/ Resource | Reference or Source | Identifier or Catalog Number |
|---|---|---|
| **Experimental models** | | |
| HEK293 (*H. sapiens*) | Oshiumi et al, 2010 | N/A |
| HeLa (*H. sapiens*) | Oshiumi et al, 2010 | N/A |
| HEK293FT (*H. sapiens*) | Oshiumi et al, 2010 | N/A |
| A549 (*H. sapiens*) | ATCC | CAT# CCL-185 |
| THP-1 (*H. sapiens*) | ATCC | CAT# TIB-202 |
| HepG2 (*H. sapiens*) | Kouwaki T et al, 2016 | N/A |
| RAW264.7 (*M. musculus*) | ATCC | CAT# TIB-71 |
| HeLa IFNAR2 KO | This paper | N/A |
| HeLa SREBP1 KO | This paper | N/A |

| Reagent/ Resource | Reference or Source | Identifier or Catalog Number |
|---|---|---|
| HeLa SREBP2 KO | This paper | N/A |
| C57BL/6JJcl (*M. musculus*) | CLEA Japan | N/A |
| **Recombinant DNA** | | |
| pEF-BOS-AcGFP | This paper | N/A |
| pEF-BOS-mCherry | This paper | N/A |
| pEF-BOS-FALG-RIG-I | Oshiumi et al, 2010 | N/A |
| pEF-BOS-FLAG-RIG-I CARDs | Oshiumi et al, c | N/A |
| pEF-BOS-SREBP1a-N | This paper | N/A |
| pEF-BOS-SREBP1c-N | This paper | N/A |
| pEF-BOS-SREBP2-N | This paper | N/A |
| pEF-BOS-SREBP1a | This paper | N/A |
| pEF-BOS-SREBP1c | This paper | N/A |
| pEF-BOS-Riplet-HA | Oshiumi et al, 2009 | N/A |
| pX459, pSpCas9(BB)-2A-Puro | addgene | Addgene #48139 |
| **Antibodies** | | |
| Anti-β-actin antibody | Sigma-Aldrich | Cat# AMAb91241; RRID: AB_2665860 |
| Anti-Flag antibody | Sigma-Aldrich | Cat# F3165; RRID: AB_259529 |
| Anti-RIG-I (D14G6) antibody | Cell Signaling Technology | Cat# 3743; RRID: AB_2269233 |
| Anti-MDA-5 (D74E4) antibody | Cell Signaling Technology | Cat#5321; RRID: AB_10694490 |
| SREBP1 monoclonal (2A4) | BD Biosciences | Cat# 557036; RRID: AB_396559 |
| SREBP-2 antibody(1C6) | SANTA CRUZ BIOTECHNOLOGY | Cat# sc-13552; RRID: AB_2194250 |
| IFN omega antibody | Invitrogen | Cat# PA5-99819; RRID: AB_2818752 |
| Anti-Mouse IgG, HRP-Linked whole Ab Sheep | GE Healthcare | Cat# NA931; RRID: AB_772210 |
| Anti-Rabbit IgG, HRP-Linked whole Ab Donkey | GE Healthcare | Cat# NA934; RRID: AB_772206 |
| PE anti-mouse CD140a Antibody | BioLegends | Cat# 135906; RRID: AB_1953269 |

| Reagent/Resource | Reference or Source | Identifier or Catalog Number |
|---|---|---|
| APC anti-mouse CD11c Antibody | BioLegends | Cat# 117310; RRID: AB_313779 |
| FITC anti-mouse CD170 (Siglec-F) Antibody | BioLegends | Cat# 155504; RRID: AB_2750233 |
| APC/Cyanine7 anti-mouse CD45 Antibody | BioLegends | Cat# 103116; RRID: AB_2888795 |
| Brilliant Violet 421™ anti-mouse F4/80 Antibody | BioLegends | Cat# 123131; RRID: AB_10901171 |
| **Oligonucleotides and other sequence-based reagents** | | |
| Silencer select pre-designed siRNA IL28Ra (IFNLR1) | Invitrogen | Cat# 4392420 siID: s46473 |
| Silencer select pre-designed siRNA STING | Invitrogen | Cat# 4392420 siID: s50645 |
| Silencer select pre-designed siRNA MAVS | Invitrogen | Cat# 4392420 siID: s33180 |
| Silencer select pre-designed siRNA HMGCR | Invitrogen | Cat# 4392420 siID: s143 |
| Silencer select pre-designed siRNA TLR3 | Invitrogen | Cat# 4390824 siID: s236 |
| **CRISPR-Cas9 guide sequence** | | |
| IFNAR2 knockout forward | This study | CACCGCCGTCCTAGAAGGATTCAGC |
| IFNAR2 knockout reverse | This study | AAACGCTGAATCCTTCTAGGACGGC |
| SREBP1 knockout forward | This study | CACCGCGCGCACACTCGAGAAGCT |
| SREBP1 knockout reverse | This study | AAACAGCTTCTCGAGTGTGCGCGC |
| SREBP2 knockout forward | This study | CACCGTGAATGACCGTTGCACTGA |
| SREBP2 knockout reverse | This study | AAACTCAGTGCAACGGTCATTCAC |
| **Primer sequences for cloning** | | |
| IRF7 promoter - luc forward | This study | AGGGCATCGGTCGACTAAATCCCAGCACTTTGGG |
| IRF7 promoter - luc reverse | This study | CGGAATGCCAAGCTTCGGGAAGGGCGCGCGCCG |
| IFNW1 promoter - luc forward | This study | AGGGCATCGGTCGACGCTTGTAAAGAATCAGGATTC |
| IFNW1 promoter - luc reverse | This study | CATTTGATTAAGCATTCATTTGAAGCTTGGCATTCCG |
| RIG-I promoter - luc forward | This study | AGGGCATCGGTCGACACTGGAGCTCTCCTACTG |
| RIG-I promoter - luc reverse | This study | CGGAATGCCAAGCTTCCCGCAGGCTGTGCCTCA |
| pEF-BOS-SREBP1a-N forward | This study | ATGATGATGATAAAGGTACCATGGACGAGCCACCCTTC |
| pEF-BOS-SREBP1a-N reverse | This study | GGACCCTCACTCTAGGTACCGCGGGAGCGGTCCAGCAT |
| pEF-BOS-SREBP1c-N forward | This study | ATGATGATGATAAAGGTACCATGGATTGCACTTTCGAAGA |
| pEF-BOS-SREBP1c-N reverse | This study | GGACCCTCACTCTAGGTACCGCGGGAGCGGTCCAGCAT |
| pEF-BOS-SREBP2-N forward | This study | ATGATGATGATAAAGGTACCATGGACGACAGCGGCGAG |
| pEF-BOS-SREBP2-N reverse | This study | GGACCCTCACTCTAGGTACCCCGTGAGCGGTCTACCAT |
| **These primer sequences are including vector site sequence to perform In-Fusion** | | |
| **Primer sequences for qPCR** | | |
| **Human** | | |
| GAPDH forward | This study | CAATATGATTCCACCCATGG |
| GAPDH reverse | This study | AATGAGCCCCAGCCTTCTCC |
| IRF1 forward | This study | ACATCCCAGTGGAAGTTGTG |
| IRF1 reverse | This study | TTCCCTTCCTCATCCTCATC |
| IRF2 forward | This study | GCGGTCCTGACTTCAAC |
| IRF2 reverse | This study | GGTCACCTCTACAACTTGG |
| IRF5 forward | This study | GGGCTCATCCTCCAGCTAC |
| IRF5 reverse | This study | ACAAGGCCCGCTCCAGAA |
| IRF7 forward | This study | AGAGCCGTACCTGTCAC |
| IRF7 reverse | This study | CATGATGGTCACGTCCAG |
| IRF9 forward | This study | CCACCAGGAATCGTCTC |
| IRF9 reverse | This study | TTCCTCCTTCCTCTCAG |
| RIG-I forward | This study | TGGCATATTGACTGGACGTG |

| Reagent/ Resource | Reference or Source | Identifier or Catalog Number |
|---|---|---|
| RIG-I reverse | This study | TGCCTTCATCAGCAACTGAG |
| MDA5 forward | This study | AGAGTGGCTGTTTACATTGCC |
| MDA5 reverse | This study | GCTGTTCAACTAGCAGTACCTT |
| MAVS forward | This study | AATGCCGTTTGCTGAAGACAA |
| MAVS reverse | This study | CAGTCGATCCTGGTCTCTTGCT |
| TLR3 forward | This study | AGTGCCGTCTATTTGCCACA |
| TLR3 reverse | This study | GCATCCCAAAGGGCAAAAGG |
| TICAM-1 forward | This study | AGCGCCTTCGACATTCTAGGT |
| TICAM-1 reverse | This study | AGGAGAACCATGGCATGCA |
| TICAM-2 forward | This study | GGGAATTCATAATGGGTATCGGGAAGTC |
| TICAM-2 reverse | This study | GGCTGCAGGTTATATGTTTCATCTCAGGC |
| cGAS forward | This study | GGGAGCCCTGCTGTAACACTTCTTAT |
| cGAS reverse | This study | CCTTTGCATGCTTGGGTACAAGGT |
| STING forward | This study | AGCATTACAACAACCTGCTACG |
| STING reverse | This study | GTTGGGGTCAGCCATACTCAG |
| IFNA1 forward | This study | GCCATCTCTGTCCTCCATGA |
| IFNA1 reverse | This study | TGGTAGAGTTCGGTGCAGAAT |
| IFNA2 forward | This study | GTGGGCTGTGATCTGCCTCAAAC |
| IFNA2 reverse | This study | AGGGATGGTTTCAGCCTTTTG |
| IFNA4 forward | This study | ACATGATTTCGGATTCCCCGAGGAGGAG |
| IFNA4 reverse | This study | TTTTTCTAGGAGGCTCTGTTCCCAAGCAGC |
| IFNA5 forward | This study | CAGACCCACAGCCTGAGTAAC |
| IFNA5 reverse | This study | GAATTTGTCTAGAAGTGTCTCATCCCAAGT |
| IFNA16 forward | This study | TTTTCAGGAGTGTAAAGAAGCA |
| IFNA16 reverse | This study | CATAAAACATGATTTAAACCTTAAAAATAG |
| IFNA17 forward | This study | CTAGGCTGTGATCTGCCTCAGAC |
| IFNA17 reverse | This study | CCATCAAACTCCTCCTGGGGAAG |

| Reagent/ Resource | Reference or Source | Identifier or Catalog Number |
|---|---|---|
| IFN-β forward | This study | TGGGAGGATTCTGCATTACC |
| IFN-β reverse | This study | CAGCATCTGCTGGTTGAA |
| IFNE1 forward | This study | AGGACACACTCTGGCCATTC |
| IFNE1 reverse | This study | TTGCTTCATGTCGTTCAAGG |
| IFNK forward | This study | ATTCATTGCTGGCACCCTAT |
| IFNK reverse | This study | GAAACTCTTGGGGCAACTCA |
| IFNW1 forward | This study | TGTCCTCCATGAGATGC |
| IFNW1 reverse | This study | CAGGTCTCCAGGTGTTG |
| IFNG forward | This study | ACTGACTTGAATGTCCAACGCA |
| IFNG reverse | This study | ATCTGACTCCTTTTTCGCTTCC |
| IFNL1 forward | This study | CGCCTTGGAAGAGTCACTCA |
| IFNL1 reverse | This study | GAAGCCTCAGGTCCCAATTC |
| IFNL3 forward | This study | GAAGGTTCTGGAGGCCACC |
| IFNL3 reverse | This study | GGCTGGTCCAAGACATCC |
| IFNAR1 forward | This study | AATGTGACGCTGCTCTCCC |
| IFNAR1 reverse | This study | TTGGTTCCCGCACACTCTTC |
| IP10 forward | This study | TCCACGTGTTGAGATCATTGC |
| IP10 reverse | This study | GGCCTTCGATTCTGGATTCAG |
| HMGCR forward | This study | CTGTGGGCGGTTGTTAAG |
| HMGCR reverse | This study | ACGAGCCTTCGACCAATAAG |
| Mouse | | |
| β-actin forward | This study | TTTGCAGCTCCTTCGTTGC |
| β-actin reverse | This study | TCGTCATCCATGGCGAACT |
| Rig-i forward | This study | TTCTCCCTTTAGTGTCTCGG |
| Rig-i reverse | This study | GTGAAGCCATCGAAAGTTGG |
| Mda5 forward | This study | GCTGCCCAGAAGACAACACAG |
| Mda5 reverse | This study | CGACAGCAGGCAGAAGACACT |
| cgas forward | This study | ACCGGACAAGCTAAAGAAGGTGCT |
| cgas reverse | This study | GCAGCAGGCGTTCCACAACTTTAT |
| Sting forward | This study | TGAAAGGCTCTTCATTGTCTCTT |
| Sting reverse | This study | TGGCATCTTCTGCTTCCTAGA |
| Ifnb forward | This study | TCCAAGAAAGGACGAACATTC |
| Ifnb reverse | This study | TGAGGACATCTCCCACGTCAA |
| Ifna1 forward | This study | GAGAAGAAACACAGCCCCTG |
| Ifna1 reverse | This study | TCAGTCTTCCCAGCACATTG |
| Ifna2 forward | This study | TGAAGGTCCTGGCACAGATG |
| Ifna2 reverse | This study | TGGCTTGAGCCTTCTGGATC |
| Ifna5 forward | This study | AGGACTCATCTGCTGCATGGAATG |
| Ifna5 reverse | This study | CACACAGGCTTTGAGGTCATTGAG |
| Ifna16 forward | This study | GCCATCCCTGTCCTAAGTGA |
| Ifna16 reverse | This study | ATCCCAAGCAGCAGATGAAT |

| Reagent/ Resource | Reference or Source | Identifier or Catalog Number |
|---|---|---|
| Ifne forward | This study | CCTTCAGCAGATCTTCACGC |
| Ifne reverse | This study | TGACTCCACGTATTCCAGCT |
| Ifnk forward | This study | AAACGCCGTCTCCTATCGTA |
| Ifnk reverse | This study | CCCGATCTGATACGTTCCCA |
| Ifnz forward | This study | GAAAGGAAACCGAACCAAAGGAG |
| Ifnz reverse | This study | CATCGCGCCTTGAGAATTGAACC |
| Ifng forward | This study | ACAATGAACGCTACACACTGCAT |
| Ifng reverse | This study | TGGCAGTAACAGCCAGAAACA |
| Others | | |
| PR8 segment6 forward | This study | AATAGCACCTGGGTAAAGGACAC |
| PR8 segment6 reverse | This study | ATATAGCCCACCCACGGATG |
| ChIP-qPCR | | |
| IFNW1 promoter (−59 ~ −221) forward | This study | CTTCTAGAACAGAGAAAATG |
| IFNW1 promoter (−59 ~ −221) reverse | This study | CTATATGCATGTAAATAATCAG |
| IFNW1 promoter (0 ~ −100) forward | This study | ATTGTTAGTGTAGTGTGTG |
| IFNW1 promoter (0 ~ −100) reverse | This study | CAAATGAATGCTTAATCAAATG |
| IFNW1 promoter (−205 ~ −290) forward | This study | ATAGAAACCTATAGGGCCTG |
| IFNW1 promoter (−205 ~ −290) reverse | This study | AAAAATCATTTTCTCTGTTC |
| IFNW1 promoter (−377 ~ −476) forward | This study | GAGGCACAGAGATTGCTTGAGCCC |
| IFNW1 promoter (−377 ~ −476) reverse | This study | GAGATGGGGTCTTAATCCGTCACC |
| IFNW1 promoter (−392 ~ −511) forward | This study | GGTGCACACCTGTTGTCCCAGG |
| IFNW1 promoter (−392 ~ −511) reverse | This study | TCCGTCACCCAGGTTGGAGTGTGG |
| IFNW1 promoter (−752 ~ −864) forward | This study | CAGATAGGGGAGCACTATTG |

| Reagent/ Resource | Reference or Source | Identifier or Catalog Number |
|---|---|---|
| IFNW1 promoter (−752 ~ −864) reverse | This study | CTTCCTTCTAGAAAACATTC |
| IFNW1 promoter (−872 ~ −958) forward | This study | GTGAGGAGACGCTGGGAAAG |
| IFNW1 promoter (−872 ~ −958) reverse | This study | CTCCCCTATCTGTCAATATC |
| IFNA16 promoter (−165 ~ −265) forward | This study | TCCAATTAGTAAAAAATACC |
| IFNA16 promoter (−165 ~ −265) reverse | This study | CTGTTTTTGCTTTCTTTATG |
| hLDLR promoter forward | This study | AGCTTCACGGGTTAAAAAGC |
| hLDLR promoter reverse | This study | TTTCAATGTGAGGTTTCTAG |
| hHMGCR promoter forward | This study | CTGTGGGCGGTTGTTAAG |
| hHMGCR promoter reverse | This study | ACGAGCCTTCGACCAATAAG |
| hRIG-I promoter (−199 ~ −346) forward | This study | CTCATCGAATTGCACACTCAG |
| hRIG-I promoter (−199 ~ −346) reverse | This study | GTTTCCCTCCGCACCGAGGAAG |
| hIFNB1 promoter (−257 ~ −351) forward | This study | GAATCCACGGATACAGAACC |
| hIFNB1 promoter (−257 ~ −351) reverse | This study | CAAGACTTGGGAGAAAGC |
| hIFNB1 promoter (−469 ~ −581) forward | This study | CAAAATCAGAGGATGCTCAATTC |
| hIFNB1 promoter (−469 ~ −581) reverse | This study | CAGGTATTATAAGTAATCTAGGG |
| hIFNB1 promoter (−671 ~ −752) forward | This study | TTCAGAAGTTTCCAAGGCCC |
| hIFNB1 promoter (−671 ~ −752) reverse | This study | AGGGAATAGTAGCTGAAACTGC |

| Reagent/Resource | Reference or Source | Identifier or Catalog Number |
|---|---|---|
| hIFNB1 promoter (−740 ~ −846) forward | This study | TTCCACTTGTGGCAGTTTCT |
| hIFNB1 promoter (−740 ~ −846) reverse | This study | GATTAGTGTGAGAGGGCCTTG |
| **Chemicals, Enzymes and other reagents** | | |
| Puromycin | InvivoGen | Cat# ant-pr-1 |
| Trypsin-EDTA solution | Wako | Cat# 1689149 |
| FCS | GIBCO | Cat# 10270 |
| FCS (for HeLa and HEK293 cells cluture) | Sigma-Aldrich | Cat# F7524 |
| 200 mmol/L L-Glutamine Solution (×100) | Wako | Cat# 073-05391 |
| Penicillin/Streptomycin | Wako | Cat# 168-23198 |
| Lipofectamine 2000 | Thermo Fisher | Cat# 11668019 |
| Lipofectamine RNAi MAX | Thermo Fisher | Cat# 13778150 |
| Immobilon-P PVDF membrane | Millipore | Cat# IPVH000010 |
| Protease inhibitor cocktail tablets | Sigma-Aldrich | Cat# 11697498001 |
| Anti-PE MicroBeads | Miltenyi Biotec | Cat# 130-048-801 |
| short polyI:C (LMW polyI:C) | InvivoGen | Cat# tlrl-picw |
| Poly(I:C) (HMW) | InvivoGen | Cat# tlrl-pic |
| Poly(I)·Poly(C) | Cytiva | Cat# 27473201 |
| 2'3'-cGAMP | InvivoGen | Cat# tlrl-nacga23-02 |
| CL097 | InvivoGen | Cat# tlrl-c97] |
| ODN 2395 | InvivoGen | Cat# tlrl-2395 |
| Pitavastatin Calcium | Sigma-Aldrich | Cat# SML2473 |
| cerivastatin sodium salt hydrate | Sigma-Aldrich | Cat# SML0005 |
| Atorvastatin Calcium | Selleck | Cat# S2077 |
| Cholesterol | Wako | Cat# 034-03002 |
| Recombinant Human IFN-ω | PEPROTECH | Cat# 300-02J |
| Ruxilitinib | Selleck | Cat# S1378 |
| **Software** | | |
| GraphPad Prism 10 | https://www.graphpad.com/ | |

| Reagent/Resource | Reference or Source | Identifier or Catalog Number |
|---|---|---|
| GENETYX-MAC | https://www.genetyx.co.jp/products/genetyx-mac-22/index.html | |
| BZ-H1A analysis software | https://www.keyence.co.jp/products/microscope/fluorescence-microscope/bz-8100/models/bz-h1a/ | |
| **Other** | | |
| Influenza A virus | Oshiumi et al, 2010 | PR8 strain |
| Sendai virus (SeV) | Oshiumi et al, 2010 | HVJ strain |
| EMCV | ATCC | VR-1762 strain |
| Dual-Luciferase® Reporter Assay System | Promega | Cat# E1980 |
| Cholesterol/Cholesterol Ester-Glo™ Assay | Promega | Cat# J3191 |
| SimpleChIP® Enzymatic Chromatin IP Kit (Magnetic Beads) | Cell Signaling Technology | Cat# 9003 |
| Human IFN-Beta ELISA Kit | PBL Assay Science | Cat# 41410 |
| VeriKine Interferon β ELISA Kit, Mouse | PBL Assay Science | Cat# 42400-1 |

## Cells, viruses, and reagents

HeLa cells were cultured in Eagle's MEM (Nissui) with 10% heat-inactivated fetal calf serum (FCS) and 10% L-Glutamine (Wako). HEK293 and HepG2 cells were cultured in low glucose DMEM (Wako) with 10% heat-inactivated FCS and penicillin/streptomycin (Wako). RAW264.7 and A549 cells were cultured in high glucose DMEM (Wako) with 10% heat-inactivated FCS and penicillin/streptomycin. THP-1 cells were cultured in RPMI (Wako) with 5% heat-inactivated FCS and penicillin/streptomycin. These cells were grown in a humidified incubator maintained at 37 °C with a 5% $CO_2$ atmosphere. IFNAR2, SREBP1, and SREBP2 KO cells were generated using the CRISPR/Cas9 system. The guide sequence was cloned into the BbsI restriction site of the pX459 plasmid, which encodes Cas9 and puromycin-resistance genes. The plasmid carrying the guide sequence was then transfected into HeLa cells. After 36 h, the cells were treated with 1 μg/mL of puromycin for 3 days to isolate the cells with the plasmids. Then, cells were cultured in fresh medium without puromycin for 2 days and were seeded onto 96-well plates to obtain single clones. The guide sequence and purchased reagents are listed in Reagents and Tools Table. Pitavastatin and cerivastatin were dissolved in DMSO at a concentration of 10 mM. Cells were treated with DMSO concentrations of 0.1% or less. The treatment of cells with pitavastatin was performed at 2 μM for 24 h unless otherwise stated. SeV and Flu were amplified using chicken egg, and the number of plaque-forming units was determined by a plaque assay as described previously

(Kouwaki et al, 2023). EMCV was purchased from ATCC and was amplified with Vero cells (Kouwaki et al, 2023). Short poly I:C was purchased from InvivoGen (poly (I:C)(LMW), with an average ranging from 0.2 to 1 kb. The poly I:C used for addition to cell culture medium or injection to mice, which contains both short and long poly I:C, was purchased from Cytiva. cGAMP was purchased from InvivoGen. The amounts and concentrations used for stimulation are indicated in each figure legend. cGAMP and short poly I:C were transfected with Lipofectamine 2000 (Thermo Fisher Scientific).

## Screening of compounds

The Validated Compound Library containing ~1600 existing approved drugs was provided by the Drug Discovery Initiative (The University of Tokyo). In the first screening, HEK293 cells were transfected with pEF-BOS-AcGFP, p125-mCherry, pEF-BOS-FLAG-tagged-RIG-I, and pEF-BOS-HA-tagged-Riplet plasmids using Lipofectamine 2000, treated with each compound simultaneously, and cultured in 96-well plates. After 48 h, the fluorescence of AcGFP and mCherry was observed using a BZ-X700 microscope (Keyence), and fluorescence intensities were measured with BZ-H1A software (Keyence). Compounds exhibiting a 1.5-fold increase or decrease in the intensity ratio of mCherry/AcGFP values normalized to the control (DMSO) were chosen for the second screening. The promoter activity of the IFN-β gene was calculated by dividing the fluorescence levels of mCherry by those of AcGFP. In the second screening, HEK293 cells were transfected with p125-Luciferase (p125 luc), pRL-TK-Renilla, pEF-BOS-FLAG-tagged-RIG-I, and pEF-BOS-HA-tagged-Riplet expression plasmids using Lipofectamine 2000 and treated with each compound, which were isolated by the first screening, and subsequently cultured in white 96-well plates (Corning). Cell lysates were prepared 24 h after transfection, and the activities of luciferase and *Renilla* luciferase were determined using the Dual-Luciferase Reporter Assay system (Promega). The luminescence was measured using a FilterMAX F5 microplate reader (MOLECULAR DEVICES), and the promoter activity of IFN-β was determined by normalizing luciferase activities to *Renilla* luciferase activities. Twenty-two compounds exhibiting significant increases or decreases in the normalized luciferase activity were chosen as candidates.

## Plasmids

The p125 luc reporter contains the IFN-β promoter region from −125 to +19 (Kouwaki et al, 2023). The promoter region of p125-luc reporter plasmids was replaced with the promoter region of human IFN-W1 (−900 to −1), RIG-I (−1000 to −1), and IRF7 (−500 to −1) using In-Fusion Snap Assembly Master Mix (Takara). RIG-I and RIG-I CARDs expression vectors were described previously (Oshiumi et al, 2013). The SREBP1a-N and SREBP1c-N expression vectors contain a 1461 bp region from the 5' ends of SREBP1 CDS (NM004176.5) and a 1389 bp region from the 5' ends of SREBP1 isoform 3 CDS (NM001321096.3), respectively. These regions are fused to a FLAG tag and cloned into a pEF-BOS vector using In-Fusion (Takara). The SREBP2-N expression vector contains a 1443 bp region from the 5' end of SREBP2 CDS (NM004599.4) fused to a FLAG tag and cloned into the pEF-BOS vector using In-Fusion Takara).

## Mice

C57BL/6JJcl mice were purchased from Kyudo. Leptin KO (ob/ob) mice and control WT mice were purchased from Jackson Laboratories, and maintained at the Kumamoto University Center for Animal Resources and Development (CARD) in a specific pathogen-free (SPF) environment. All experimental procedures were approved by the Institutional Animal Committee of Kumamoto University and performed according to their guidelines. To isolate the alveolar macrophages, the lungs were flushed with 800 μL BAL buffer (PBS with 2 mM EDTA and 0.5% EDTA) at least five times. The collected BAL fluid (BALF) was centrifuged at $400 \times g$.

## Cholesterol concentration

Cellular and mouse serum cholesterol levels were measured using the Cholesterol/Cholesterol Ester-Glo Assay Kit (Promega). Cells were cultured in 96-well plates and then treated with the compounds at various concentrations. The cells were gently washed with PBS three times, and cell lysates were prepared according to the manufacturer's protocol. Blood was collected from the tail vein of each mouse and allowed to stand for 30 min in tubes. Incubated blood was centrifuged at $1300 \times g$, and then serum was collected. The serum was diluted 50–80 times using the reagents provided in the kit. Diluted samples were dispensed into 96-well plates, and then experiments were performed according to the manufacturer's instructions. The luminescence level was measured using a FilterMAX F5 microplate reader, and cholesterol concentrations were determined based on standard curves.

## Microarray analysis

HeLa cells were cultured in 24-well plates and treated with DMSO or pitavastatin. After 24 h, the cells were infected with SeV at a multiplicity of infection (MOI) of 10 for 24 h. Total RNA was isolated using TRIZOL reagent. cDNA and Amino Allyl aRNA were synthesized using the Amino Allyl MessageAmp II aRNA Amplification Kit (Ambion#1753). CyeDye coupling and fragmentation were performed according to the protocol supplied by TORAY. Hybridization was performed for 16 h at 37 °C using a rotary shaker (set at 250 rpm). The hybridization buffer and washing protocol were performed according to the protocol provided by TORAY. A 3D-Gene scanner (TORAY) was used for scanning. Images were quantified by extraction (TORAY). The raw data for each spot were normalized by substituting it with the mean intensity of the background signal, which was determined using the signaling intensities of all blank spots within 95% confidence intervals. The microarray data were deposited in GEO under the accession number GSE199783. ISGs were selected from the GeneCards (https://www.genecards.org) database by searching "interferon-stimulated gene" and choosing only protein-coding genes.

## Western blotting

Cells were washed with PBS and lysed with lysis buffer (20 mM Tris-HCl pH 7.5, 125 mM NaCl, 1 mM EDTA, 10% glycerol, 1% NP-40, 30 mM NaF, 5 mM $Na_3VO_4$) in the presence of a protease inhibitor cocktail (Sigma-Aldrich). Cell lysates were placed on ice for 30 min and were subsequently centrifuged for 20 min at

15,000 rpm at 4 °C. The supernatants were transferred to 1.5 mL tubes. Protein levels were measured using the BCA Protein Assay Kit (TaKaRa). β-mercaptoethanol was added to cell lysates, which were subsequently boiled for 5 min at 95 °C. Samples and protein markers (Bio-Rad) underwent Tris-glycine SDS-polyacrylamide gel electrophoresis (SDS-PAGE) and were transferred onto PVDF membranes. The membrane was blocked with 5% skim milk in rinse buffer (0.1% tween 20, 10 mM Tris-HCl at pH 7.5, 0.8% NaCl, and 1 mM EDTA) and subsequently incubated with primary antibody (1:1000) at 4 °C overnight, and then incubated with HRP-conjugated secondary antibody (1:10,000) for 60 min at room temperature. Immunoblots were visualized using Amersham ECL prime western blotting detection reagent (GE Healthcare) and detected with a Bio-Rad ChemiDoc Touch imaging system. The antibodies used for western blotting are listed in the Reagents and Tools Table. To detect IFN-ω protein, $2 \times 10^5$ HepG2 cells were inoculated in a 24-well plate. After overnight incubation, HepG2 cells were treated with 2 μM of pitavastatin. Whole-cell lysates were harvested at indicated time points and subjected to SDS-PAGE.

## Reporter assay

HEK293 cells were cultured in 24-well plates and transfected with the indicated reporter and protein expression plasmids together with the pEF-BOS-Renilla plasmid using Lipofectamine 2000. Cell lysates were prepared 24 h after transfection, and luciferase and Renilla luciferase activities were determined using the Dual-Luciferase Reporter Assay system (Promega). The luminescence level was measured using a luminometer (ATTO), and the promoter activity was determined by normalizing luciferase activity to *Renilla* luciferase activities.

## ELISA

Cells were cultured in 24-well plates and treated with pitavastatin for 24 h, and infected with SeV or Flu for 24 h. Cell culture supernatants were collected and analyzed for IFN-β using an ELISA kit (Verikine Human IFN-Beta ELISA kit) (PBL). Serum was collected from the mouse tail vein and IFN-β protein levels were measured with the ELISA kit according to the manufacturer's instructions.

## RT-qPCR

Total RNA was isolated using TRI reagent (MOR), according to the manufacturer's protocol and then treated with DNase I. cDNA was generated using a high-capacity cDNA Reverse Transcription Kit (Life Technologies). RT-qPCR was performed with Luna Universal qPCR Master Mix (Biolabs) using the Step-One Real-time PCR system (Life Technologies). Target mRNA expression was normalized to the expression of GAPDH or β-actin, as indicated. The fold increase was calculated by dividing the mRNA levels at the indicated time points by those at 0 h or by dividing the mRNA levels following pitavastatin treatment with the vehicle. The primer sequences are listed in Reagents and Tools Table.

## ChIP-qPCR

$4 \times 10^6$ of HEK293 cells were cultured in a 10 cm dish and transfected with the indicated FLAG-tagged protein expression vectors. After 24 h, cells were subjected to a ChIP assay, according to the manufacturer's instructions (Cell Signaling Technology number 9003). The cells were sonicated 15 times, at 30 s intervals. Immunoprecipitation was performed using an anti-FLAG antibody and control IgG. The qRT-PCR was performed using the primers listed in the Reagents and Tools Table.

## 10X single-cell RNA sequencing

Pitavastatin calcium was dissolved in DMSO to achieve a concentration of 25 mg/mL. Thirteen-week-old female mice were intraperitoneally injected with 8 μL of DMSO or pitavastatin calcium solution in 200 μL of PBS (final pitavastatin dosage: 10 mg/kg/day) for 3 days. The lung tissues were harvested, snap-frozen in liquid nitrogen, and stored at −80 °C. The lung tissue was fixed with formaldehyde and dissociated using a GentleMACS Octo Dissociator (Miltenyi Biotec). After hybridizing the detection probe from the chromium-fixed RNA kit for the mouse transcriptome (10x Genomix) with intracellular RNA, a ligation reaction was performed to join the hybridized probes. After mixing each sample to ensure the number of cells was equal, the barcoded primer beads and cells were encapsulated using the Chromium Next GEM Chip Q Single Cell Kit on the Chromium X platform (10x Genomix). The probe and primer were annealed, and an extension reaction was performed to create a detection probe. Thereafter, a sequence library was prepared by PCR amplification and purification using the indexed primers included in the Dual Index Kit TS Set A (10x Genomix). A quality check of the sequence library was performed with an Agilent 2100 BioAnalyzer using a high-sensitivity DNA kit. Libraries were sequenced on an Illumina NovaSeq X Plus in pair-end mode. Library preparation and sequencing after tissue fixation were performed by Takara Bio Inc.

## Processing of the scRNA-seq data set

Raw sequencing data were processed and the reads were aligned to the mouse genome (mouse mm10; refdata-gex-mm10-2020-A). Following this, gene expression was quantified using Cell Ranger (v.7.1.0) software. The expression matrices were processed in R (v.4.3.1) using Seurat (v.5.0.1) as described below. We only selected the genes whose expression (unique molecular identifiers; UMI ≥ 1) was detected in >3 cells. Cells with more than 200 detected UMIs were retained, and filtering was applied (≥200 or <5000 genes per cell and % mitochondrial UMI counts <5). In total, 41,259 cells (DMSO group: 18,415 cells and pitavastatin group: 22,844 cells) were selected for further analysis out of a total of 42,606 cells (DMSO group: 19,024 cells and pitavastatin group: 23,582 cells). Data were normalized, scaled, and subjected to principal component analysis (PCA) using 2000 highly variable genes identified using the Seurat vst algorithm. We used the IntegrateLayers function with the HarmonyIntegration method to integrate different samples without the batch effect. The RunUMAP, FindNeighbors, and FindClusters functions were applied for clustering, with 30 dimensions identified using Harmony at a 0.5 resolution. For automatic cell type annotation, we used Single R (v2.2.0) software and LungMAP_MouseLung_CellRef (celltype_level3) (Guo et al, 2023) data sets. For extracting the alveolar macrophage subcluster, we chose cells defined as "AM" by LungMAP_MouseLung_CellRef and applied another filter using CellSelector software. The marker genes specifically expressed in each cluster were identified using the FindAllMarkers function and represented using a dot plot and heatmap using the Seurat function.

To perform cell cycle scoring, a list of cell cycle markers was loaded, and the CellCycleScoring function in Seurat was employed. The Monocle 3 software was used for trajectory analysis of the alveolar macrophage. To analyze the mean expression level of given gene signatures, we applied AddModuleScore functions in addition to a calculation method described in previous reports (Higa et al, 2022; Oka et al, 2023). Using the abovementioned methods, the normalized and log-scaled expression value of each gene was min–max scaled to adjust each gene to have an equal dynamic range (max = 1.0, min = 0). The mean expression levels of the gene set for each single cell were then calculated.

## Statistical analysis

Error bars represent the standard deviation (SD). Statistical significance (p-value) was determined using a two-tailed Student's t-test, one-way ANOVA, or two-way ANOVA using Prism ver. 10 (GraphPad Software).

# Data availability

Microarray and scRNA-seq data were deposited into the GEO database, and the accession numbers were GSE199783 and GSE264004, respectively.

The source data of this paper are collected in the following database record: biostudies:S-SCDT-10_1038-S44319-024-00346-9.

# Peer review information

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

## Acknowledgements

We thank our laboratory members for their helpful discussion and technical assistance. This work was supported in part by the Japan Society for the Promotion of Science (JSPS, Grant No.: 23H02949), the Japan Agency for Medical Research and Development (AMED, Grant No. 23gm1610011h), the Kose Cosmetology Foundation, Takeda Science Foundation (2023040769), and the Uehara Memorial Foundation (202310133).

## Author contributions

**Tasuku Nishimura**: Data curation; Formal analysis; Investigation; Methodology. **Takahisa Kouwaki**: Supervision; Funding acquisition; Investigation; Methodology. **Ken Takashima**: Funding acquisition; Investigation. **Akie Ochi**: Investigation. **Yohana S Mtali**: Investigation. **Hiroyuki Oshiumi**: Supervision; Funding acquisition; Writing—original draft; Project administration; Writing—review and editing.

Source data underlying figure panels in this paper may have individual authorship assigned. Where available, figure panel/source data authorship is listed in the following database record: biostudies:S-SCDT-10_1038-S44319-024-00346-9.

## Disclosure and competing interests statement

The authors declare no competing interests.

# Expanded View Figures

**Figure EV1.  Effects of statins on HeLa cells.**

(**A–H**) HeLa cells were treated with pitavastatin or DMSO (mock) for 24 h, Subsequently, they were stimulated by either transfecting 200 ng/mL of short poly I:C (**A**, **B**), infection with SeV (**C**) at MOI = 10 or Flu at MOI = 10 (**D**), adding poly I:C (50 μg/mL) to cell culture medium (**E–G**), or transfecting cGAMP (8 μg/mL) (**H**). siRNAs for negative control (si Con) and TLR3 (si TLR3) were transfected into HeLa cells for 2 days and then stimulated by adding poly I:C (50 μg/mL) to the cell culture medium (**G**). The expression of each gene was determined by RT-qPCR and normalized to that of GAPDH ($n = 3$, (**A–C**); technical replicates, (**D–H**); biological replicates). (**I–K**) HepG2 (**I**), A549 (**J**), and HeLa (**K**) cells were treated with pitavastatin (**I**, **J**), cerivastatin (**K**), or solvent (DMSO) for 24 h, and the cellular cholesterol levels were measured. Statin-pretreated cells were infected with SeV for 24 h, and then the expression of IFN-β mRNA was determined by RT-qPCR and normalized to that of GAPDH ($n = 3$ biological replicates). (**L**, **M**) HeLa (**L**) and HepG2 (**M**) cells were transfected with HMGCR or control siRNA. Twenty-four hours after siRNA transfection, the cells were stimulated by short poly I:C transfection (200 ng/mL) for the indicated time period. The mRNA expression of each gene was determined using RT-qPCR and normalized to that of GAPDH ($n = 3$ biological replicates). Data Information: in (**A–M**), Data are represented as mean ± SD ($n = 3$, ****$p < 0.0001$, one-way ANOVA (**I–K**), two-way ANOVA (**A–H**), NS: not significant).

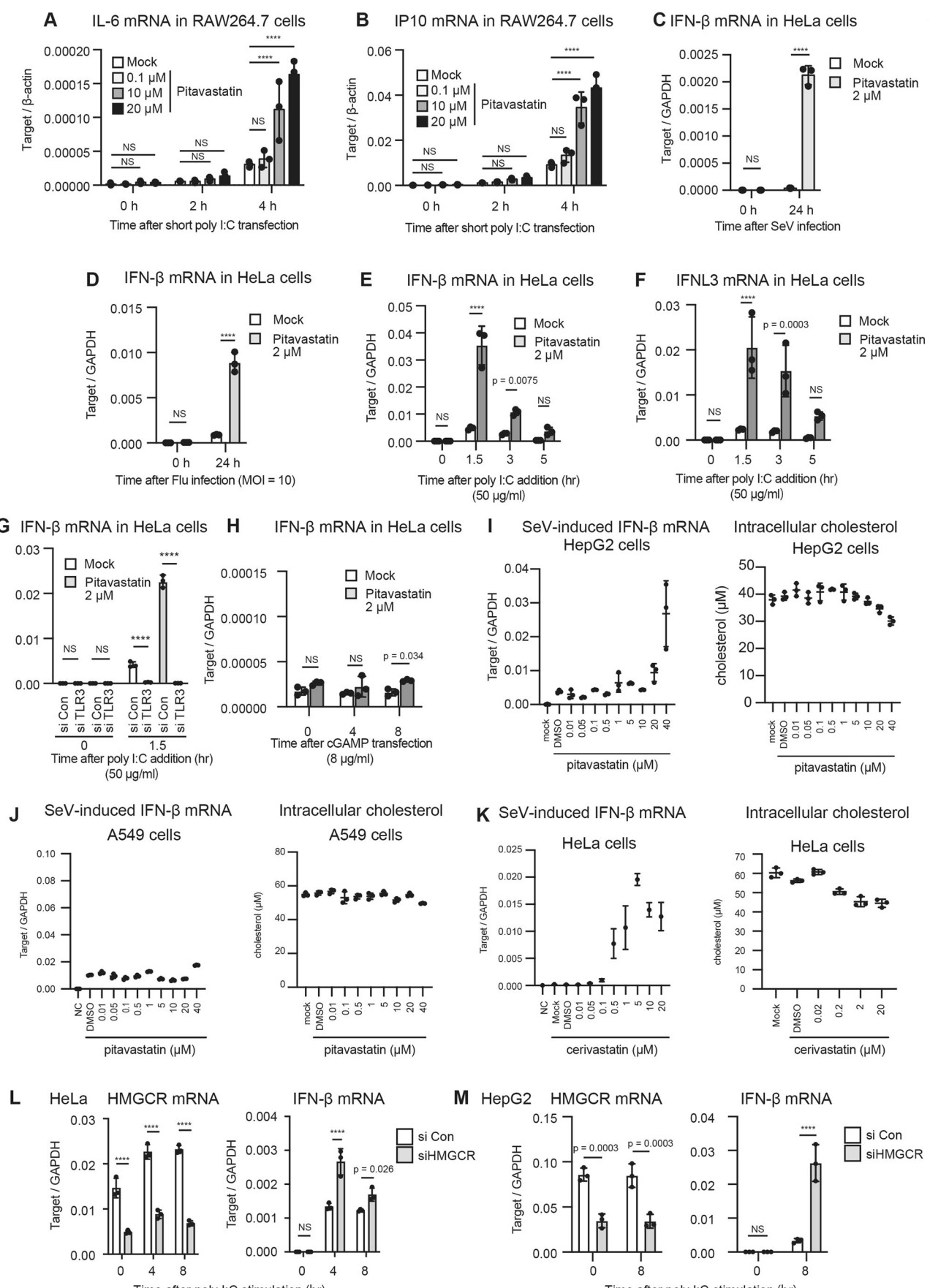

**A**

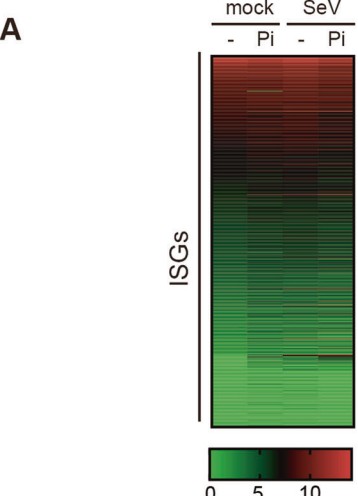

**B**

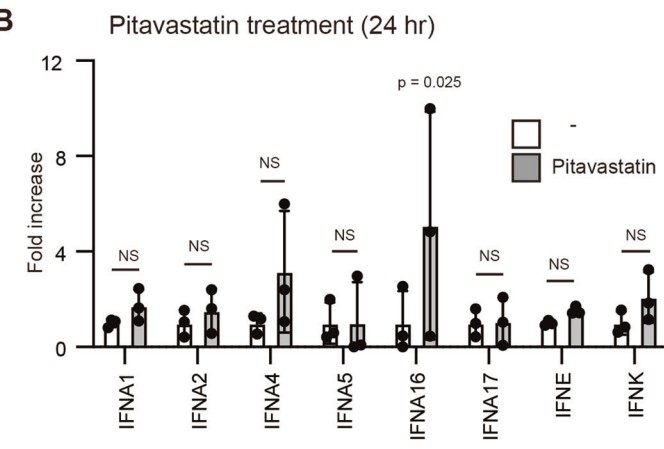

**C**

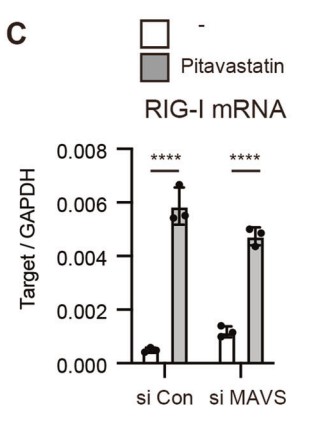
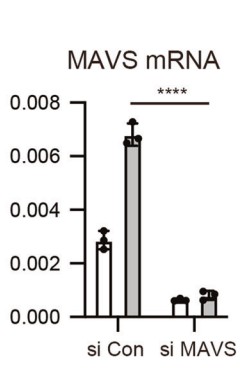

**D**

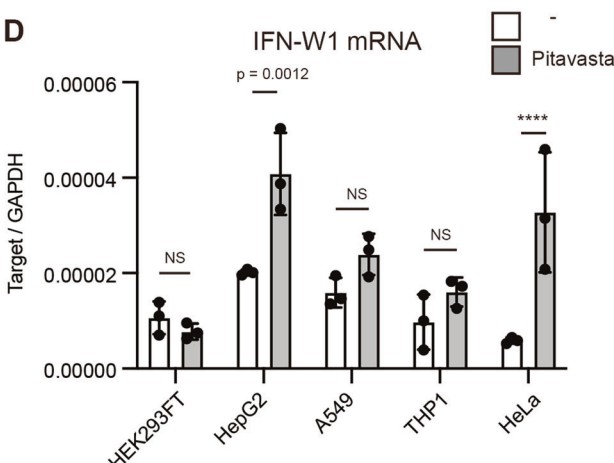

**Figure EV2. Expression of non-canonical type I IFNs after pitavastatin treatment.**

(A) Heatmap of interferon-stimulated genes (ISGs) from the microarray analysis presented in Fig. 2. ISGs were selected by searching for interferon-stimulated genes encoding proteins in the GeneCARDs database (https://www.genecards.org). (B) HeLa cells were treated with pitavastatin for 24 h. The expression of each gene was determined by RT-qPCR and normalized to that of β-actin. The fold increase was calculated by dividing the normalized expression level of a pitavastatin-treated sample by a pitavastatin-untreated sample value ($n = 3$, biological replicates). (C) HeLa cells were transfected with siRNA for MAVS (si MAVS) and negative control (si Con) for 72 h. Cells were then treated with pitavastatin for 48 h. The expression of RIG-I and MAVS were determined by RT-qPCR and normalized to GAPDH ($n = 3$, biological replicates). (D) HEK293, HepG2, A549, THP-1, and HeLa cells were treated with pitavastatin for 24 h, and IFN-ω expression was determined by RT-qPCR and normalized to that of GAPDH ($n = 3$, technical replicates). Data Information: In (B–D), data are represented as mean ± SD (****$p < 0.0001$, two-way ANOVA, NS: not significant).

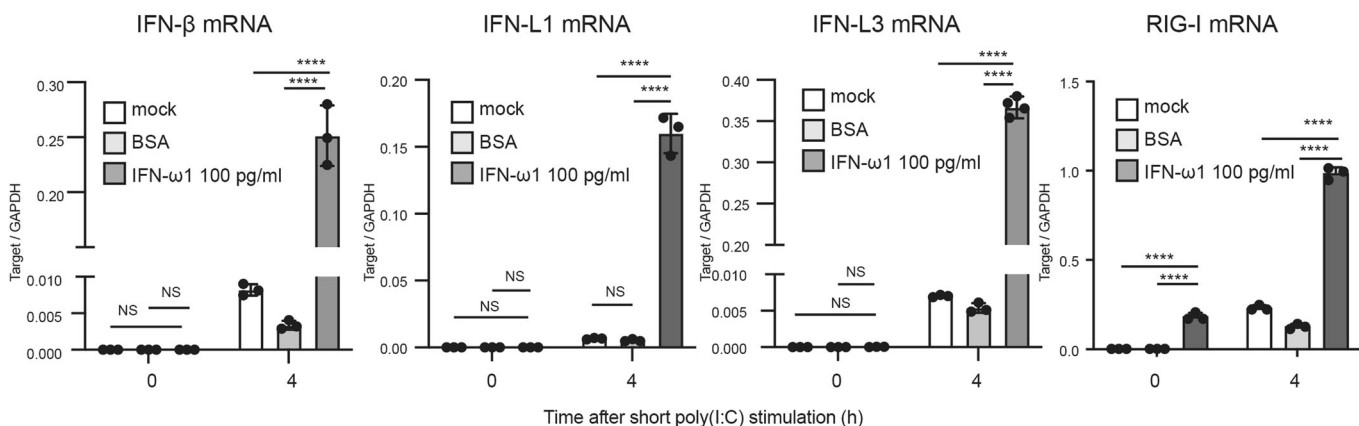

**Figure EV3. Effect of IFN-ω on the RIG-I-dependent innate immune response.**

HeLa cells were treated with 100 pg/mL of recombinant IFN-ω protein for 24 h, and stimulated with short poly I:C (200 ng/mL) transfection. The expression of each gene was determined by RT-qPCR and normalized to that of GAPDH. Data Information: Data are represented as mean ± SD ($n = 3$, biological replicates, ****$p < 0.0001$, two-way ANOVA, NS: not significant).

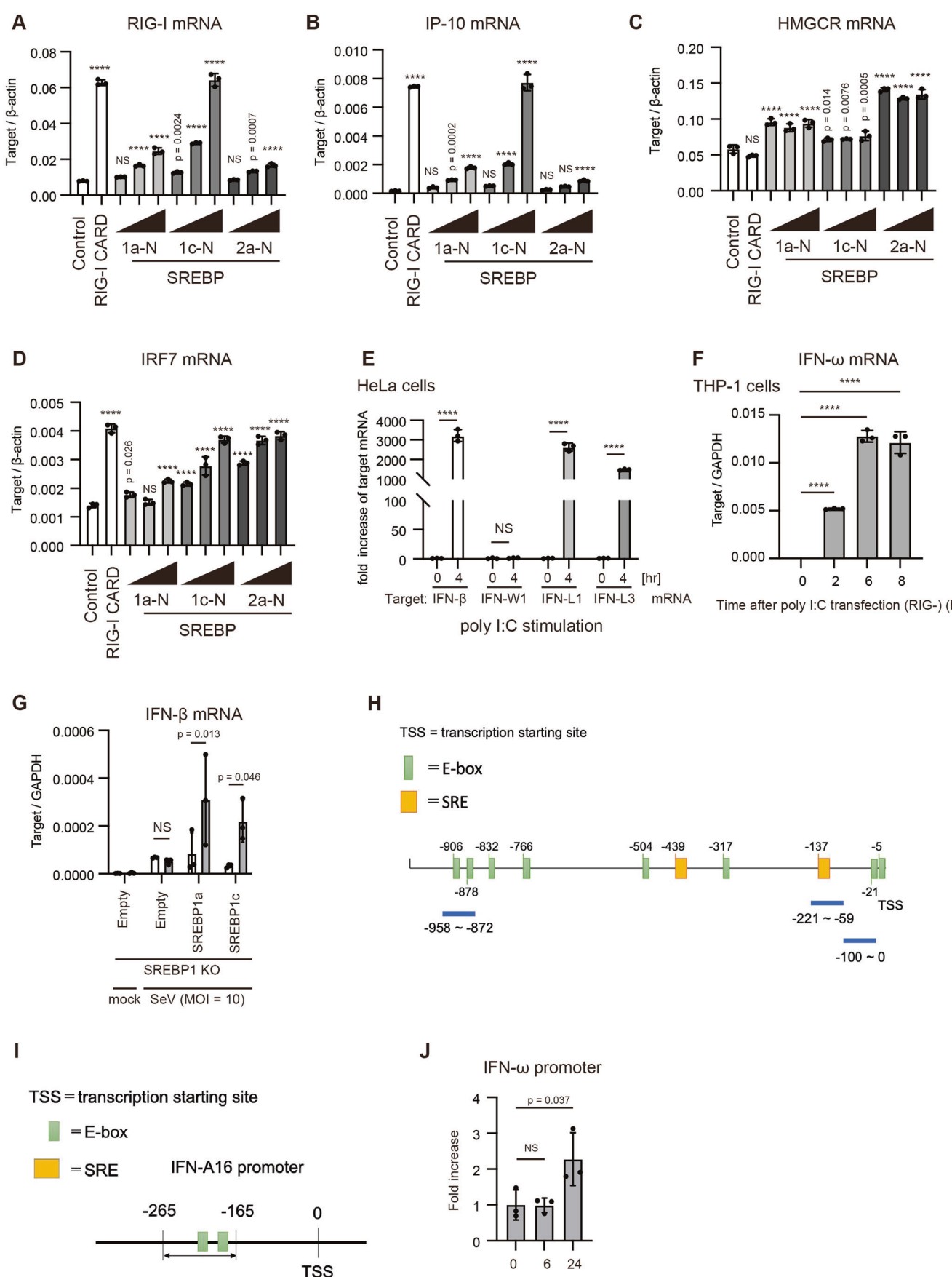

◄ **Figure EV4.  The role of SREBPs in type I IFN responses.**

(**A–D**) HEK293 cells were transfected with 200 ng/mL of RIG-I CARD expressing vectors or 200, 1000, or 2000 ng/mL of SREBP1a-N, 1c-N, or 2a-N expressing vectors in a 24-well plate. The total amount of DNA was adjusted to the same amount by adding an empty vector. Twenty-four hours after transfection, the expression of each gene was determined by RT-qPCR and normalized to that of β-actin. Each *p*-value represents the comparison with the control (*n* = 3, technical replicates). (**E**) HeLa cells were stimulated via transfection with 200 ng/mL of short poly I:C, and the mRNA expression of each gene was determined by RT-qPCR. The fold increase was calculated by dividing mRNA expression at each time point by that at 0 h (*n* = 3, biological replicates). (**F**) THP1 cells were stimulated via transfection with short poly I:C, and the expression of IFN-ω mRNA was determined by RT-qPCR and normalized to that of GAPDH (*n* = 3, technical replicates). (**G**) SREBP1 KO HeLa cells were transfected with empty vector (Empty) or full-length SREBP1a or SREBP1c. Twenty-four hours after transfection, cells were treated with pitavastatin for 24 h, and subsequently infected with SeV or mock for 24 h. The expression of IFN-β mRNA was measured by RT-qPCR and normalized to GAPDH (*n* = 3, biological replicates). (**H, I**) Promoter regions of human IFN-ω and IFN-α16. The E-box and SRE motifs are shown in the green and yellow boxes, respectively. Numbers represent the positions of the nucleotides from the start of the ORF. (**J**) The IFN-ω reporter plasmid was transfected into HeLa cells with *Renilla* luciferase plasmid (internal control). Twenty-four hours after transfection, cells were treated with pitavastatin (2 μM) at indicated h. Cell lysates were prepared, and the luciferase activities were measured and normalized to *Renilla* luciferase activities (*n* = 3, biological replicates). Data Information: In (**A–G, J**), data are represented as mean ± SD (****$p$ < 0.0001, one-way ANOVA (**A–D, J**), two-way ANOVA (**E, G**)).

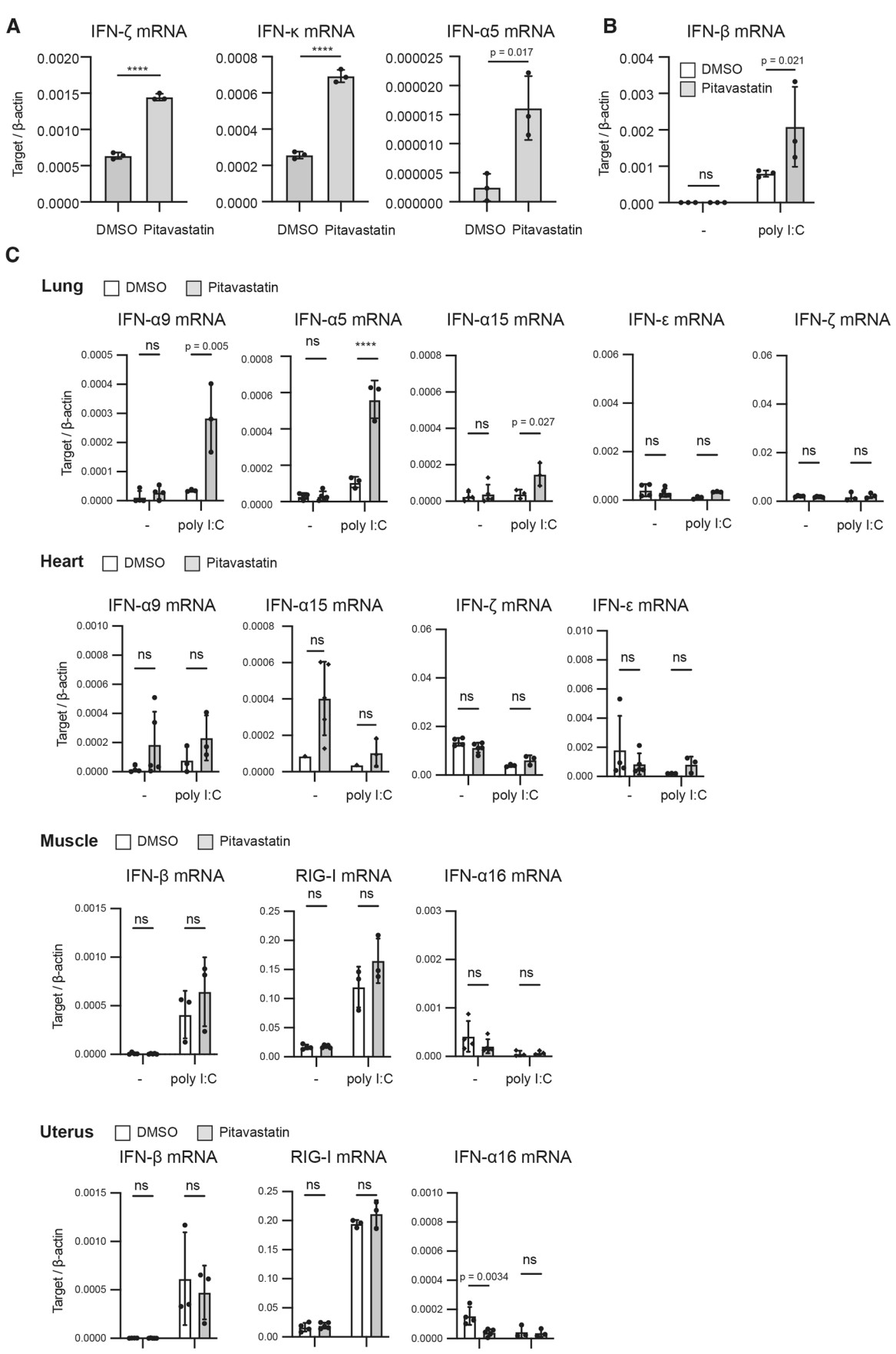

Figure EV5.  Tissue-specific effects of pitavastatin on type I IFN responses.

(A) BMDMs of mice were treated with pitavastatin (10 μM) for 24 h, and the expression of noncanonical type I IFNs was measured by RT-qPCR normalized to β-actin. (B) BMDMs treated with pitavastatin (10 μM) for 24 h were stimulated with transfection of poly I:C (200 ng/ml) for 4 h. The expression of IFN-β mRNA was measured by RT-qPCR normalized to β-actin ($n = 3$, biological replicates). (C) Pitavastatin or solvent (DMSO) was intraperitoneally injected into mice every day for three days, followed by intraperitoneal injection of poly I:C (200 μg/head) or solvent (DMSO). Twenty-four hours after poly IC injection, the lung, heart, muscle, and uterus were isolated. The total RNAs were extracted from each tissue. The expression of each gene was determined by RT-qPCR and normalized to that of β-actin ($n = 3$–6 mice for each group). Data Information: In (A, C), data are represented as mean ± SD. Statistical significance was determined using t-test (A) and two-way ANOVA (B, C). ****$p < 0.0001$. ns: not significant.

