## [Peer Review File · EMBO Reports]

Cholesterol Restriction Primes Antiviral Innate Immunity via SREBP1-Driven Noncanonical Type I IFNs

Tasuku Nishimura, Takahisa Kouwaki, Ken Takashima, Akie Ochi, Yohana Mtali, and Hiroyuki Oshiumi

Corresponding author(s): Hiroyuki Oshiumi (oshiumi@kumamoto-u.ac.jp), Takahisa Kouwaki (kouwaki@kumamoto-u.ac.jp)

Review Timeline:

Submission Date:	22nd May 24
Editorial Decision:	8th Jul 24
Revision Received:	7th Oct 24
Editorial Decision:	15th Nov 24
Revision Received:	24th Nov 24
Editorial Decision:	27th Nov 24
Revision Received:	28th Nov 24
Accepted:	28th Nov 24

Editor: Achim Breiling

Transaction Report:

Dear Dr. Oshiumi,

Thank you for the transfer of your manuscript to EMBO reports. I have now received the reports from the three referees that were asked to evaluate your study, which can be found at the end of this email. As you will see, the referees have several comments, concerns, and suggestions, indicating that a major revision of the manuscript is necessary to allow publication of the study in EMBO reports. As the reports are below, and all the concerns need to be addressed, I will not detail them further here.

Given the constructive referee comments, I would like to invite you to revise your manuscript with the understanding that the concerns of the referees must be addressed in the revised manuscript or in a detailed point-by-point response. Acceptance of your manuscript will depend on a positive outcome of a second round of review. It is EMBO reports policy to allow a single round of revision only and acceptance of the manuscript will therefore depend on the completeness of your responses included in the next, final version of the manuscript.

Revised manuscripts should be submitted within three months of a request for revision. Please contact me to discuss the revision (also by video chat) if you have questions or comments, or should you need additional time.

1) a .docx formatted version of the final manuscript text (including legends for main figures, EV figures and tables), but without the figures included. Figure legends should be compiled at the end of the manuscript text.

2) individual production quality figure files as .eps, .tif, .jpg (one file per figure), of main figures and EV figures. Please upload these as separate, individual files upon re-submission.

4) a complete author checklist, which you can download from our author guidelines

(<https://www.embopress.org/page/journal/14693178/authorguide>). Please insert page numbers in the checklist to indicate where the requested information can be found in the manuscript. The completed author checklist will also be part of the RPF.

5) that primary datasets produced in this study (e.g. RNA-seq, ChIP-seq, structural and array data) are deposited in an appropriate public database. If no primary datasets have been deposited, please also state this in a dedicated section (e.g. 'No primary datasets have been generated and deposited'), see below.

The accession numbers and database should be listed in a formal "Data Availability" section (placed after Materials & Methods) that follows the model below. This is now mandatory (like the COI statement). Please note that the Data Availability Section is restricted to new primary data that are part of this study. This section is mandatory. As indicated above, if no primary datasets have been deposited, please state this in this section

Data availability

8) Regarding data quantification and statistics, please make sure that the number "n" for how many independent experiments were performed, their nature (biological versus technical replicates), the bars and error bars (e.g. SEM, SD) and the test used to calculate p-values is indicated in the respective figure legends (also for EV figures and all those in an Appendix). Please also check that all the p-values are explained in the legend, and that these fit to those shown in the figure. Please provide statistical testing where applicable. Please avoid the phrase 'independent experiment', but clearly state if these were biological or technical replicates. Please also indicate (e.g. with n.s.) if testing was performed, but the differences are not significant. In case n=2, please show the data as separate datapoints without error bars and statistics. See also: <http://www.embopress.org/page/journal/14693178/authorguide#statisticalanalysis>

9) Please also note our reference format:

10) We updated our journal's competing interests policy in January 2022 and request authors to consider both actual and perceived competing interests. Please review the policy <https://www.embopress.org/competing-interests> and update your competing interests if necessary. Please name this section 'Disclosure and Competing Interests Statement' and put it after the Acknowledgements section.

11) We now use CRediT to specify the contributions of each author in the journal submission system. CRediT replaces the author contribution section. Please use the free text box to provide more detailed descriptions and do NOT provide your final manuscript text file with an author contributions section. See also our guide to authors: <https://www.embopress.org/page/journal/14693178/authorguide#authorshippinguidelines>

12) All Materials and Methods need to be described in the main text using our 'Structured Methods' format, which is required for all research articles. According to this format, the Materials and Methods section should include a Reagents and Tools Table (listing key reagents, experimental models, software, and relevant equipment and including their sources and relevant identifiers), uploaded as separate file, followed by a Methods and Protocols section in which we encourage the authors to describe their methods using a step-by-step protocol format with bullet points, to facilitate the adoption of the methodologies across labs. More information on how to adhere to this format as well as downloadable templates (.doc) for the Reagents and Tools Table can be found in our author guidelines (section 'Structured Methods'):

Finally, please provide the abstract written in present tense and order the manuscript sections like this, using these names: Title page - Abstract - Keywords - Introduction - Results - Discussion - Methods - Data availability section - Acknowledgements - Disclosure and Competing Interests Statement - References - Figure legends - Expanded View Figure legends

I look forward to seeing a revised version of your manuscript when it is ready. Please let me know if you have questions or comments regarding the revision.

Yours sincerely,

Referee #1:

This is an interesting paper that suggests cholesterol limitation is associated with the enhancement of RIG-I signaling via cell-type specific induction of non-canonical type I IFNs. In vivo they found they found suppressing cholesterol synthesis increased RIG-I expression in certain subsets of alveolar macrophages. The authors have generated a large amount of supporting in vitro and in vivo data to support their claims. However, this makes the paper somewhat dense, and I felt that many of the results were discussed superficially. Specific examples of results discussed in insufficient detail include the screening, the microarray, and the single-cell RNA-seq.

Comments:

- Authors studied a variety of different cell lines in their in vitro studies and often reported cell-specific effects. However, they provide little explanation or thought as to what may account for these differences. It is also unclear, given their in vivo findings with alveolar macrophages, why more emphasis wasn't put into the characterization of THP-1 and RAW cells. Also, would be helpful to provide some justification for their differential use across experiments. Also, need some acknowledgment of potential limitations in using cell lines in discussion

- Screening is identified as a key component of the study no specific results were presented. How many different statins were pulled out of the screening? It is unclear exactly how 22 compounds were identified other than mentioning that they examined the ratio of mCherry to AcGFP.

- Unclear what role TLR3 may play in these studies. For in vitro studies, the authors transfected cells with Poly(i:c), which should limit TLR activation but are you sure? The bigger concern is for in vivo work. Do you think TLR3 could be contributing? Should be investigated further.

- Would have liked to see some level of evaluation of protein expression particularly for RIG and IFN- ω in in vitro models.

Other minor comments:

- Missing details on the number of technical and biological replicates across many of the experiments. Looks like mostly n=3 but should be mentioned somewhere.

- Limited information is provided for the infection protocols. Would like to see some type reference for this work.

Referee #2:

This study by Nishimura et al. finds that restricting cholesterol synthesis (by pitavastatin treatment) increases the basal expression of non-canonical type I interferons, which subsequently primes the RIG-I pathway, leading to enhanced canonical interferon responses. This is dependent on the transcription factor SREBP1, which binds to the promoter of non-canonical IFNs (e.g. IFN- ω) and drives their expression. The authors recapitulate some of their findings in mouse models, i.e. they demonstrate that *in vivo* treatment with pitavastatin upregulates certain non-canonical IFNs. Finally, the authors employ scRNAseq to demonstrate that a specific subset of mouse alveolar macrophages expresses RIG-I in response to pitavastatin. Overall, the manuscript reports some interesting observations. However, many important controls and experimental details are lacking, especially in Fig. 1 to 4. In addition, the overall model and message is not always clear and linear, and the data are not always consistent with the conclusions that are drawn and the overall model. For example, it remains vague to what extent pitavastatin treatment impacts on the IFN- β promoter and increases canonical IFN production as well. Also, the data regarding the involvement of SREBP1/2 are confusing. Finally, it is unclear why the proposed model would selectively prime RIG-I-dependent responses, but not MDA5-dependent responses. The *in vivo* treatment of mice with pitavastatin is interesting and worthwhile to report, but the scRNAseq experiments described in Figure 6/7 do not add much value to this manuscript. I have addressed my major concerns here point-by-point. These concerns should be addressed by major revisions in order to make this manuscript suitable for publication in EMBO Reports.

Major points:

- Fig. 1: how does pitavastatin treatment impact on the response to high-molecular weight poly(I:C) or EMCV infection, both of which trigger an MDA5-dependent IFN response? If pitavastatin treatment upregulates non-canonical IFNs, then this should also lead to increased MDA5 expression and responses. I.e. it is unclear why the RIG-I pathway would be selectively primed.
- Fig. 1C-J: Are the cells that are treated with or without statins equally well transfected with poly(I:C) or infected with SeV? I am concerned that the loss of cholesterol through statin treatment or HMGCR depletion may impact on membrane composition and the infectivity/transfectability of these cells.
- Fig. 1: Is the increased IFN response in pitavastatin treated cells leading to reduced viral replication? This could easily be tested by qPCR analysis of SeV viral transcripts.
- Fig. EV1: how do the authors explain that pitavastatin treatment increases RIG-I and TLR3-induced IFN responses but not cGAMP-induced IFN responses?
- Fig. 2: the effects of cholesterol depletion on basal IFN/ISG expression should be recapitulated with HMGCR knockdown, as an alternative to pitavastatin treatment.
- The data in Fig. 2A-C suggests that the Pitavastatin-treated cells have an elevated basal IFN/ISG response. It would be informative to include all ISGs from the microarray analysis in a single heatmap to illustrate this.
- Fig. 2D-E: as above. Which of these targets are ISGs? Please indicate this in the figure, e.g. by color coding.
- Is the increase in the basal IFN/ISG response dependent on signaling via RLRs or cGAS? This could easily be addressed by repeating the experiment from Fig. 2D in cells depleted of MAVS or STING by siRNAs (or, as a positive control, siIFNAR).
- The authors conclude that pitavastatin treatment induces non-canonical IFNs, but not canonical IFNs such as IFN- β , at steady state. This conclusion is based on the microarray data in Fig. 2B. To convincingly make this conclusion, IFN α and IFN β levels should also be measured by qPCR in Fig. 2D and directly compared to IFNL3 and IFN ω .
- Fig. 3H/I: the authors find that the effect of pitavastatin on basal ISG levels is IFNAR2 dependent at early time points but not at late time points. This may be explained by the production of type III IFNs (Fig. 2E: increased expression of IFN-L3) that signal through the type III IFN receptor (not IFNAR). The authors can test this by knocking out or knocking down the type III IFN receptor or, alternatively, by using the JAK inhibitor ruxolitinib, which blocks both type I and type III IFN signaling.
- Fig. EV3A: 4 hrs of poly(I:C) treatment should be sufficient to increase IFN β transcripts in mock or BSA treated cells (as the authors also show in EV4F). Why is that not visible in this experiment?
- Fig. 4: Given the set of tools and reagents that the authors have to their disposal, they could directly test whether pitavastatin treatment increases the expression of the IFN- ω reporter.
- Fig. 4A/B: why is the SREBP1c-N construct not causing a significant increase in IFN- β promoter reporter activity (panel A), while it strongly induces IFN- β mRNA expression (panel B). In the text the authors conclude that SREBP1 regulates non-canonical IFNs but not canonical IFNs (IFN- β). This conclusion cannot be drawn based on panel 4B.
- Fig. 4C: the FLAG-RIG-I-CARD construct is not visible on the FLAG immunoblot, suggesting it is not well expressed. In addition, expression of the RIG-I CARD domain should strongly induce an IFN response and upregulate ISGs, which is not evident from the RIG-I and MDA5 immunoblot.
- Fig. 4D: Validation experiments of the SREBP1/2 KO cells should be included. In addition, a rescue experiment should be performed in the SREBP1 KO cell line to demonstrate that the observed effects are truly due to the loss of SREBP1.
- Fig. 4F: the IFN- β promoter should be included as a control in these experiments to support the conclusion that SREBP1 localizes to the promoter of non-canonical IFNs but not IFN- β .
- Fig. 5D: were other (non-canonical) IFNs tested here? This would be important to validate the hypothesis that pitavastatin impacts on IFN β expression through upregulating non-canonical IFNs.
- Fig. 7E: other transcripts (non-canonical IFNs) should be measured as well, as done in Fig. 7B.

Minor points:

- The materials and methods section lacks details on the transfection procedure of different PAMPs (e.g. polyI:C, cGAMP). Details regarding the concentration, the commercial source, and the transfection reagent should be included.
- Page 6: "...Riplet protein, a coreceptor of RIG-I.". While Riplet is clearly important for RIG-I signaling, I believe the word co-receptor is inappropriate, as it implies that Riplet is binding to an RNA ligand as well, alongside RIG-I. Please rephrase this sentence.
- The name of the compound library should be mentioned in the text. Is this the Sigma Lopac library?
- Page 6: it is unclear in the text if the second screening with the p125 luc plasmid was done using the same compound library, or with the 22 hits from screen 1 only.
- Fig. 1B: indicate in the figure that this is an IFN β -Luc assay.
- Fig. 1E: indicate in the legend that cerivastatin was used in this experiment. Indicate in the legend what concentrations of statins were used.
- Fig. 1F: X-axis indicates that this is flu infection, while the text and legend refers to Sendai virus infection.
- Fig. 1E and 1F: is IFN-beta really completely undetectable after 24 or even 48 hrs of SeV infection? It is unclear why no IFN-beta is produced at all in mock-treated SeV infected cells (Fig. 1E) or in cells infected for 1 day with SeV (Fig. 1F).
- Fig. EV1G: was flu or Sendai virus used here?
- Referral to Fig. EV1G is lacking in the text.
- In Fig. 3 an IFNAR2 KO cell line is used. I assume this KO cell line was properly validated (e.g. through measurement of IFNAR expression by immunoblotting or flow cytometry, sequencing, or by measuring the response to recombinant IFN). Please include validation data in the supplement.
- Fig. 4B: right panel (IFN-w): is the 2a-N-mediated increase in IFN-w expression truly not significant? (highest dose of plasmid). Please double check the statistical analysis.
- I would suggest moving supplemental Fig. EV4A to the main figures.

 Referee #3:

The manuscript presents intriguing observations linking the inhibition of the cholesterol synthesis pathway mediated by statins with the induction of certain IFN-inducible genes. The authors describe the activation of some type I IFNs dependent on statin treatment, including IFN omega, certain types of IFN alpha, and IFN lambda. Additionally, they report that statin-mediated induction of certain ISGs is partially dependent on type I IFN signaling.

The manuscript also provides interesting data related to the alternative induction of certain IFNs dependent on SREBPs and SRE binding sites in the promoters of some IFN production genes. Statin treatment or similar mechanisms could induce the expression of these interferons and stimulate IFN-inducible genes. Statins (PMID: 34321273) and HDL receptor inhibitors SR-BI (BTX-I and ITX5061) have previously been described to decrease ISG expression through effects on the IFN signaling cascade (PMID: 27622065).

Main Concerns:

1. Title: The title does not reflect the discovery. It should mention that statins can stimulate the production of IFNs other than IFN-beta through a mechanism dependent on SREBP activation. It is not RIG-I dependent only, as activation of TLR3 can achieve the same effect.
2. Abstract: The abstract does not reflect the effect of statins in inhibiting type I IFN signaling.
3. While the study focuses on the effect of statins on type I IFN induction, it also shows that statins induce IFNs lambda.
4. The authors describe differential induction of various interferons dependent on SRE promoter elements but do not describe the potential position of these elements in all IFN promoters, including type III IFNs. The use of "non-canonical type-I IFN" should specify the IFNs. The canonical vs. non-canonical concept is not widely accepted.
5. The proposed mechanism suggests that statins can induce SREBPs, leading to cytokine production that stimulates the type I IFN signaling cascade. The possibility that type III IFNs production might induce the IFNLR-dependent signaling pathway, which could explain some experiments, is not discussed. The authors should test the effect of statins in IFNLR KO cells and STAT1 KO cells. They also describe the effect on alveolar macrophages, which depend on IFN-gamma. Do statins affect IFN-gamma signaling (IFNGR KO cells)?
6. Potential Effects: The proposed mechanism implies that SREBPs-dependent IFN induction produces certain IFNs that would activate the IFN signaling cascade, potentially increasing ISGs that could further potentiate the IFN induction pathway (types I, II, and III). The manuscript partially reflects this potential effect and should explore all possibilities. A graphical abstract would clarify these concepts.
7. Figure 7: The figure describes the effect of statins on alveolar macrophages, showing statin treatment affects the production of various type I IFNs. It should include the effect on IFN lambda and gamma induction.

8. The effect of statins on CD140+ macrophages is poorly presented. The authors should repeat the experiment with a larger sample size and analyze the effect more exhaustively, including more genes, both ISGs and IFNs.

Minor Concerns:

1. Introduction: The introduction does not discuss the effects of statins on the type I IFN signaling cascade, despite prior evidence (some references indicated before). The authors should consider expanding this section.
2. Results: The authors describe the effect of statins not only on the RLR pathway but also on TLR-3. Can statins affect TLR4, TLR7, and TLR8/9 signaling?
3. Reporter Assays: The promoters of IFN- β used in reporter assays are poorly described in the materials and methods section. Define the length of the promoters used.
4. Plasmids: The materials and methods section does not describe the plasmids used in reporter assays for other IFNs or RIG-I in the manuscript. Include descriptions of these promoters and their lengths.
5. Overexpression Plasmids: The manuscript mentions various plasmids for protein overexpression that are not detailed in the materials and methods. Include these details.
6. Virus details: The amount of virus used in some infection experiments is not defined. This should be described in the materials and methods section and in relevant results sections, including virus growth and titration.
7. Discussion: The discussion should better explain why the effect of statins is not universal (i.e., in A549 cells). Specify potential reasons and test primary cells, not just cell lines, such as in vitro differentiated macrophages.
8. Type III and II IFNs: Provide evidence of statins' effects on type III and type II IFN signaling (Figure 3).
9. SREBPs Analysis: Better describe the differential induction by various SREBPs analyzed, including their DNA binding motifs.
10. Promoter Analysis: Present an analysis of different SREs in the promoters of various IFNs (Figure 4).
11. Administration of Treatments: Treatment administration is only described as an injection, even for poly IC and lung treatments. Does this refer to intranasal administration? Review the document and specify whether the injection is intranasal, intravenous, intraperitoneal, retro-orbital, or otherwise. Define the administration methods and times until analysis in the text and all figure legends.
12. Cluster Numbering: Is it possible to order the numbering of clusters sequentially based on the possible evolution of the macrophage (Figure 6)?

Referee #1:

This is an interesting paper that suggests cholesterol limitation is associated with the enhancement of RIG-I signaling via cell-type specific induction of non-canonical type I IFNs. In vivo they found they found suppressing cholesterol synthesis increased RIG-I expression in certain subsets of alveolar macrophages. The authors have generated a large amount of supporting in vitro and in vivo data to support their claims. However, this makes the paper somewhat dense, and I felt that many of the results were discussed superficially. Specific examples of results discussed in insufficient detail include the screening, the microarray, and the single-cell RNA-seq.

Response:

We are grateful to you for your valuable comments. We have sufficiently revised our manuscript to address all comments as described below.

(Note: The line numbers referred to in this response are based on the original word file, as the line numbers in the automatically generated PDF version may differ.)

Comment 1:

- Authors studied a variety of different cell lines in their in vitro studies and often reported cell-specific effects. However, they provide little explanation or thought as to what may account for these differences. It is also unclear, given their in vivo findings with alveolar macrophages, why more emphasis wasn't put into the characterization of THP-1 and RAW cells. Also, would be helpful to provide some justification for their differential use across experiments. Also, need some acknowledgment of potential limitations in using cell lines in discussion.

Response:

Thank you for your insightful comment. In the revised manuscript, we have provided more detailed explanations of the possible mechanisms underlying the cell-type-specific effects of statins. First, we have added an explanation of the cell-type specific effects of statins (lines 208-211) and have discussed the possible mechanisms in the discussion, such as differences in epigenetic

regulation and varying levels of tolerance to statins among different cell types (487–493). These factors may influence how different cells respond to statin treatment, leading to the observed variability in effects. Second, we have provided additional data regarding RAW264.7 cells (Appendix Figure S1C–E) in which pitavastatin treatment enhance IFN- β mRNA expression in response to TLR3 or TLR4 stimulation in RAW264.7 cells. Additionally, we have investigated another type of macrophage, BMDMs, and found that pitavastatin treatment also enhances IFN- β expression in these cells (new Figure EV5A and B). As shown in Figure EV2C, pitavastatin hardly enhances IFN- β expression in THP-1 cells. This suggests that THP-1 cells are less responsive to statin treatment under our experimental conditions, possibly due to differences in epigenetic regulation. Therefore, we did not further characterize THP-1 cells. This observation also supports the notion of cell-type-specific effects of statins, as discussed in our manuscript. Third, we have acknowledged the potential limitations in using cell lines in discussion (lines 490–491). We believe that these new data, along with additional explanations and discussions, further strengthen our manuscript by providing a more comprehensive understanding of the cell-type-specific effects of statins, justifying our experimental approach, and highlighting areas for future research.

Comment 2:

Screening is identified as a key component of the study no specific results were presented. How many different statins were pulled out of the screening? It is unclear exactly how 22 compounds were identified other than mentioning that they examined the ratio of mCherry to AcGFP.

Response:

Thank you for your helpful comment. We regret that the explanation of the screening process was unclear. In this screening, we have identified three statins, pitavastatin, atorvastatin, and cerivastatin. To provide more clarity, we have improved Figure 1A and added detailed explanations of the chemical screening process in the results and methods (lines 112–120, 552–561), including the number of statins identified through screening and the criteria used to select the compounds.

Comment 3:

Unclear what role TLR3 may play in these studies. For in vitro studies, the authors transfected cells with Poly(i:c), which should limit TLR activation but are you sure? The bigger concern is for in vivo work. Do you think TLR3 could be contributing? Should be investigated further.

Response:

Thank you for your valuable comment. As the reviewer pointed out, it is still possible that transfected short poly I:C is recognized by TLR3 as well as RIG-I. However, it has been reported that TLR3 is dispensable for viral-induced type I IFNs in the case of SeV and Flu infection. We have added this explanation (lines 127–128) and cited references 17 and 35, which show that TLR3 is not involved in type I IFN expression following SeV and Flu infection. Moreover, we have confirmed that SeV-induced type I IFN expression was abolished by knockdown of MAVS, which is an essential adaptor of RIG-I (Appendix Figure S1A and B). These data further support that enhanced type I IFN response by pitavastatin depends on RIG-I pathways.

Additionally, to address your comment regarding the potential involvement of TLR3, we have further investigated the involvement of TLR3 in the enhanced IFN- β expression by statin treatment. Our new Fig EV1G shows that knockdown of TLR3 attenuated the type I IFN response induced by adding poly I:C to the cell culture medium without transfection, indicating that TLR3 contributes to the response under these conditions. Therefore, it is possible that TLR3 may contribute to the enhanced type I IFN response following statin treatment in vivo. We have added a discussion about the effect of statins on the TLR3 pathways in vivo (lines 458–463).

Comment 4:

Would have liked to see some level of evaluation of protein expression particularly for RIG and IFN- ω in in vitro models.

Response:

Thank you for your helpful comment. We have provided new data showing that pitavastatin treatment increased the protein levels of IFN- ω and RIG-I as well as

MDA5 (new Figures 2F and 3L).

Minor comment 1:

Missing details on the number of technical and biological replicates across many of the experiments. Looks like mostly n=3 but should be mentioned somewhere.

Response:

Thank you for your comments. We have added the information of the number of technical and biological replicates, along with statistical analyses, to each Figure legend where applicable.

Minor comment 2:

Limited information is provided for the infection protocols. Would like to see some type reference for this work.

Response:

We have added the protocols and references for the experiments utilizing viruses in the methods (lines 540–543).

Referee #2:

This study by Nishimura et al. finds that restricting cholesterol synthesis (by pitavastatin treatment) increases the basal expression of non-canonical type I interferons, which subsequently primes the RIG-I pathway, leading to enhanced canonical interferon responses. This is dependent on the transcription factor SREBP1, which binds to the promotor of non-canonical IFNs (e.g. IFN- ω) and drives their expression. The authors recapitulate some of their findings in mouse models, i.e. they demonstrate that in vivo treatment with pitavastatin upregulates certain non-canonical IFNs. Finally, the authors employ scRNAseq to demonstrate that a specific subset of mouse alveolar macrophages expresses RIG-I in response to pitavastatin.

Overall, the manuscript reports some interesting observations. However, many important controls and experimental details are lacking, especially in Fig. 1 to 4. In addition, the overall model and message is not always clear and linear, and the data are not always consistent with the conclusions that are drawn and the overall model. For example, it remains vague to what extent pitavastatin treatment impacts on the IFN-beta promoter and increases canonical IFN production as well. Also, the data regarding the involvement of SREBP1/2 are confusing. Finally, it is unclear why the proposed model would selectively prime RIG-I-dependent responses, but not MDA5-dependent responses. The in vivo treatment of mice with pitavastatin is interesting and worthwhile to report, but the scRNAseq experiments described in Figure 6/7 do not add much value to this manuscript. I have addressed my major concerns here point-by-point. These concerns should be addressed by major revisions in order to make this manuscript suitable for publication in EMBO Reports.

Response:

We are grateful to you for reviewing our manuscript and for your constructive comments. We have carefully revised our manuscript to address all of your concerns as described in detail below.

(Note: The line numbers referred to in this response are based on the original word file, as the line numbers in the automatically generated PDF version may differ.)

Major point 1:

Fig. 1: how does pitavastatin treatment impact on the response to high-molecular weight poly(I:C) or EMCV infection, both of which trigger an MDA5-dependent IFN response? If pitavastatin treatment upregulates non-canonical IFNs, then this should also lead to increased MDA5 expression and responses. I.e. it is unclear why the RIG-I pathway would be selectively primed.

Response:

Thank you for your insightful comments, and we regret that our previous explanation was insufficient. In this study, we focused on RIG-I but not excluded

the statin effects on other pathways. We have included new data (new Fig 1I) demonstrating type I IFN expression following EMCV infection, which triggers an MDA5-dependent response. As shown in new Figure 1I, pitavastatin treatment also enhanced EMCV-induced type I IFN expression. Thus, although we focused on RIG-I signaling in this manuscript, not only the RIG-I but also MDA5 pathways are enhanced by statins. We have improved our explanation in the results (lines 136–153) and have discussed this point in our revised manuscript (lines 451–463).

Major point 2:

Fig. 1C-J: Are the cells that are treated with or without statins equally well transfected with polyI:C or infected with SeV? I am concerned that the loss of cholesterol through statin treatment or HMGCR depletion may impact on membrane composition and the infectivity/transfectability of these cells.

Response:

Thank you for your insightful comment. It is possible that statin treatment may increase the transfection efficiency or viral infectivity due to changes in membrane composition, leading to enhanced type I IFN expression. To address this concern, we measured SeV RNA levels after infection in new Fig 1H. New Fig 1H shows that SeV RNA levels were actually lower in pitavastatin-treated cells compared to non-treated cells, suggesting that pitavastatin does not increase SeV infectivity over the course of infection. This weakens the possibility that pitavastatin enhances viral infectivity. Additionally, as shown in new Fig EV2C and S1A, the knockdown efficiency of siRNA for MAVS is comparable between control and pitavastatin treatment. This also suggests that pitavastatin treatment does not cause significant effects on transfectability. However, we have to carefully consider the effect of statins, and thus we have discussed this point in our revised manuscript (lines 501–513).

Major point 3:

Fig. 1: Is the increased IFN response in pitavastatin treated cells leading to reduced viral replication? This could easily be tested by qPCR analysis of SeV viral transcripts.

Response:

Thank you for your helpful comment. As described above, we have measured the SeV viral RNA levels in infected cells. New Fig 1H showed that pitavastatin treatment reduced viral RNA levels.

Major point 4:

Fig. EV1: how do the authors explain that pitavastatin treatment increases RIG-I and TLR3-induced IFN responses but not cGAMP-induced IFN responses?

Response:

Thank you for your comment. We have addressed this point and have provided new data. New appendix Figure S1C shows that RAW264.7 cells, but not HeLa cells, can induce type I IFN expression in response to cGAMP stimulation under the same experimental condition. Previous studies have reported cell-type-specific responses to cGAS/STING pathway (Ritchie C et al., Annu. Rev. Biochem. 2022. 91: 599-628), suggesting that the lack of type I IFN expression in HeLa cells following cGAMP stimulation may be due to this cell-type specificity. We have added this data in the revised manuscript (Appendix Figure S1C).

Major point 5:

Fig. 2: the effects of cholesterol depletion on basal IFN/ISG expression should be recapitulated with HMGCR knockdown, as an alternative to pitavastatin treatment.

Response:

Thank you for your helpful comment. We have performed HMGCR knockdown assay, as shown in new Figure 2H. Figure 2H shows that HMGCR knockdown also increased IFN- ω 1 and RIG-I expression, similar to the effect of pitavastatin treatment.

Major point 6:

The data in Fig. 2A-C suggests that the Pitavastatin-treated cells have an elevated basal IFN/ISG response. It would be informative to include all ISGs from the microarray analysis in a single heatmap to illustrate this.

Response:

Thank you for your comments. We have added a new heatmap panel in new Fig EV2A, in which the expression of all ISGs is shown in a single heatmap.

Major point 7:

Fig. 2D-E: as above. Which of these targets are ISGs? Please indicate this in the figure, e.g. by color coding.

Response:

Thank you for your comment. We have indicated the ISGs in red in Figure 2D and new Figure 2G. Although many ISGs were upregulated, some of the ISGs were not. We have discussed possible reasons in the revised manuscript (lines 483–486).

Major point 8:

Is the increase in the basal IFN/ISG response dependent on signaling via RLRs or cGAS? This could easily be addressed by repeating the experiment from Fig. 2D in cells depleted of MAVS or STING by siRNAs (or, as a positive control, siIFNAR).

Response:

Thank you for your helpful comment. We have performed knockdown assays. New Fig EV2C shows that siRNA for MAVS failed to attenuate pitavastatin-induced RIG-I expression in uninfected cells. These data suggest basal IFN/ISG response is independent of RLRs. We have mentioned this point (lines 202–206). As described above, HeLa cells used for this study do not induce type I IFNs in response to stimulation of the cGAS/STING pathway, which also weakens the possibility that cGAS is involved in the basal response.

Major point 9:

The authors conclude that pitavastatin treatment induces non-canonical IFNs, but not canonical IFNs such as IFN-beta, at steady state. This conclusion is based on the microarray data in Fig. 2B. To convincingly make this conclusion, IFNA and IFNB levels should also be measured by qPCR in Fig. 2D and directly compared to IFNL3 and IFNW.

Response:

Thank you for your insightful comment. We have directly compared the expression of IFNW with IFNA and IFNL3 in new Figure 2E. The figure showed that pitavastatin treatment induced noncanonical type I IFN but not canonical type I IFNs.

Major point 10:

Fig. 3H/I: the authors find that the effect of pitavastatin on basal ISG levels is IFNAR2 dependent at early time points but not at late time points. This may be explained by the production of type III IFNs (Fig. 2E: increased expression of IFN-L3) that signal through the type III IFN receptor (not IFNAR). The authors can test this by knocking out or knocking down the type III IFN receptor or, alternatively, by using the JAK inhibitor ruxolitinib, which blocks both type I and type III IFN signaling.

Response:

Thank you for your helpful comment. We have performed a knockdown assay. New Fig 3J shows that knockdown of IFNLR1 does not reduce the RIG-I expression following pitavastatin treatment in IFNAR2 KO cells. These data suggest that type III IFNs are unlikely to be involved. Therefore, we think that other pathways may be responsible for inducing RIG-I expression. We have discussed this point in the revised manuscript (lines 494-501).

Major point 11:

Fig. EV3A: 4 hrs of polyI:C treatment should be sufficient to increase IFNB transcripts in mock or BSA treated cells (as the authors also show in EV4F). Why is that not visible in this experiment?

Response:

Thank you for your comment. This is because the poly I:C-induced IFNB transcript level in IFNW1-treated cells was extremely high at 4 hours post-stimulation, which made the IFNB transcript level in mock or BSA-treated cells not visible. To clarify this, we have adjusted Fig EV3A by splitting the Y-axis to make the lower IFNB1 transcript level visible (Fig EV3A).

Major point 12:

Fig. 4: Given the set of tools and reagents that the authors have to their disposal, they could directly test whether pitavastatin treatment increases the expression of the IFN-w reporter.

Response:

Thank you for your helpful comment. We have investigated whether the pitavastatin treatment increase the expression of the IFN-W promoter, and found that pitavastatin treatment could induce the expression of IFN-W promoter (new Figure EV4J).

Major point 13:

Fig. 4A/B: why is the SREBP1c-N construct not causing a significant increase in IFN-b promoter reporter activity (panel A), while it strongly induces IFN-b mRNA expression (panel B). In the text the authors conclude that SREBP1 regulates non-canonical IFNs but not canonical IFNs (IFN-b). This conclusion cannot be drawn based on panel 4B.

Response:

Thank you for your insightful comment. As shown in new Appendix Figure S4, the IFN-B promoter contains an E-box sequence, which is a known binding site of SREBP1. Accordingly, overexpression of constitutive active SREBP1c-N can directly increase IFN-B expression, as observed in old Fig 4B (new Fig 4C). In Fig 4A, although there is a trend toward increased promoter activity of IFN-B with SREBP1c-N overexpression, this increase was not statistically significant. This discrepancy might be due to differences in the sensitivity between the promoter

reporter assay and mRNA expression analysis or the presence of additional regulatory elements in the genomic context that are not captured in the reporter construct. These results suggest that SREBP1 has the potential to induce IFN-B expression when over-expressed. However, in our in vitro studies, pitavastatin treatment did not induce IFN-B expression, as shown in new Fig 2E. This indicates that under more physiological conditions, SREBP1 activation by pitavastatin is not sufficient to increase IFN-B expression, possibly due to the presence of regulatory mechanisms that suppress SREBP1-induced IFN-B expression. For instance, epigenetic regulation or the involvement of co-receptors might limit IFN-B induction in response to pitavastatin. However, considering the cell-type specific effects of pitavastatin, it is still possible that not only noncanonical type I IFNs but also canonical type I IFNs, including IFN-B, are induced by pitavastatin in specific cells in vivo. We have discussed this possibility in the revised manuscript (446–450).

Major point 14:

Fig. 4C: the FLAG-RIG-I-CARD construct is not visible on the FLAG immunoblot, suggesting it is not well expressed. In addition, expression of the RIG-I CARD domain should strongly induce an IFN response and upregulate ISGs, which is not evident from the RIG-I and MDA5 immunoblot.

Response:

Thank you for your helpful comment. As you pointed out, the RIG-I CARDs fragment was not adequately expressed in the old Fig 4C. We have replaced the data with new results where the RIG-I CARDs fragment is adequately expressed, leading to the increase in the RIG-I and MDA5 protein levels (new Fig 4D).

Major point 15:

Fig. 4D: Validation experiments of the SREBP1/2 KO cells should be included. In addition, a rescue experiment should be performed in the SREBP1 KO cell line to demonstrate that the observed effects are truly due to the loss of SREBP1.

Response:

Thank you for your helpful comment. We have provided the data showing the

loss of the SREBP1 and 2 protein expression in the KO cells (new Figure S3A). Moreover, we have investigated whether ectopically expressed SREBP1 can rescue the defect of SREBP1 KO cells. New Figure EV4G show that ectopically expressed full-length SREBP1a and c rescue the pitavastatin effects in SREBP1 KO cells. These data further support our conclusion that SREBP1 is responsible for the effect by cholesterol restriction.

Major point 16:

Fig. 4F: the IFN-beta promoter should be included as a control in these experiments to support the conclusion that SREBP1 localizes to the promoter of non-canonical IFNs but not IFN-beta.

Response:

Thank you for your comments. We have included the data of the IFN- β promoter. Although there is an E-box in the -351 ~ -257 region, we could not detect the significant binding of SREBPs by the ChIP assay (new Figure 4G). This data also supports our conclusion that SREBP1 localized to the non-canonical type I IFN. However, it is still possible that the transcription factor binds to other promoter regions within type I IFN genes. We have discussed this limitation in the revised manuscript (lines 509–513).

Major point 17:

Fig. 5D: were other (non-canonical) IFNs tested here? This would be important to validate the hypothesis that pitavastatin impacts on IFNB expression through upregulating non-canonical IFNs.

Response:

Thank you for your insightful comments. We have assessed noncanonical interferon expression in Fig 5D; however, we did not observe an increase in noncanonical interferon expression in the whole lung tissue following pitavastatin treatment. Considering that only specific cell types responded to pitavastatin treatment, it is possible that the increase in noncanonical type I IFNs was not detectable in the whole lung due to their expression being limited to a small subset of cells, such as specific alveolar macrophages. We have explained this

point in detail in our revised manuscript (lines 327-329).

Major point 18:

Fig. 7E: other transcripts (non-canonical IFNs) should be measured as well, as done in Fig. 7B.

Response:

Thank you for your helpful comments. We have investigated the expression of noncanonical type I IFNs in CD140a+ alveolar macrophages, and found that IFN- ζ expression was induced by pitavastatin treatment (new Fig. 7E). This data also supports our model that noncanonical type I IFNs were induced by pitavastatin treatment, leading to RIG-I expression.

Minor point 1:

The materials and methods section lacks details on the transfection procedure of different PAMPs (e.g. polyI:C, cGAMP). Details regarding the concentration, the commercial source, and the transfection reagent should be included.

Response:

Thank you for your helpful suggestions. We have described the details of the transfection procedure of different PAMPs, including the concentration, commercial source, and transfection reagents in the methods and each figure legend (lines 540–549 and each figure legend).

Minor point 2:

Page 6: "...Riplet protein, a coreceptor of RIG-I.". While Riplet is clearly important for RIG-I signaling, I believe the word co-receptor is inappropriate, as it implies that Riplet is binding to an RNA ligand as well, alongside RIG-I. Please rephrase this sentence.

Response:

Thank you for your comment. We have replaced "co-receptor" with "a positive regulator" (line 109).

Minor point 3:

The name of the compound library should be mentioned in the text. Is this the Sigma Lopac library?

Response:

The library name is “Validated Compound Library” provided by Drug Discovery Initiative at the University of Tokyo. We have added this information in the method (line 552).

Minor point 4:

Page 6: it is unclear in the text if the second screening with the p125 luc plasmid was done using the same compound library, or with the 22 hits from screen 1 only.

Response:

We regret that our previous explanation was insufficient and confusing. We have improved the explanation of the screening (lines 112–120) and also improved Fig 1A to clarify that the second screening was performed for the 140 hits from the 1st screening.

Minor point 5:

Fig. 1B: indicate in the figure that this is an IFN β -Luc assay.

Response:

We have added a label indicating “p125-luc (IFN- β promoter)” in Fig 1B.

Minor point 6:

Fig. 1E: indicate in the legend that cerivastatin was used in this experiment. Indicate in the ligand what concentrations of statins were used.

Response:

We have described the concentration of each statin in Figure 1E legend.

Minor point 7:

Fig. 1F: X-axis indicates that this is flu infection, while the text and legend refers to Sendai virus infection.

Response:

Thank you for your comments. We have used Flu in Fig 1F and SeV in Fig 1G, and thus I have corrected our explanation for Figure 1F (lines 127–130, Figure 1F and G legends).

Minor point 8:

Fig. 1E and 1F: is IFN-beta really completely undetectable after 24 or even 48 hrs of SeV infection? It is unclear why no IFN-beta is produced at all in mock-treated SeV infected cells (Fig. 1E) or in cells infected for 1 day with SeV (Fig. 1F).

Response:

We regret that our explanation for Figs 1E and F was insufficient. We have replaced old Fig 1E with new data because there were no mock infected data in old Fig 1E. New Fig 1E includes the data of mock infection as a control and also contains the sub-graph in which the Y-axis is enlarged to be able to see the increase of IFN- β mRNA expression. In Figure 1F, flu infection did not induce the IFN- β protein in the culture sup. in this experimental condition. We have added this explanation for the main text (lines 130–131).

Minor point 9:

Fig. EV1G: was flu or Sendai virus used here?

Response:

We regret that our explanation was confusing. Flu is used in old Figure EV1G (new Figure EV1D).

Minor point 10:

Referral to Fig. EV1G is lacking in the text.

Response:

Thank you for your comment. We have added the explanation for old Fig EV1G (new Fig EV1D) (lines 130) and re-arranged the order of each panel to adjust to the explanation.

Minor point 11:

In Fig. 3 an IFNAR2 KO cell line is used. I assume this KO cell line was properly validated (e.g. through measurement of IFNAR expression by immunoblotting or flow cytometry, sequencing, or by measuring the response to recombinant IFN). Please include validation data in the supplement.

Response:

Thank you for your comment. To confirm the IFNAR2 KO, we investigated whether recombination type I IFN failed to induce ISG expression in IFNAR2 KO cells. We have added the validation data showing that IFNAR2 KO cells fail to induce the expression of RIG-I and IP-10 mRNA in response to type I IFN stimulation (new Appendix Figure S2A).

Minor point 12:

Fig. 4B: right panel (IFN-w): is the 2a-N-mediated increase in IFN-w expression truly not significant? (highest dose of plasmid). Please double check the statistical analysis.

Response:

Thank you for your helpful comment. In old Fig 4B (new Fig 4C), we have performed One-way ANOVA comparing each data with control and confirmed that there are no significant differences between control and 2a-N samples. We have added the information of the statistical analyses in Fig 4B (new Fig 4C) figure legends.

Minor point 13:

I would suggest moving supplemental Fig. EV4A to the main figures.

Response:

Thank you for your suggestion. We have moved old Fig EV4A to the main figure (new Fig 4B).

Referee #3:

The manuscript presents intriguing observations linking the inhibition of the cholesterol synthesis pathway mediated by statins with the induction of certain IFN-inducible genes. The authors describe the activation of some type I IFNs dependent on statin treatment, including IFN omega, certain types of IFN alpha, and IFN lambda. Additionally, they report that statin-mediated induction of certain ISGs is partially dependent on type I IFN signaling.

The manuscript also provides interesting data related to the alternative induction of certain IFNs dependent on SREBPs and SRE binding sites in the promoters of some IFN production genes. Statin treatment or similar mechanisms could induce the expression of these interferons and stimulate IFN-inducible genes. Statins (PMID: 34321273) and HDL receptor inhibitors SR-BI (BTX-I and ITX5061) have previously been described to decrease ISG expression through effects on the IFN signaling cascade (PMID: 27622065).

Response:

We are grateful to you for reviewing our manuscript. We have addressed all of your concerns as described below.

(Note: The line numbers referred to in this response are based on the original word file, as the line numbers in the automatically generated PDF version may differ.)

Main Concerns:

1. *Title: The title does not reflect the discovery. It should mention that statins can stimulate the production of IFNs other than IFN-beta through a mechanism dependent on SREBP activation. It is not RIG-I dependent only, as activation of TLR3 can achieve the same effect.*

Response:

Thank you for the helpful comments. We have revised the title according to the reviewer's suggestion. Additionally, we have emphasized the importance of SREBP1 in the abstract (line 35), because SREBP1 is included in the title. Due to the limitation in the word count for the title, we cannot include all the details; however, we believe that the revised title appropriately reflects our discovery.

2. *Abstract: The abstract does not reflect the effect of statins in inhibiting type I IFN signaling.*

Response:

We regret that we did not include the previous important studies in the previous version of our manuscript. We have revised the abstract to reflect the effect of statins on inhibiting type I IFN signaling. Specifically, we have included a description of the pleiotropic effects of statins on type I IFN responses in the abstract (lines 26-27). Additionally, in the introduction (lines 92-95), we have cited previous studies that show the inhibitory effect of statins on type I IFNs.

3. *While the study focuses on the effect of statins on type I IFN induction, it also shows that statins induce IFNs lambda.*

Response:

Thank you for your comment. We have added several data in our revised manuscript to address this comment. Although type III IFN expression was enhanced by pitavastatin in viral infected cells (Fig 2G), new Fig 2E showed that type III IFN was not induced by pitavastatin treatment in uninfected cells under the condition that induces IFN- ω 1. In addition, we performed a knockdown

assay for the type III IFN receptor. New Fig 3J shows that the knockdown of type III IFN receptor does not reduce the RIG-I expression induced by pitavastatin treatment even in IFNAR2 KO cells. These data weaken the possibility that type III IFN plays a major role in cholesterol restriction-mediated priming of the RIG-I pathway. However, it is still possible that type III IFNs is important in specific cell types in vivo. Thus, we have discussed the potential roles of type III IFNs (lines 494–501).

4. The authors describe differential induction of various interferons dependent on SRE promoter elements but do not describe the potential position of these elements in all IFN promoters, including type III IFNs. The use of "non-canonical type-I IFN" should specify the IFNs. The canonical vs. non-canonical concept is not widely accepted.

Response

Thank you for your helpful comment. We have described the potential positions of SRE promoter elements of type I and type III IFN promoters in Appendix Figure S4. Since some of the E-box do not function as binding sites for SREBP1 as shown in Fig 4G, we have mentioned this point in the revised manuscript to avoid any confusion (lines 286–290). Moreover, we have specified the noncanonical type I IFNs in the introduction to clarify this concept (lines 73–77).

5. The proposed mechanism suggests that statins can induce SREBPs, leading to cytokine production that stimulates the type I IFN signaling cascade. The possibility that type III IFNs production might induce the IFNLR-dependent signaling pathway, which could explain some experiments, is not discussed. The authors should test the effect of statins in IFNLR KO cells and STAT1 KO cells. They also describe the effect on alveolar macrophages, which depend on IFN-gamma. Do statins affect IFN-gamma signaling (IFNGR KO cells)?

Response:

Thank you for your insightful comment. We have tested the effect of statins in IFNLR knockdown and IFNAR KO cells in new Fig 3J. New Fig 3J shows that knockdown of IFNLR does not attenuate the effect of statins, which weaken the

possible involvement of type III IFNs. Regarding type II IFN, it is well-known to be mainly produced by NK cells and T cells, and thus it is unlikely that type II IFNs plays a major role under our experimental conditions. Indeed, our new data shows that IFN- γ is not induced in alveolar macrophages in response to statin treatment (Fig 7B rightmost panel). To properly assess the effect of statins on type II IFNs, NK cells and T cells would need to be used. We believe that this assessment is beyond the scope of the current study. We have discussed the possible involvement of type II and III IFNs in the revised manuscript (lines 494–497).

6. Potential Effects: The proposed mechanism implies that SREBPs-dependent IFN induction produces certain IFNs that would activate the IFN signaling cascade, potentially increasing ISGs that could further potentiate the IFN induction pathway (types I, II, and III). The manuscript partially reflects this potential effect and should explore all possibilities. A graphical abstract would clarify these concepts.

Response:

Thank you for your helpful comment. As described above, we have explored possible involvements of type II and III IFNs in the effect of statins under our experimental conditions, and thus we have developed a graphical abstract including noncanonical type I IFNs to clarify the concept that SREBP1-mediated expression of noncanonical type I IFNs leads to enhanced type I IFN response following viral infection.

7. Figure 7: The figure describes the effect of statins on alveolar macrophages, showing statin treatment affects the production of various type I IFNs. It should include the effect on IFN lambda and gamma induction.

Response:

We have investigated the effects on IFN-lambda and gamma induction. New Fig 7B showed that pitavastatin treatment did not increase the expression of IFN- λ 2/3 and IFN- γ .

8. *The effect of statins on CD140+ macrophages is poorly presented. The authors should repeat the experiment with a larger sample size and analyze the effect more exhaustively, including more genes, both ISGs and IFNs.*

Response:

Thank you for your helpful comments. We have repeated the experiment with a larger sample size to investigate the expression of type I IFNs in CD140a+ alveolar macrophages (new Figure 7E and F). We found that IFN- ζ expression was induced by pitavastatin treatment, but not IFN- ϵ (new Fig. 7E). This data also supports our model that noncanonical type I IFNs were induced by pitavastatin treatment, leading to RIG-I expression.

Minor Concerns:

1. *Introduction: The introduction does not discuss the effects of statins on the type I IFN signaling cascade, despite prior evidence (some references indicated before). The authors should consider expanding this section.*

Response:

Thank you for your helpful comment. We have cited previous papers and added a description of the effect of statins on type I IFN signaling cascade in the introduction (lines 92–95).

2. *Results: The authors describe the effect of statins not only on the RLR pathway but also on TLR-3. Can statins affect TLR4, TLR7, and TLR8/9 signaling?*

Response:

Thank you for your helpful comment. As shown in Figure 2, the IRF7 expression was induced by pitavastatin treatment. We have added new data showing that pitavastatin enhances the IFN- β expression in response to TLR4 stimulation (new Appendix Figure S1E). Since IRF7 is an important transcription factor downstream of TLR7, 8, and 9, it is likely that pitavastatin treatment also affected TLR7, 8, and 9 signaling. We have discussed the potential involvement of these

TLRs in the revised manuscript (lines 456–463).

3. *Reporter Assays: The promoters of IFN- β used in reporter assays are poorly described in the materials and methods section. Define the length of the promoters used.*

Response:

We have added the information of p125 luc (IFN- β reporter) and cite the previous paper in the method (line 575).

4. *Plasmids: The materials and methods section does not describe the plasmids used in reporter assays for other IFNs or RIG-I in the manuscript. Include descriptions of these promoters and their lengths.*

Response:

We have described the constructs of the reporter plasmids of IFN- β , IFN- ω , and RIG-I in the Methods (lines 575–585).

5. *Overexpression Plasmids: The manuscript mentions various plasmids for protein overexpression that are not detailed in the materials and methods. Include these details.*

Response:

We have described the details of the plasmids in the Methods (lines 575–585).

6. *Virus details: The amount of virus used in some infection experiments is not defined. This should be described in the materials and methods section and in relevant results sections, including virus growth and titration.*

Response:

We have added the details of the preparation of virus and viral infection assay in the methods (lines 540–543), and also described the amounts of the viruses

used for each experiment in each figure legend.

7. Discussion: The discussion should better explain why the effect of statins is not universal (i.e., in A549 cells). Specify potential reasons and test primary cells, not just cell lines, such as in vitro differentiated macrophages.

Response:

Thank you for your insightful comment. We have expanded the discussion to specify potential reasons why the effect of statins is not universal among different cells, such as differences in epigenetic regulation and varying levels of tolerance to statins (lines 488–491). These mechanisms may influence the expression of genes involved in the statin response or after cellular metabolism. Moreover, as you suggested, we have tested the effect of pitavastatin on in vitro differentiated macrophages in new Figures EV5A and B. Our results show that pitavastatin increases the expression of noncanonical type I IFNs and enhances IFN- β expression following poly I:C stimulation in bone-marrow-derived macrophages.

8. Type III and II IFNs: Provide evidence of statins' effects on type III and type II IFN signaling (Figure 3).

Response:

Thank you for your comment. As described above, we have tested the possible involvement of type III IFNs in new Fig 3J, and it shows that type III IFN receptor knockdown does not attenuate the effect of statins. This weakens the possibility that type III plays a major role in the effect of statins under our experimental condition. Regarding to type II IFNs, as shown in Figure 7, macrophages do not produce type II IFNs, and thus it is unlikely that type II IFNs play a crucial role under our experimental condition. We believe that these additional data and discussion further strengthen our manuscript.

9. SREBPs Analysis: Better describe the differential induction by various SREBPs analyzed, including their DNA binding motifs.

Response:

Thank you for your helpful comment. We have further described the difference between SREBP1 and SREBP2 (lines 433–437).

10. *Promoter Analysis: Present an analysis of different SREs in the promoters of various IFNs (Figure 4).*

Response:

We have presented the position of SRE elements and E-boxes in the promoter regions of type I IFNs as well as type III IFNs and IRF7 (Appendix Figure S4).

11. *Administration of Treatments: Treatment administration is only described as an injection, even for poly IC and lung treatments. Does this refer to intranasal administration? Review the document and specify whether the injection is intranasal, intravenous, intraperitoneal, retro-orbital, or otherwise. Define the administration methods and times until analysis in the text and all figure legends.*

Response:

Thank you for your comment. As described in the main text and Figure legends, we utilized the intraperitoneal injection method in this study. We have improved our explanation to avoid any confusion (lines 303–307).

12. *Cluster Numbering: Is it possible to order the numbering of clusters sequentially based on the possible evolution of the macrophage (Figure 6)?*

Response:

Thank you for your comment. We understand the importance of sequentially ordering the cluster numbering based on the possible evolution of macrophages. However, due to technical limitations, we were unable to re-order the numbering. As a result, the numbering remains unchanged in the revised manuscript.

Dear Prof. Oshiumi

Thank you for the submission of your revised manuscript to our editorial offices. I have now received the reports from the referees that I asked to re-evaluate the study, you will find below. As you will see, all referees now support publication of the study in EMBO reports. However, referees #2 and #3 have remaining concerns and suggestions to improve the study, I ask you to address in a final revised manuscript. Please also provide a further p-b-p-response addressing the remaining points.

- Please have your final manuscript carefully proofread by a native speaker (see also the report by referee #2).
- Please order the manuscript sections like this, using these names:
Title page - Abstract - Keywords - Introduction - Results - Discussion - Methods - Data availability section - Acknowledgements (including funding information) - Disclosure and Competing Interests Statement - References - Figure legends - Expanded View Figure legends
- Please use our reference format:
<http://www.embopress.org/page/journal/14693178/authorguide#referencesformat>
- Please make sure that the number "n" for how many independent experiments were performed, their nature (biological versus technical replicates), the bars and error bars (e.g. SEM, SD) and the test used to calculate p-values is indicated in the respective figure legends (main, EV and Appendix figures). Please also check that all the p-values are explained in the legend, and that these fit to those shown in the figure. Please provide statistical testing where applicable. Please avoid the phrase 'independent experiment', but clearly state if these were biological or technical replicates. The standard sentence that is present in all the legends of the manuscript - 'The data represent at least two experiments with similar results (means {plus minus} SDs, n = 3 biological replicates) - is rather unclear. Please indicate for each panel how many replicates were done (where applicable). Please also indicate (e.g. with n.s.) if testing was performed, but the differences are not significant. In case n=2, please show the data as separate datapoints without error bars and statistics. See also:
<http://www.embopress.org/page/journal/14693178/authorguide#statisticalanalysis>
- If n<5, please show single datapoints for diagrams. It seems that presently many diagrams (main, EV and Appendix figures) have no or only partial stats, or the 'ns' is missing. Please check. Moreover:
 - Please note that the legend figure EV 4e, g is mislabeled as figure EV 4e-g in the statistical test section of the legend in the manuscript. This needs to be rectified.
 - Please note that the exact p values are not provided in the legends of figures 1c-e, g-i, k-l; 2d-e, g-h; 3g-k; 4a-c, e-g; 5a-b, d-f; 7b, d-f; EV 1a-d, g, i-k; EV 2b-d; EV 3; EV 4a-e, g, j; EV 5a-c.
 - Please note that information related to n is missing in the legends of figures 4c; 6d, h; 7c.
 - Although 'n' is provided, please describe the nature of entity for 'n' in the legends of figures EV 1a-m; EV 2b-d; EV 3; EV 4a-g, j.
- Please add to each legend (main, EV figures, where applicable) a 'Data Information' section explaining the statistics used or providing information regarding replicates and scales. See:
<https://www.embopress.org/page/journal/14693178/authorguide#figureformat>
- Please add the primer information provided in the Appendix Table S1 to the Reagents and tools table and update the related callouts (e.g. 'see reagents and tools table'). Then, please remove the table from the Appendix.
- Please add a title page to the Appendix (Appendix for: "...", but without author names and affiliations) with a table of contents with page numbers. Please also move the legends below each Appendix figure.
- Please remove now the referee tokens from Data Availability section (DAS) and make sure the datasets are public latest upon online publication of the study.
- Thank you for providing the requested source data. Please upload this as one folder per figure (with all files for one figure in one folder and ZIPed together).

In addition, I would need from you uploaded separately:

Best,

Referee #1:

I think the authors have done a significant amount of work to address the comments from all reviewers. These changes have significantly improved the quality of the manuscript and impact of the findings. Based on the response and changes made, I think the paper can now be published.

Referee #2:

The authors have addressed all my comments and the manuscript has improved by the inclusion of additional experiments. In addition, the authors nuance their conclusions better and discuss the limitations of their work in the discussion. The overall observation that cholesterol fluctuations increase both the basal and ligand-induced IFN response is interesting. However, the underlying mechanism still remains rather puzzling and incompletely understood. Nonetheless, the authors have done their best to obtain some insights into a potential mechanism. There are multiple grammar mistakes and sometimes the text does not flow well; I suggest that the text is carefully reviewed by a text editor prior to publication.

A few minor points:

- Fig. 2F: I am surprised that it is possible to detect IFN- ω by western blot (Fig. 2F). Can the authors add to the Materials and Methods how this was done? Was Brefeldin A used to block protein secretion?
- Fig. 3L: the label 'pitavastatin' is missing
- Line 267: SREBP1 (1c-N) also seems to have a (modest) effect on IFN- β expression (fig. 4C). This observation is largely ignored in the Results section, although it is briefly elaborated on in the Discussion. Please discuss this observation briefly in the Results section as well.
- Line 510: typo (faction instead of factor)
- The following paper should be referenced, as there may be some links with the current manuscript. Could statins perhaps disturb the actin cytoskeleton and thereby prime RIG-I activation? <https://pubmed.ncbi.nlm.nih.gov/36113429/> / doi: 10.1016/j.cell.2022.08.011.

Referee #3:

Thank you for your effort in addressing these issues. Below are some remaining points that need further clarification to enhance the understanding of statin mechanisms within the IFN system:

Main Concerns

Main Concerns 5 & 7: While it's true that macrophages don't produce IFN- γ , they can respond to external (in vivo) IFN- γ , and statins may influence this response. Could IFN- γ signaling be affected by GAS genes that contain SRE elements? Please prove the response of macrophages to external IFN- γ in the presence of statins and discuss how the mechanisms proposed in the article align with IFN- γ signaling in the context of all IFNs.

Main Concern 6: The graphical abstract illustrating the proposed mechanisms appears to be missing. Could you indicate where it has been placed?

Minor Points

Figure 7d: Label the right bar as "CD140 PE."

Figure EV4(H) and Appendix Figure 4: Ensure consistency in the SRE element counts for the IFN- ω 1 promoter across figures.

Figure Legends: Specify that the promoter analysis and SRE box positions refer to human IFN genes.

Thank you again for your revisions, which are helping clarify these mechanisms in the context of IFN signaling.

Point-by-point response:

Referee #1:

I think the authors have done a significant amount of work to address the comments from all reviewers. These changes have significantly improved the quality of the manuscript and impact of the findings. Based on the response and changes made, I think the paper can now be published.

Response:

We sincerely thank you for your positive evaluation. In this revision, we have specifically addressed the comments from Reviewer 2 and 3. All changes made in this revision are marked with yellow highlights for clarity.

Referee #2:

The authors have addressed all my comments and the manuscript has improved by the inclusion of additional experiments. In addition, the authors nuance their conclusions better and discuss the limitations of their work in the discussion. The overall observation that cholesterol fluctuations increase both the basal and ligand-induced IFN response is interesting. However, the underlying mechanism still remains rather puzzling and incompletely understood. Nonetheless, the authors have done their best to obtain some insights into a potential mechanism. There are multiple grammar mistakes and sometimes the text does not flow well; I suggest that the text is carefully reviewed by a text editor prior to publication.

Response:

We sincerely thank you for throughout review and constructive comments. We have addressed all remaining concerns as described below. Additionally, the manuscript has been carefully proofread by a professional English editing service. All changes made in this revision are marked with yellow highlights for clarity.

A few minor points:

Comment 1: Fig. 2F: I am surprised that it is possible to detect IFN- ω by western blot (Fig. 2F). Can the authors add to the Materials and Methods how this was done? Was

Brefeldin A used to block protein secretion?

Response:

Thank you for your comments. To detect IFN- ω protein by western blotting, we did not use Brefeldin A. We have now included a detailed description of the method for detecting IFN- ω protein in the Materials and Methods (lines 681–684). The question of why IFN- ω remains associated with cells is indeed an important issue. Several possibilities could explain this observation, such as secreted IFN- ω binding to the cell surface via the type I IFN receptor, or the existence of an intracellular signaling pathway. However, elucidating this mechanism falls beyond the scope of the current study. We plan to address this intriguing question in our future research.

Comment 2: Fig. 3L: the label 'pitavastatin' is missing

Response: We have added the label in Fig 3L.

Comment 3: Line 267: SREBP1 (1c-N) also seems to have a (modest) effect on IFN-beta expression (fig. 4C). This observation is largely ignored in the Results section, although it is briefly elaborated on in the Discussion. Please discuss this observation briefly in the Results section as well.

Response:

Thank you for your insightful comment. We have added a description and a brief discussion of the effect of SREBP1 on IFN- β expression and have discussed (lines 279–282), as you suggested.

Comment 4: Line 510: typo (faction instead of factor)

Response: Thank you for your comment. We have corrected the typo.

Comment 5: The following paper should be referenced, as there may be some links with the current manuscript. Could statins perhaps disturb the actin cytoskeleton and thereby

prime RIG-I activation? <https://pubmed.ncbi.nlm.nih.gov/36113429/> / doi: 10.1016/j.cell.2022.08.011.

Response:

Thank you for your intriguing comment. We have now cited the manuscript and included a discussion on the possible involvement of actin cytoskeleton in the association between cholesterol metabolism and antiviral innate immune responses (lines 545-552).

Referee #3:

Thank you for your effort in addressing these issues. Below are some remaining points that need further clarification to enhance the understanding of statin mechanisms within the IFN system:

Response:

We are grateful to you for reviewing our manuscript. We have addressed all comments as described below. All changes made in this revision are marked with yellow highlights for clarity.

Main Concerns

Comment 1: Main Concerns 5 & 7: While it's true that macrophages don't produce IFN- γ , they can respond to external (in vivo) IFN- γ , and statins may influence this response. Could IFN- γ signaling be affected by GAS genes that contain SRE elements? Please prove the response of macrophages to external IFN- γ in the presence of statins and discuss how the mechanisms proposed in the article align with IFN- γ signaling in the context of all IFNs.

Response:

Thank you for your insightful comment. As the reviewer suggested, macrophages respond to IFN- γ . In the new Appendix Figure S5, we examined whether pitavastatin treatment affects IFN- γ -mediated expression of genes that are also regulated by SREBP1 in macrophages. We found that IFN- γ -induced IP-10 expression was

moderately enhanced by pitavastatin treatment, as you suggested (lines 306–316). Additionally, we have discussed the association of antiviral innate immune responses with cholesterol synthesis restriction in the context of all IFNs in the revised manuscript (lines 525–529).

Comment 2: Main Concern 6: The graphical abstract illustrating the proposed mechanisms appears to be missing. Could you indicate where it has been placed?

Response:

We sincerely apologize for this oversight. During the previous revision, we inadvertently upload the graphical abstract as a visual synopsis image file, which may not have been accessible to the reviewers. To address this concern, we have now included the graphical abstract as Appendix Figure S8 in revised supplemental materials.

Minor Points

Minor comment 1: Figure 7d: Label the right bar as "CD140 PE."

Response:

Thank you for your comment. We have labeled the right bar in Figure 7D as CD140a PE.

Minor comment 2: Figure EV4(H) and Appendix Figure 4: Ensure consistency in the SRE element counts for the IFN-W1 promoter across figures.

Response:

Thank you for your comment. In old Figure EV4 (H), the region between -752 and -290 was omitted, which contains one SREE. This omission led to an apparent discrepancy in the SRE counts. In the revised Figure EV4 (H), we have included this region, ensuring that the SRE counts are consistent across both figures.

Minor comment 3: Figure Legends: Specify that the promoter analysis and SRE box positions refer to human IFN genes.

Response:

Thank you for your comments. We have clarified in the figure legends of Figure EV4H and appendix Figure S4 that the promoter analysis and SRE box positions refer to human IFN genes.

Dear Prof. Oshiumi,

Thank you for the submission of your further revised manuscript to our editorial offices. I looked through this now and your further p-b-p-response and consider the remaining points of the referees as adequately addressed.

However, before I can proceed with formal acceptance, I have this further editorial requests I ask you to address in a final revised manuscript:

It seems, this point of my previous decision letter has not been fully addressed:

- Please make sure that the number "n" for how many independent experiments were performed, their nature (biological versus technical replicates), the bars and error bars (e.g. SEM, SD) and the test used to calculate p-values is indicated in the respective figure legends (main, EV and Appendix figures). Please also check that all the p-values are explained in the legend, and that these fit to those shown in the figure. Please provide statistical testing where applicable. Please avoid the phrase 'independent experiment', but clearly state if these were biological or technical replicates. The standard sentence that is present in all the legends of the manuscript - 'The data represent at least two experiments with similar results (means {plus minus} SDs, n = 3 biological replicates) - is rather unclear. Please indicate for each panel how many replicates where done (where applicable). Please also indicate (e.g. with n.s.) if testing was performed, but the differences are not significant. In case n=2, please show the data as separate datapoints without error bars and statistics. See also:
<http://www.embopress.org/page/journal/14693178/authorguide#statisticalanalysis>

Still, most diagrams show no or only partial statistics, have the p-values not indicated or ns (i.e. 1B,C,D,F,G,L; 2D,G,H,I; 3A-F,H; 4A,B,C,E,F; 7E, all EV figures and also S1, S2 and S3 - Fig 5 is fine). Please add this information. Moreover, please clearly indicate what has been compared and to which comparison the p-value or ns belongs (as in the panels of Fig. 5). Please also make sur the statistical test and the nature of n is indicated in the respective legends.

Best,

Point-by-point response:**Comment:**

It seems, this point of my previous decision letter has not been fully addressed:

- Please make sure that the number "n" for how many independent experiments were performed, their nature (biological versus technical replicates), the bars and error bars (e.g. SEM, SD) and the test used to calculate p-values is indicated in the respective figure legends (main, EV and Appendix figures). Please also check that all the p-values are explained in the legend, and that these fit to those shown in the figure. Please provide statistical testing where applicable. Please avoid the phrase 'independent experiment', but clearly state if these were biological or technical replicates. The standard sentence that is present in all the legends of the manuscript - 'The data represent at least two experiments with similar results (means {plus minus} SDs, n = 3 biological replicates) - is rather unclear. Please indicate for each panel how many replicates were done (where applicable). Please also indicate (e.g. with n.s.) if testing was performed, but the differences are not significant. In case n=2, please show the data as separate datapoints without error bars and statistics. See also:

<http://www.embopress.org/page/journal/14693178/authorguide#statisticalanalysis>

Still, most diagrams show no or only partial statistics, have the p-values not indicated or ns (i.e. 1B,C,D,F,G,L; 2D,G,H,I; 3A-F,H; 4A,B,C,E,F; 7E, all EV figures and also S1, S2 and S3 - Fig 5 is fine). Please add this information. Moreover, please clearly indicate what has been compared and to which comparison the p-value or ns belongs (as in the panels of Fig. 5). Please also make sure the statistical test and the nature of n is indicated in the respective legends.

Response:

Thank you for your detailed editorial requests. We have carefully addressed all the comments and revised the manuscript accordingly. In the revised manuscript, we have ensured that all diagrams now include complete statistical information, including p-value and NS, where applicable. We have also clearly indicated the comparison pairs and specified which comparison each p-values or NS belongs. Additionally, the statistical test and the nature of n have been described in the respective figure legends. All changes made in this revision are shown in blue text for clarity. Below is our point-by-point response to

each comment.

In Fig 1B, we investigated the dose dependency, and therefore we have described the EC50 value and its statistical calculation method in the figure legend.

In Fig 1C, we have indicated p-values in the figure and described the statistical test in the figure legend. As shown in the source data, IFN-beta mRNA levels at 0 h were not detected under the experimental conditions. Therefore, statistical analysis at 0 h is not shown.

In Fig 1D, we have indicated p-values in the figure and described the statistical test in the figure legend. As shown in the source data, some of IFN-beta mRNA levels at 0 h were not detected under the experimental conditions. Therefore, statistical analysis at 0 h is not shown.

In Fig 1F, as shown in the figure, IFN-beta protein levels in the control samples, as determined by ELISA, were below the detection limit (not detected, ND), which has been stated in the figure legend. Therefore, the p-values are not included in Fig 1F.

Fig 1G also show the results of ELISA, and some IFN-beta protein levels in the control samples were below the detection limit. Therefore, the p-values for the comparison between the control vs pitavastatin-treated samples at MOI = 10 on day 1 day are shown only. We have stated in the figure legend that ND indicates “not detected”.

In Fig 1L, we have indicated p-values in the figure and described the statistical test in the figure legend.

In the original Fig 2D, G and H, NS (not significant) was omitted. In the revised Fig 2D, G, and H, we have now included NS where applicable.

In Fig 2I, we have included p-values and NS in the figure and described the statistical test in the figure legend.

In Fig 3A–F, H, we have included p values and NS in the figure and described the statistical test in the figure legend.

In Fig 4A and B, we have included p-values and NS in the figure. To avoid overcomplicating the figure with multiple comparisons, each p-value has been placed directly on the corresponding bar. We have also specified in the figure legend that each p-value represents the comparison with the control.

In Figure 4E and F, we have included p values and NS in the figure and described the statistical test in the figure legend.

In Figure 7E, we have included p values and NS in the figure and described the statistical test in the figure legend.

In all Fig EV and S, we have included p values and NS in the figure and described the statistical test in the figure legend.

Prof. Hiroyuki Oshiumi
Kumamoto University
Department of Immunology, Faculty of Life Sciences
1-1-1 Honjo
Kumamoto 860-8550
Japan

Dear Prof. Oshiumi,

I am very pleased to accept your manuscript for publication in the next available issue of EMBO reports. Thank you for your contribution to our journal.

Yours sincerely,
